# Bradley–Terry and Multi-Objective Reward Modeling Are Complementary

**Zhiwei Zhang**[1,2,*] **Hui Liu**[2], **Xiaomin Li**[3], **Zhenwei Dai**[2], **Jingying Zeng**[2], **Fali Wang**[1], **Minhua Lin**[1],
**Ram Chandradevan**[2], **Linlin Wu**[4], **Zhen Li**[2], **Chen Luo**[2], **Zongyu Wu**[1], **Xianfeng Tang**[2], **Qi He**[2]
**Suhang Wang**[1,†]
[1]The Pennsylvania State University    [2]Amazon    [3]Harvard University    [4]University of Utah
{zbz5349, szw494}@psu.edu

## Abstract

Reward models trained on human preference data have demonstrated strong effectiveness in aligning Large Language Models (LLMs) with human intent under the framework of Reinforcement Learning from Human Feedback (RLHF). However, RLHF remains vulnerable to reward hacking, where the policy exploits imperfections in the reward function rather than genuinely learning the intended behavior. Although significant efforts have been made to mitigate reward hacking, they predominantly focus on and evaluate in-distribution scenarios, where the training and testing data for the reward model share the same distribution. In this paper, we empirically show that state-of-the-art methods struggle in more challenging out-of-distribution (OOD) settings. We further demonstrate that incorporating fine-grained multi-attribute scores helps address this challenge. However, the limited availability of high-quality data often leads to weak performance of multi-objective reward functions, which can negatively impact overall performance and become the bottleneck. To address this issue, we propose a unified reward modeling framework that jointly trains Bradley-Terry (BT) single-objective and multi-objective regression-based reward functions using a shared embedding space. We theoretically establish a connection between the BT loss and the regression objective and highlight their complementary benefits. Specifically, the regression task enhances the single-objective reward function's ability to mitigate reward hacking in challenging OOD settings, while BT-based training improves the scoring capability of the multi-objective reward function, enabling a 7B model to outperform a 70B baseline. Extensive experimental results demonstrate that our framework significantly improves both the robustness and the scoring performance of reward models.

## 1 Introduction

Pretrained large language models perform exceptionally well on a wide range of tasks, including language understanding (Chang et al., 2024; Nam et al., 2024), text generation (Zhao et al., 2023; Wang et al., 2024b), code synthesis (Jiang et al., 2024; Nguyen et al., 2025), and decision-making (Ye et al., 2025; Lin et al., 2025). However, strong task performance alone does not ensure that these models behave safely or in alignment with human values (Dai et al., 2024; Li et al., 2025; Shen et al., 2023; Zhang et al., 2025b). To address these concerns, two main alignment methods have been developed. Supervised fine-tuning (SFT) adjusts a base model using human-curated prompt–response pairs to shape its behavior directly. RLHF (Bai et al., 2022; Ouyang et al., 2022) follows a two-step process: first, a proxy reward model is trained on human preference data to capture desired outcomes; second, the model is optimized against this reward using algorithms like PPO (Schulman et al., 2017), RLOO (Ahmadian et al., 2024), or GRPO (Shao et al., 2024). By decoupling reward learning from policy learning, RLHF can leverage vast amounts of unlabeled data and generalize alignment to novel inputs, enhancing both safety and overall model capabilities.

---

[*]Work done during internship at Amazon.
[†]Corresponding Author.

However, RLHF is susceptible to reward hacking (Gao et al., 2023; Yang et al., 2024), wherein the policy discovers shortcuts that maximize proxy rewards, such as by producing repetitive or formulaic content, without genuinely advancing the intended behaviors. Similar issues also arise in inference-time strategies like Best-of-N (BoN) sampling (Gulcehre et al., 2023; Dong et al., 2023; Gui et al., 2024). Prior research has investigated several directions to address this issue. One line of work focuses on improving the reward function through ensemble methods (Coste et al., 2024; Eisenstein et al., 2023; Zhang et al., 2024a; Yan et al., 2024; Ramé et al., 2024; Zhang et al., 2024b). However, they often require training multiple reward models, making them resource-intensive and less practical for applications. Another line investigates constrained policy optimization (Moskovitz et al., 2024; Zhang et al., 2024b; Liu et al., 2024b; Zhang et al., 2024c; Laidlaw et al., 2024), but performance is often unstable due to sensitivity to hyperparameter tuning. ODIN (Chen et al., 2024) trains separate reward functions for quality and length; however, our results (Fig. 2) show that using length alone as a biasing factor fails to prevent reward hacking. Recently, GRM(Yang et al., 2024; Dai et al., 2025) incorporates text generation regularization into reward modeling and outperforms prior methods. However, the conflicting objectives of reward modeling and text generation cause training instability and sensitivity to the balancing weight (Appendix J). Moreover, existing studies focus on in-distribution evaluations, and the effectiveness of these methods in OOD settings, where training and test prompt-response pairs come from different distributions, remains unexplored.

In this study, we first show experimentally that state-of-the-art methods fail when prompts used during PPO and BoN are drawn from a distribution different from the training data. This highlights a critical limitation in the generalization ability of current reward models under OOD settings. We hypothesize that a BT model trained only on chosen/rejected labels remains biased and cannot distinguish fine-grained quality differences. Inspired by recent multi-objective reward modeling methods (Wang et al., 2024e;d), which leverage annotations for attributes like helpfulness, verbosity, and correctness, we examine their potential to mitigate reward hacking. By learning from multi-dimensional supervision, multi-objective reward models (MORMs) capture nuanced distinctions in response quality and compel policy models to improve across all attributes simultaneously, making it harder to generate low-quality outputs that nonetheless score highly (Sec. 3). While prior work focused on interpretability and steerability, the robustness of MORMs in policy learning remains underexplored.

Despite their potential, the performance of MORMs is constrained by the limited availability of large-scale, high-quality annotated data (Wang et al., 2024e;d). This limitation arises either from the low-quality annotations produced by LLM-as-Judge (Cui et al., 2023; Kim et al., 2023; Li et al., 2024; Gu et al., 2024) or from the high-quality annotations produced by humans that are difficult to scale (Wang et al., 2024f). A more detailed discussion of the data availability challenge is provided in Appendix F. Consequently, their scoring performance often falls behind that of single-objective reward models (SORMs) (Lambert et al., 2024; Liu et al., 2025b), which are typically trained on large-scale preference datasets with chosen/rejected labels that are easier to collect. A promising approach is to train a strong SORM and complement it with an MORM to enhance robustness against reward hacking. Though our empirical results in Sec. 3 verify its promise, this approach faces two key issues: **(1)** it requires two independent inference passes, making the process computationally expensive in practice; and **(2)** the weaker performance of the MORM directly degrades the quality of the aggregated output, thereby becoming a bottleneck in the overall system, as shown in Fig. 2 (d).

Thus, we study a novel problem: *how to efficiently mitigate reward hacking in challenging OOD settings using fine-grained attribute scores, without additional costly multi-attribute preference data?* To address this, we propose a simple, yet effective and theoretically grounded reward modeling framework, termed the Joint Single and Multi-Objective Reward Model (**SMORM**). The proposed SMORM framework addresses the first challenge by requiring only a single forward pass through a shared backbone. We theoretically establish the connection between the commonly used Bradley–Terry (Bradley & Terry, 1952) loss for training the SORM and the regression loss used for training the MORM. Our theoretical analysis and extensive empirical results demonstrate that: **(1)** training the multi-objective head refines the embedding space such that the representations capture quality distinctions across multiple attributes, thereby *enhancing the generalizability of the single-objective head and improving its robustness against reward hacking.*; and **(2)** training the single-objective head helps correct the positioning of responses in the embedding space, *enabling the multi-objective head to perform competitively even with limited data.* Overall, the joint training of both heads over a shared embedding space leads to **complementary benefits**. Using the same multi-objective dataset,

SMORM enables a 7B model to outperform a 70B baseline model. Importantly, SMORM training is flexible in that the prompt-response pairs used to train the two heads do not need to be identical.

Our primary contributions are: **(1)** We empirically show that state-of-the-art reward models suffer from reward hacking during PPO in OOD settings—a critical issue largely overlooked in prior work. **(2)** We introduce SMORM, a training framework that, while simple in form, is *novel, non-trivial, and theoretically grounded.* **(3) This is the first work to provide theoretical analysis establishing a principled connection between Bradley–Terry preference modeling and multi-objective regression, revealing their complementary benefits.** **(4)** SMORM also tackles the key challenge of improving the performance of a multi-objective reward model without requiring additional, hard-to-obtain data.

## 2 BACKGROUND

**Bradley–Terry Single-Objective Reward Modeling.** Single-objective reward modeling typically builds on the Bradley-Terry framework (Bradley & Terry, 1952), which distinguishes between a chosen response $y_c$ and a rejected response $y_r$ for a given prompt $x$. This is achieved by optimizing the following loss function:

$$\min_\theta \mathcal{L}_{\text{reward}}(\theta) = -\mathbb{E}_{(x,y_c,y_r)\sim\mathcal{D}_s}\left[\log\left(\sigma\left(r_\theta(x,y_c) - r_\theta(x,y_r)\right)\right)\right], \tag{1}$$

where $r_\theta$ is the reward model parameterized by $\theta$, $r_\theta(x,y)$ denotes the reward score assigned by $r_\theta$ for the output $y$ given the prompt $x$, and $\sigma(\cdot)$ is the sigmoid function. Minimizing this loss encourages the model to assign higher scores to outputs that are preferred by humans.

**Multi-Objective Reward Modeling.** In many practical settings, evaluating language model outputs requires considering multiple aspects such as correctness, coherence, and verbosity. Single reward signals often fail to capture this complexity. To address this, multi-objective reward models (Wang et al., 2024e;d; 2023b; 2024c) generate separate reward signals for different response attributes. The model is trained as: $\min_\psi \mathcal{L}(\psi) = \mathbb{E}_{(x,y,\mathbf{r})\sim\mathcal{D}_M}\|R_\psi(x,y) - \mathbf{r}\|_2^2$, where $R_\psi$ is the multi-objective reward model and $\mathbf{r} \in \mathbb{R}^K$ denotes attribute scores (e.g., correctness, verbosity). Each dimension reflects a specific quality, enabling more interpretable and steerable evaluations. However, existing work has not examined how such models relate to reward hacking.

**Best-of-n Sampling (BoN).** Given an input $x$, BoN (Gulcehre et al., 2023; Dong et al., 2023; Gui et al., 2024) first draws a set $\mathcal{Y}_{\text{gen}}$ of $n$ candidate outputs from the policy model and then selects the one that maximizes the reward-model score. It can be applied either to improve outputs at inference time or to drive an iterative optimization procedure: $y_{\text{BoN}}(x) = \arg\max_{y\in\mathcal{Y}_{\text{gen}}} r_\theta(x,y)$.

**Proximal Policy Optimization (PPO).** PPO is a widely adopted method for RLHF in optimizing language models (Ouyang et al., 2022; Stiennon et al., 2020; Wu et al., 2024). Using a proxy reward model $r_\theta$, PPO refines the policy model $\pi_\phi$ by maximizing its score under the proxy reward while incorporating a KL divergence penalty: $\max_\phi \mathcal{L}_{\text{RLHF}}(\phi) :=$ $\mathbb{E}_{x\sim S}\left[\mathbb{E}_{y\sim\pi_\phi(\cdot|x)}\left(r_\theta(x,y)\right) - \lambda\cdot\text{KL}\left(\pi_\phi(\cdot|x)\,\|\,\pi_\phi^{\text{ref}}(\cdot|x)\right)\right].$ where $S$ is a training set of prompts and $\lambda \geq 0$ is a KL regularization that controls how much $\pi_\phi$ deviates from the initial policy $\pi_\phi^{\text{ref}}$.

**Reward Hacking.** Reward hacking occurs when a policy exploits flaws in the reward function, attaining high scores by overfitting to spurious patterns rather than accomplishing the intended task (Fu et al., 2025; Yang et al., 2024; Gao et al., 2023). In PPO, this manifests when the policy achieves higher scores from a proxy reward model but performs worse under a more reliable, human-aligned golden reward model. Additional discussion of related work and comparisons with existing approaches are provided in Appendix A.

## 3 SMORM: MITIGATING REWARD HACKING IN OOD SETTING

In this section, we conduct experiments to evaluate the robustness of existing SOTA methods and our proposed framework against reward hacking in both PPO and BoN under OOD settings, a more challenging setting largely overlooked in prior work (Yang et al., 2024; Coste et al., 2024; Eisenstein et al., 2023; Zhang et al., 2024a; Yan et al., 2024; Ramé et al., 2024; Dai et al., 2025).

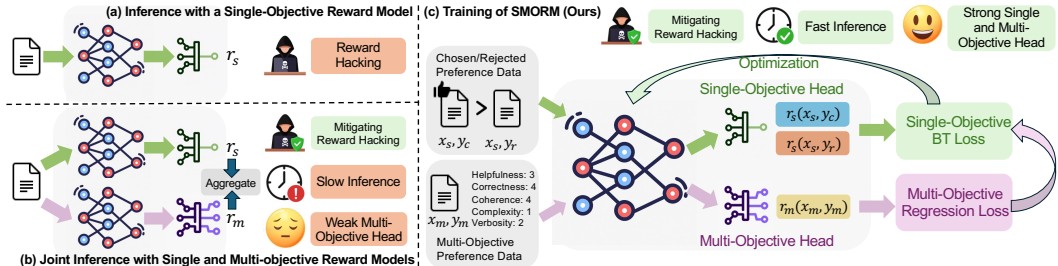

Figure 1: Illustration of SMORM training and its advantages over baseline methods. The training objective of SMORM is an original contribution, fundamentally differing from previous approaches and supported by theoretical analysis (see Appendix A for detailed comparisons).

Intuitively, a multi-objective reward model (MORM) evaluates response quality across multiple dimensions, forcing the policy to balance various attributes during generation. For example, it cannot maximize the overall score by optimizing helpfulness alone while neglecting verbosity. However, the performance of MORM is limited by the need for large-scale, high-quality multi-attribute labels, which are typically more difficult to collect than binary chosen/rejected labels (Wang et al., 2024f). This motivates us to design an ensemble framework that complements the stronger SORM with a MORM to enhance robustness against reward hacking, as shown in Fig. 1 (b). However, this framework encounters two major issues: (1) it requires two separate inference passes, leading to significant computational overhead; and (2) the weaker performance of the MORM degrades the quality of the aggregated output, becoming a bottleneck for the system, as verified in Fig. 2.

To overcome these limitations, we introduce a simple yet effective, theoretically grounded reward modeling framework called the Single and Multi-Objective Reward Model (**SMORM**). As shown in Fig. 1 (c), SMORM jointly trains a BT single- and multi-objective reward function based on a shared embedding space. It addresses the issue of efficiency by requiring only a single forward pass through the backbone. Furthermore, training the single-objective function along the chosen/rejected dimension helps shape the embedding space and correct the positioning of samples within it, thereby enabling the multi-objective head to achieve competitive performance with limited data. We will theoretically and empirically justify this effect in Section 4.

Specifically, given a pre-trained decoder-only LLM without the original output linear layer as the feature extractor $f_\theta$. We pass $x \oplus y$, the concatenation of $x$ and $y$, through the decoder layers and take the hidden state of the final decoder layer as a $d$-dimensional feature. On top of $f_\theta$, we attach two linear heads: a single-objective head with weights $\mathbf{w}_S \in \mathbb{R}^{d \times 1}$, which outputs a scalar rating, and a multi-objective head with weights $\mathbf{w}_M \in \mathbb{R}^{d \times k}$, which produces a $k$-dimensional vector of attribute scores. Given $\mathcal{D}_S = \{x_s, y_c, y_r\}$ as the chosen-rejected preference dataset and $\mathcal{D}_M = \{x_m, y_m, \mathbf{r}\}$ as the multi-attribute preference dataset, SMORM is trained with the following loss function:

$$\min_{\theta, \mathbf{w}_S, \mathbf{w}_M} -\mathbb{E}_{\mathcal{D}_S} \left[ \log \sigma(\mathbf{w}_S^\top ((f_\theta(x_s, y_c) - f_\theta(x_s, y_r)))) \right] + \mathbb{E}_{\mathcal{D}_M} \left\| \mathbf{w}_M^\top f_\theta(x_m, y_m) - \mathbf{r} \right\|_2^2 \quad (2)$$

**Although the objective may superficially resemble a simple combination of BT and regression losses, their joint training is non-trivial due to their fundamentally different forms. How these objectives influence one another has remained unexplored in prior work; we are the first to establish a theoretical connection and to show their complementary benefits (Sec. 4)..** The first head with weight $\mathbf{w}_S$ outputs a score along the chosen/rejected dimension, while the second head with weight $\mathbf{w}_M$ outputs scores on multiple attributes (e.g. helpfulness, coherence and verbosity). SMORM supports multiple inference strategies: SMORM-F uses only the first head to produce the reward score, SMORM-L computes the mean of the scores from the second head, and SMORM-M averages the scores from both the first and second heads.

**Experimental Setup.** For training SMORM, we use `Skywork80K` (Liu et al., 2024a) as $\mathcal{D}_S$ and `Helpsteer2` (Wang et al., 2024f) as $\mathcal{D}_M$. To train the multi-attribute head using only label information without extra data or domain knowledge, we filter $\mathcal{D}_M$ to retain only samples that also appear in $\mathcal{D}_S$. We compare `SMORM` against several baselines, including: (1) *Baseline Classifier*, trained using the original reward loss defined in Eq. 1; (2) *ODIN*, which trains two separate reward functions for quality and length (Chen et al., 2024). (3) *Baseline SM*, which trains a baseline SORM

and MORM separately and aggregates their results during inference and (4) *GRM* (Yang et al., 2024), which incorporates supervised fine-tuning (SFT) regularization. We use `gemma-2B-it` (Team et al., 2024) as the backbone model for all methods. In PPO experiment following (Yang et al., 2024), we downsample 20K samples from the `Unified-Feedback` dataset to optimize the PPO policy, reserving an additional 1K samples as a held-out test set for evaluation. Following prior work (Gao et al., 2023; Coste et al., 2023; Yang et al., 2024), we perform BoN sampling on this evaluation set, selecting the best of $n$ responses per prompt based on proxy model scores. The selected responses are then evaluated using the gold reward model, and gold scores are averaged across all prompts to assess true quality. We vary KL divergence from 0 to 5 by adjusting $n$ from 1 to 405, using the relation $\text{KL}_{\text{BoN}} = \log n - \frac{n-1}{n}$. For both experiments, `gemma-2B-it` serves as the policy model, and the gold reward model [1] is a 7B human-preference model fine-tuned on the full `Unified-Feedback` dataset.

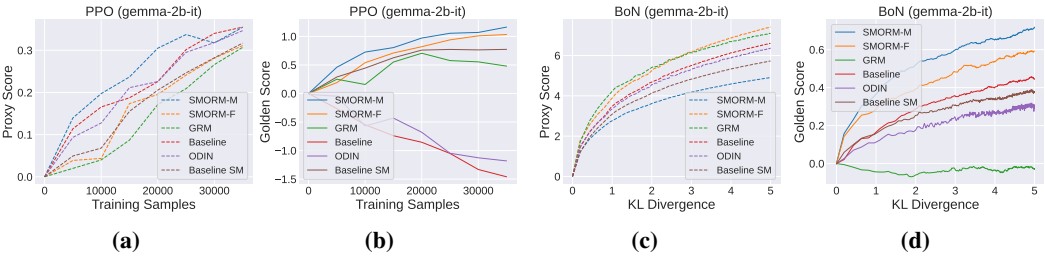

Figure 2: Proxy and gold scores from (a)(b) PPO and (c)(d) BoN experiments under the OOD setting, using gemma-2B-it as the base model. All rewards are normalized to start from 0.

The results of the PPO and BoN experiments are presented in Fig. 2 (a)(b) and (c)(d), respectively. From these figures, we observe the following: **(1) Baselines Fail to Mitigate Reward Hacking.** We observe that although GRM demonstrates increased golden scores during PPO, its average gold scores decrease as KL divergence increases in the BoN setting, suggesting reward hacking. ODIN shows a modest increase in golden scores under BoN, but a decrease during PPO, indicating that addressing length alone as a bias is insufficient to mitigate reward hacking effectively. **(2) Incorporating Multi-Attribute Scores Helps Mitigate Reward Hacking in OOD Setting.** In both experimental settings, we find that the Baseline SM outperforms both GRM and ODIN, exhibiting a steady increase in golden scores as the proxy score increases. **(3) Weak Multi-Objective Reward Functions Become a Bottleneck.** While Baseline SM shows promise, its performance in the BoN experiment is even worse than that of the baseline single-objective reward function. This suggests that a weak multi-objective reward function can be detrimental and become a bottleneck in the overall system. **(4) SMORM Shows Superior Performance.** Both SMORM-F and SMORM-M significantly outperform all baselines across both experiments. Notably, SMORM-F performs comparably to SMORM-M. To theoretically justify this phenomenon, we propose the following theorem:

**Theorem 1** (Implicit Multi-Attribute Effect). *Let a reward model be trained under the SMORM framework, and suppose the following conditions hold: (1) Bounded features: There exists $B < \infty$ such that $\|f_\theta(x, y)\| \leq B$ for every $(x, y)$. (2) Positive-definite covariances: let $f_c = f_\theta(x_s, y_c)$, $f_r = f_\theta(x_s, y_r)$, $f_m = f_\theta(x_m, y_m)$. $\Sigma_S := \mathbb{E}_{\mathcal{D}_S}[(f_c - f_r)(f_c - f_r)^\top]$ and $\Sigma_M := \mathbb{E}_{\mathcal{D}_M}[f_m f_m^\top]$ are positive-definite matrices. (3) Positive correlation: Let $\mu_S := \mathbb{E}_{(x_s, y_c, y_r) \sim \mathcal{D}_S}[f_\theta(x_s, y_c) - f_\theta(x_s, y_r)]$ and let $C_M := \mathbb{E}_{(x_m, y_m, r) \sim \mathcal{D}_M}[f_\theta(x_m, y_m) \, \mathbf{r}^\top] \in \mathbb{R}^{d \times K}$. Then $\alpha := \mu_S^\top \Sigma_M^{-1} C_M$ has non-negative sum, i.e. $\mathbf{1}^\top \alpha \geq 0$. As the optimization of both reward heads converge to their population minimizers, there exist constants $c = \frac{\mathbf{1}^\top \alpha}{K \left( \mu_S^\top \Sigma_S^{-1} \mu_S \right)}$ and $\varepsilon \geq 0$—depending only on $B$ and second-order moments—such that for every pair $(x, y)$:*

$$r_m(x, y) = \frac{1}{K} \sum_{i=1}^{K} w_{M,i}^\top f_\theta(x, y) \geq c \left( w_S^\top f_\theta(x, y) \right) - \varepsilon = c r_s(x, y) - \varepsilon. \quad (3)$$

A generalized theory that explicitly accounts for data size is presented in Theorem 3 in the Appendix. We provide detailed justifications of the assumptions and the proofs in the Appendix D.2. The

---

[1] Ray2333/reward-model-Mistral-7B-instruct-Unified-Feedback

theorem is proved under the assumption that the aggregated attribute scores in the multi-objective preference data are positively correlated with the score provided by a pretrained single-objective reward function $\mathbf{w}_S$. This is well-justified: $\mathbf{w}_S$ can be trained on large-scale, high-quality preference data and thus provides reliable scoring. Moreover, multi-objective annotations typically follow the principle that chosen responses should have higher aggregated scores than rejected ones, and thus a positive correlation can be reasonably expected. Notably, this assumption is general and does not require the training data for the two heads to share the same prompt–response pairs or domain.

From this theorem, we derive two conclusions: **(1)** If a response achieves a high single-objective score $r_s \geq \tau$, its multi-attribute average score $r_m$ is lower-bounded by $c\,\tau - \varepsilon$. This result implies that even when the multi-attribute head is ignored, a high single-objective score alone ensures a respectable level of fine-grained quality. This explains why policies trained with SMORM-F attain performance comparable to those trained with SMORM-M. **(2)** For any two responses $y_A$ and $y_B$ to the same prompt $x$, if $r_s(x, y_A) > r_s(x, y_B)$, then it follows that $c\,r_s(x, y_A) - \varepsilon > c\,r_s(x, y_B) - \varepsilon$. That is, a higher single-objective score implies a strictly higher lower bound on the corresponding multi-attribute score $r_m$. This enables training a strong single-objective function from abundant preference data and leveraging it to guide the multi-objective function, without the need for costly fine-grained multi-attribute annotations. Empirical results in Sec. 5.2 confirm this conclusion.

## 4 SUPERIOR SCORING PERFORMANCE OF SMORM

We have empirically and theoretically demonstrated that SMORM effectively mitigates reward hacking, even under challenging OOD scenarios. Although Theorem 1 establishes that SMORM can implicitly leverage a stronger single-objective reward function to guide the multi-objective one, a potential concern remains: joint training may degrade the performance of that single-objective function. To address this concern, we begin this section by introducing a lemma that connects Bradley–Terry reward modeling with multi-objective reward regression. We then present a theorem demonstrating that SMORM improves the performance of both reward functions by learning a more effective feature extractor. Unlike ODIN (Chen et al., 2024), which trains reward functions for both quality and length using the BT loss, the interaction between BT loss and MSE regression loss when sharing a common embedding space is not straightforward to characterize. Therefore, we propose the following lemma to bridge this gap.

**Lemma 1** (Pairwise Preference Error to MSE Loss). *Let $y_A, y_B$ be a pair of responses. Assume $g_s(y)$ is the ground truth score and $r_s(y)$ is the predicted score under a Bradley–Terry model. Then:*

$$\mathbb{P}(y_A \succ y_B) = \sigma\big(r_s(y_A) - r_s(y_B)\big), \quad \mathbb{P}^\star(y_A \succ y_B) = \sigma\big(g_s(y_A) - g_s(y_B)\big),$$

*where $\sigma(t) = \frac{1}{1+e^{-t}}$. The expected preference error satisfies:*

$$\mathbb{E}_{\mathcal{D}_S} |\mathbb{P}(y_A \succ y_B) - \mathbb{P}^\star(y_A \succ y_B)| \leq \frac{1}{4}\mathbb{E}_{\mathcal{D}_S}\left(\sqrt{2\,\mathit{MSE}(r_s)}\right), \tag{4}$$

*with $\mathit{MSE}(r_s) = \big(r_s(y) - g_s(y)\big)^2$.*

**Lemma 1 shows that the expected BT loss is upper bounded by the MSE, i.e., the square of the regression loss**. With this lemma, we propose the following theorem:

**Theorem 2.** *Under the same assumptions as in Theorem 1 and assuming that the feature extractor $f_\theta$ is differentiable, let $\widehat{\theta}$ denote the maximum likelihood estimator (MLE) of the ground truth optimal parameter $\theta^\star$. Let $\widehat{\theta}_s$ and $\widehat{\theta}_m$ denote the maximum likelihood estimators of the single- and multi-objective reward functions, respectively. Define $M_S(y) = \mathbf{w}_S^\top f_{\theta^\star}(y)$, $M_M(y) = \mathbf{w}_M^\top f_{\theta^\star}(y)$. Then, for a response $y$, the mean squared error (MSE) of the predicted reward can be approximated as:*

$$\mathrm{MSE}_S \approx \nabla_\theta M_S(y)^\top \mathrm{Cov}\left(\widehat{\theta}_s\right) \nabla_\theta M_S(y) + \sigma_{00}, \mathrm{MSE}_M \approx \nabla_\theta M_M(y)^\top \mathrm{Cov}\left(\widehat{\theta}_m\right) \nabla_\theta M_M(y) + \sigma_{00},$$

*where $\sigma_{00}$ is the intrinsic randomness in the label. Moreover, SMORM yields lower asymptotic MSE for both the single- and multi-objective heads compared to training either head alone:*

$$\mathrm{MSE}_S^{SMORM} < \mathrm{MSE}_S^{single}, \quad \mathrm{MSE}_M^{SMORM} < \mathrm{MSE}_M^{multi} \tag{5}$$

Detailed proofs of the above lemma and theorem are provided in Appendix D. By Lemma 1 and Theorem 2, a reduction in MSE directly tightens the bound on pairwise preference prediction error. **This is the first theoretical guarantee that a shared BT–regression architecture is strictly superior to training the two heads independently, a connection never explored in prior work**. See Appendix A for details highlighting the originality and significance of our theoretical contribution.

## 5 EXPERIMENTS

In this section, we present a comprehensive evaluation of SMORM. We conduct experiments under both in-distribution and out-of-distribution settings to assess how SMORM improves scoring capability and mitigates reward hacking. Our results demonstrate the following advantages of SMORM: **(1) Enhancing scoring performance**. SMORM significantly improves the scoring capability of the multi-objective reward function without requiring additional multi-attribute preference data, which are often difficult to collect (Sec. 5.2). **(2) Mitigating reward hacking**. SMORM substantially enhances the robustness of the single-objective reward function against reward hacking in both ID and challenging OOD settings (Sec. 5.3), while also improving its scoring performance (Sec. 5.2). **(3) Flexible training**. SMORM does not require the single-objective dataset $\mathcal{D}_S$ and the multi-objective dataset $\mathcal{D}_M$ to share the same prompt-response pairs, allowing for a more flexible training process.

### 5.1 EXPERIMENTAL SETUP

**Datasets and Benchmarks.** In Section 5.2, we use the `Unified-Feedback` dataset as the single-objective preference dataset $\mathcal{D}_S$, one of the largest collections of pairwise human preferences. To assess robustness across data scales, we evaluate two settings: one with 400K samples from $\mathcal{D}_S$ and another with 40K. For the multi-objective dataset $\mathcal{D}_M$, we use `UltraFeedback` (Cui et al., 2023) in the 400K setup, which includes 240K GPT-annotated prompt–response pairs with fine-grained attribute scores. In the 40K setup, we use `HelpSteer2` (Wang et al., 2024f), a high-quality human-annotated dataset with 20K samples. SORM is trained only on $\mathcal{D}_S$, MORM on $\mathcal{D}_M$, and SMORM jointly on both. The trained reward models are evaluated on RewardBench (Lambert et al., 2024) and RM-Bench (Liu et al., 2025b). In the RLHF experiments (Section 5.3), we consider both ID and OOD settings. For ID, we downsample 20K samples from `Unified-Feedback` as $\mathcal{D}_S$ to train reward models and the PPO policy. For OOD, we use `Skywork80K` (Liu et al., 2024a) as $\mathcal{D}_S$. In both cases, `HelpSteer2` serves as $\mathcal{D}_M$, and 1K samples from `Unified-Feedback` are reserved for policy evaluation.

**Base Models.** Following (Yang et al., 2024), we adopt gemma-2B-it (Team et al., 2024) and Mistral-7B-Instruct-v0.2 (Jiang et al., 2023) as the base models for preference learning. For the RLHF experiments, gemma-2B-it serves as the policy model in both the BoN and PPO settings. The gold reward model is a 7B human preference model fine-tuned on the entire Unified-Feedback dataset.

**Baselines.** We adopt representative and SOTA baselines: (1) Baseline Classifier, trained using the original reward loss as defined in Eq. 1; (2) Margin (Touvron et al., 2023), which augments the original reward loss with an additional margin term; (3) Label Smooth (Wang et al., 2024a), which addresses overfitting by penalizing overconfident predictions; (4) Ensemble, which combines the outputs of three reward models by taking either the average or the minimum score (Coste et al., 2023); and (5) GRM(Yang et al., 2024) with two types of regularization: GRM w/ dpo and GRM w/ sft. Details on baselines and implementation are in Appendix I.

**RLHF.** The training and evaluation pipeline of PPO and BoN follow the setting in Sec. 3.

### 5.2 EVALUATION ON REWARD MODELING

**Comparison to Single-Objective Reward Model.** Table 1 compares SMORM-F with baseline SORMs on RewardBench using gemma-2B-it as the base model; results with Mistral-7B-Instruct are reported in Appendix C.6. SMORM-F consistently achieves the highest average performance across all settings, supporting Theorem 2, which shows that joint training yields superior performance over training single-objective reward functions in isolation. Additional OOD evaluation results for SMORM-F are provided in Appendix G.

Table 1: Comparison of SMORM-F and baselines on RewardBench.

| Reward model | $\mathcal{D}_S/\mathcal{D}_M$: UnifiedFeedback 400k/UltraFeedback | | | | | $\mathcal{D}_S/\mathcal{D}_M$: UnifiedFeedback 40k/HelpSteer2 | | | | |
|---|---|---|---|---|---|---|---|---|---|---|
| | Chat | Chat-Hard | Safety | Reasoning | Avg | Chat | Chat-Hard | Safety | Reasoning | Avg |
| Baseline (Single) | 95.5 | 38.0 | 73.8 | 65.3 | 68.2 | 94.7 | 37.5 | 66.2 | 58.4 | 64.2 |
| Baseline + margin | 95.8 | 38.4 | 73.9 | 72.5 | 70.2 | 97.2 | 37.5 | 56.8 | 72.7 | 66.1 |
| Label smooth | 94.4 | 37.3 | 73.2 | 77.4 | 70.6 | 91.6 | 39.0 | 53.8 | 60.2 | 61.1 |
| Ensemble | 98.0 | 37.5 | 77.3 | 71.3 | 71.0 | 96.1 | 38.2 | 58.8 | 67.6 | 65.2 |
| GRM (linear) w/ dpo | 96.7 | 39.0 | 76.4 | 68.5 | 70.2 | 94.7 | 38.4 | 62.5 | 51.2 | 61.7 |
| GRM (linear) w/ sft | 96.1 | 40.1 | 80.3 | 69.3 | 71.5 | 94.7 | 40.8 | 65.4 | 77.0 | 69.5 |
| GRM w/ dpo | 95.8 | 40.1 | 78.7 | 66.2 | 70.2 | 92.5 | 39.9 | 72.5 | 61.4 | 66.6 |
| GRM w/ sft | 97.8 | 42.1 | 77.9 | 65.2 | 70.8 | 94.1 | 41.9 | 69.5 | 61.5 | 66.8 |
| SMORM-F | 96.1 | 45.5 | 78.8 | 70.9 | **72.8** | 96.1 | 44.1 | 81.1 | 62.7 | **71.0** |

Table 2: Comparison of SMORM-L and baseline MORM on RewardBench and RM-Bench.

| Reward model | Chat | Chat Hard | Safety | Reasoning | RewardBench | RM-Bench |
|---|---|---|---|---|---|---|
| Base Model: Gemma 2b it, $\mathcal{D}_S/\mathcal{D}_M$: UnifiedFeedback 400k/UltraFeedback | | | | | | |
| Baseline (Multi) | 64.2 | 50.0 | 46.1 | 42.3 | 50.6 | 50.2 |
| SMORM-L | 90.8 | 48.3 | 61.5 | 53.7 | **63.6** | **55.1** |
| Base Model: Gemma 2b it, $\mathcal{D}_S/\mathcal{D}_M$: UnifiedFeedback 40k/HelpSteer2 | | | | | | |
| Baseline (Multi) | 84.1 | 39.0 | 57.4 | 42.6 | 55.8 | 51.8 |
| SMORM-L | 94.9 | 39.3 | 75.8 | 51.4 | **65.4** | **54.1** |
| Base Model: Mistral 7b Instruct, $\mathcal{D}_S/\mathcal{D}_M$: UnifiedFeedback 400k/UltraFeedback | | | | | | |
| Baseline (Multi) | 95.5 | 65.1 | 68.6 | 75.9 | 76.3 | 57.2 |
| SMORM-L | 97.8 | 61.0 | 86.4 | 77.7 | **80.7** | **62.6** |
| Base Model: Mistral 7b Instruct, $\mathcal{D}_S/\mathcal{D}_M$: UnifiedFeedback 40k/HelpSteer2 | | | | | | |
| Baseline (Multi) | 71.8 | 60.1 | 54.3 | 78.0 | 66.0 | 52.0 |
| SMORM-L | 94.4 | 61.8 | 83.6 | 79.7 | **79.9** | **64.4** |

**Comparison to Multi-Objective Reward Model.** Table 2 compares SMORM-L with baseline MORMs on RewardBench and RM-Bench. With Mistral-7B-Instruct as the base model and 40K samples from `UnifiedFeedback`, SMORM achieves gains of **+13.9** on RewardBench and **+12.4** on RM-Bench. These results validate Theorem 1, which shows that BT modeling corrects response positioning so that the multi-objective score is lower-bounded by the single-objective score, and further support Theorem 2, confirming that SMORM produces reward models that consistently outperform their baselines.

**Comparison to Advanced Reward Models.** We follow (Wang et al., 2024d) and train a gating network that assigns weights to each attribute during inference. Our 7B model is initialized from Mistral-7B-Instruct-v0.2 (Jiang et al., 2023) and trained on $\mathcal{D}_S/\mathcal{D}_M$: `UnifiedFeedback` (40K) and `HelpSteer2` (20K). The 8B model is initialized from Llama-3.1-8B-Instruct (Grattafiori et al., 2024) and trained on `Skywork80K` (Liu et al., 2024a) and `HelpSteer2` (20K). In both cases, the gating network is trained with BT loss on $\mathcal{D}_S$. We compare against large multi-objective reward models trained on `HelpSteer2` with tuned attribute weights (Wang et al., 2024f), and ArmoRM-Llama3-8B-v0.1 (Wang et al., 2024d), which is trained on far more data: 585.4K samples for its multi-objective head and 1,004.4K for its gating network, versus only 20K, 40K, and 80K for ours. As shown in Table 3, our 7B model underperforms the 340B baseline but surpasses the 70B model. Remarkably, our 8B model matches ArmoRM-Llama3-8B-v0.1 despite using **15.9×** less data.

Table 3: Comparison with Advanced Multi-Objective Reward Models on RewardBench.

| Reward model | Size of $\mathcal{D}_M$ | Model Size | Chat | Chat Hard | Safety | Reasoning | Avg |
|---|---|---|---|---|---|---|---|
| Nemotron-4-340B-RM (Wang et al., 2024f) | 20K | 340B | 92.0 | 95.8 | 87.1 | 91.5 | 93.7 |
| ArmoRM-Llama3-8B-v0.1 (Wang et al., 2024d) | 585.4K | 8B | 96.9 | 76.8 | 90.5 | 97.3 | 90.4 |
| Llama-3-70B-RM (Wang et al., 2024f) | 20K | 70B | 91.3 | 80.3 | 92.8 | 90.7 | 88.8 |
| SMORM-L 7B (Ours) | 20K | 7B | 95.0 | 80.5 | 91.6 | 89.0 | 89.0 |
| SMORM-L 8B (Ours) | 20K | 8B | 94.7 | 85.1 | 90.5 | 91.3 | 90.4 |

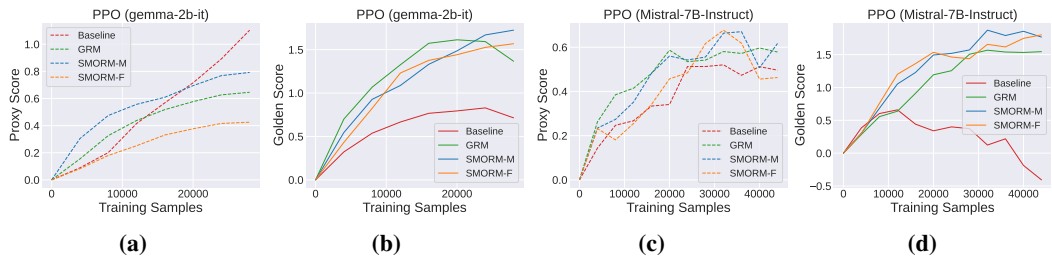

Figure 3: Proxy scores and gold scores of PPO experiments for reward model based on (a)(b) gemma2b-it and (c)(d) Mistral-7B-Instruct. All rewards are normalized to start from 0.

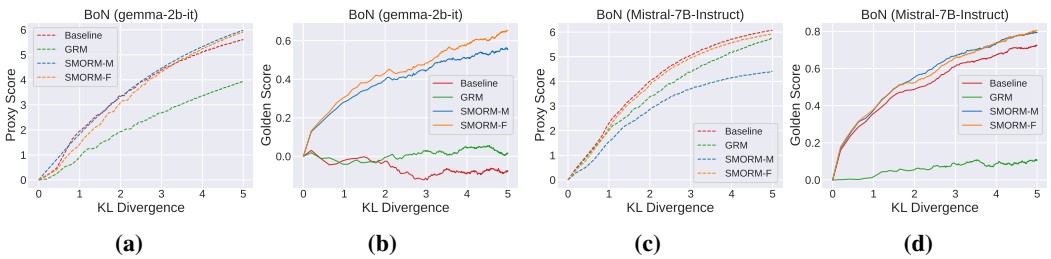

Figure 4: Proxy scores and gold scores of BoN experiments for base models of (a)(b) gemma-2b-it and (c)(d) Mistral-7B-Instruct. Rewards are normalized to start from 0.

## 5.3 EVALUATION ON RLHF

**In-Distribution Setting.** Figures 3 and 4 present the results of PPO and BoN under the in-distribution setting. For *PPO*, we observe that the gold score of the baseline reward function increases slowly in the early stages of training but subsequently declines, while the proxy score continues to rise. This pattern indicates reward overoptimization. In contrast, GRM exhibits a rapid initial increase in gold score, followed by a similar decline trend. Notably, both SMORM-F and SMORM-M show a consistent increase in gold score throughout training. SMORM-F achieves performance comparable to SMORM-M, which further supports Theorem 1 regarding the implicit benefit of joint training. In the *BoN* experiments, when using gemma-2B-it as the base model, we find that the gold score of the baseline reward function decreases as the KL divergence increases. For GRM, the gold score increases slowly and then plateaus. Both SMORM-F and -M show a consistent increase in gold scores across KL ranges, demonstrating the robustness of our framework in the ID setting.

**Out-of-Distribution Setting.** Figure 5 presents PPO and BoN results under the OOD setting using Mistral-7B-Instruct as the base model. Results with gemma-2B-it were shown earlier in Section 3. While the baseline reward function and GRM do not exhibit clear reward hacking with the stronger model in the OOD setting, the performance gap between SMORM and GRM becomes more pronounced compared to the in-distribution results in Figure 3. This underscores the limitations of existing methods and demonstrates the robustness of SMORM, especially under challenging OOD conditions. Additional results comparing PPO-optimized models are provided in Appendix H.

## 6 CONCLUSION

In this paper, we propose an effective and theoretically grounded method that jointly trains a Bradley–Terry reward function and a multi-objective reward function using a shared embedding space. Our theoretical analysis and extensive empirical results demonstrate that this joint training approach enhances the robustness of the BT function against reward hacking and significantly improves the scoring performance of the multi-objective function, revealing their complementary benefits.

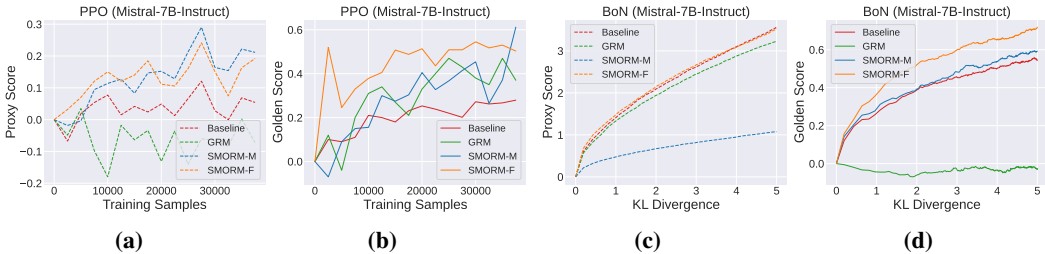

Figure 5: Proxy and gold scores from (a)(b) PPO and (c)(d) BoN experiments under the OOD setting.

## ACKNOWLEDGMENT

This material is based upon work supported by, or in part by the Army Research Office (ARO) under grant number W911NF-21-10198 and the Department of Homeland Security (DHS) under grant number 17STCIN00001-05-00. The views and conclusions contained in this material are those of the authors and should not be interpreted as necessarily representing the official policies, either expressed or implied, of the funding agencies.

## ETHICS STATEMENT

This work does not involve human subjects or the collection of new datasets. Experiments use established corpora and benchmarks under their licenses.

## REPRODUCIBILITY STATEMENT

We provided a detailed description of experimental settings in Sec. 5.

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

## A    RELATED WORK AND KEY DIFFERENCES FROM PRIOR APPROACHES

In this section, we provide additional related work and clarify how SMORM differs from existing approaches.

**Reward Modeling.** Reward models are designed to produce preference-aligned signals that guide the behavior of language models. In the context of large language models (LLMs), they act as proxies for human preferences, providing feedback to the policy model during the alignment process (Ouyang et al., 2022; Bai et al., 2022; Dong et al., 2024; Wen et al., 2025; Dong et al., 2024; Razin et al., 2025; Lin et al., 2023; Sun et al., 2025). These models are typically constructed by attaching a classification head to a pretrained LLM, allowing it to assign scores to responses conditioned on prompts (Zhu et al., 2023; Adler et al., 2024). To ensure alignment with principles such as helpfulness, harmlessness, and honesty, reward models are fine-tuned using human preference datasets (Wu et al., 2023; Guo et al., 2023; Dai et al., 2024; Chiang et al., 2024; Nakano et al., 2021). The resulting reward signals are then used in policy optimization, enhancing LLM performance on complex downstream tasks, including mathematical reasoning (Shao et al., 2024; Wang et al., 2023a; Luo et al., 2024; Zhang et al., 2025a).

**Mitigating Reward Hacking in RLHF.** Reward hacking in reinforcement learning from human feedback (RLHF) arises when the policy model exploits imperfections in the reward model, thereby failing to learn the intended behaviors. A variety of strategies have been proposed to address this issue. One line of work focuses on enhancing the reward function using ensemble methods (Coste et al., 2024; Eisenstein et al., 2023; Zhang et al., 2024a; Yan et al., 2024; Ramé et al., 2024; Zhang et al., 2024b). While effective, these approaches typically require training multiple reward models, making them computationally expensive and less feasible for real-world deployment. Another approach investigates constrained policy optimization (Moskovitz et al., 2024; Zhang et al., 2024b; Liu et al., 2024b; Zhang et al., 2024c; Laidlaw et al., 2024; Zhu et al., 2024). However, these methods often suffer from performance instability due to their sensitivity to hyperparameter tuning. More recently, GRM (Yang et al., 2024; Dai et al., 2025) incorporates text generation regularization into reward modeling and achieves superior performance compared to prior methods. Nonetheless, the inherent conflict between reward modeling and generation objectives introduces training instability and increases sensitivity to the choice of balancing weights.

**Multi-objective reward modeling.** (Pitis et al., 2024) demonstrates that reward models benefit from explicitly encoding conversational or task-specific context, mitigating preference inconsistencies under varying prompt conditions. (Shen et al., 2025) introduces mixture-of-experts routing to capture heterogeneous user preferences, highlighting that a single global reward model often fails to generalize across diverse human behaviors. (Luo et al., 2025) proposes a PCA-based analysis of preference directions, showing that many reward-model failures stem from collapsed or poorly estimated preference subspaces. (Poddar et al., 2024) adopts a probabilistic generative framework to model user-specific latent preference factors, enabling personalized RLHF policies that better reflect individual annotators' reward structures. These works collectively underscore that preference distributions are often multimodal and context-dependent, and they motivate our focus on analyzing robustness and over-optimization in reward modeling under prompt and model distribution shifts.

### ORIGINALITY AND NOVELTY OF SMORM

ODIN (Chen et al., 2024) is a framework that trains two reward functions for response quality and length, sharing a common embedding space. However, our work differs substantially from ODIN in several key aspects: **(1)** While ODIN uses the BT loss for both heads, we theoretically establish a connection between BT loss and multi-attribute regression loss. This advancement enables the integration of multiple fine-grained attributes beyond just response length. **(2)** Our work explicitly investigates policy optimization with PPO in out-of-distribution (OOD) settings, a scenario largely overlooked by existing studies. Our empirical results in Sec. 3 also demonstrate that ODIN fails in this setting. **(3)** Whereas ODIN focuses solely on mitigating reward hacking, we additionally uncover a complementary benefit: training a single-objective reward function significantly enhances the scoring capability of the multi-objective reward model.

RRM (Liu et al., 2025a) is a general framework that employs a causal approach to learn preferences independent of spurious artifacts via data augmentation during reward model training. However, **(1)** the data augmentation strategy significantly increases the computational cost of training a reward

model; **(2)** it remains unclear how to adapt such augmentation methods to multi-objective reward modeling, especially when labels consist of fine-grained scores; and **(3)** its effectiveness in mitigating reward hacking in RLHF settings has yet to be empirically validated. Similarly, (Wang et al., 2025a) propose a causal reward modeling approach that incorporates causal inference to reduce spurious correlations. InfoRM (Miao et al., 2024) introduces a variational information bottleneck objective to filter out irrelevant information during reward modeling. However, similar to GRM, this objective is fundamentally at odds with the goal of accurate reward modeling, and the performance of InfoRM on RewardBench has not been evaluated. In contrast, beyond addressing reward hacking, a key contribution of our work is enhancing the performance of weak multi-objective reward models without requiring additional preference data.

Our SMORM introduces significant methodological and empirical innovations compared to (Wang et al., 2025b), for the following reasons:

**(1) Significant Differences in the Training Paradigm.** The training paradigm in (Wang et al., 2025b) is *sequential*: either initializing a BT reward model from a regression reward model or vice versa. In contrast, **SMORM** adopts a *joint training* paradigm, where both the BT and regression reward functions are optimized simultaneously at each training step. This co-training leads to synergistic learning and mutual improvement between the two reward heads, rather than a one-directional transfer.

**(2) Significant Differences in Empirical Discoveries.** We summarize the empirical findings of [1] and contrast them with our work below:

| Method | BT → Regression | Regression → BT | Co-Training |
|---|---|---|---|
| Paper (Wang et al., 2025b) (Scaled Preference + Multi-Attribute) | Worse than random (93.0 → 92.2) | Better than random (91.5 → 92.7) | – |
| Paper (Wang et al., 2025b) (Pairwise Preference + Multi-Attribute) | – | Worse than random (93.0 → 92.9) | – |
| **SMORM** (Pairwise + Multi-Attribute) | – | – | Mutual improvement of both heads |

Specifically:

- (Wang et al., 2025b) does not explore co-training. Their conclusions are limited to initializing BT models with regression weights in specific datasets.

- Even in that case, when using pairwise preferences (e.g., chosen/rejected labels), performance **decreases** after initialization—suggesting instability or incompatibility.

- **Our finding is novel**: through *joint co-training* on pairwise preferences (BT head) and multi-attribute regression labels (regression head), we observe **mutual benefits** during training. These results indicate the two heads help each other learn better representations.

**(3) Generalization Beyond Dataset Constraints.** In (Wang et al., 2025a), both reward heads are trained on the same prompt-response pairs. In contrast, our SMORM framework is **explicitly designed to remove this constraint**. That is, the BT and regression heads can be trained on *different* prompt-response distributions, a generalization supported both:

- **Theoretically** – by our formal analysis in Sec. 4.

- **Empirically** – with experiments in Sec. 5 that demonstrate robust performance even when the heads are trained on distinct datasets.

Moreover, (Wang et al., 2025b) only considers a single attribute (e.g., Helpfulness) in regression. We extend this to a true **multi-attribute regression setting**, both in theory and in practice, making SMORM applicable in more realistic, attribute-rich environments.

## B  SIGNIFICANCE OF OOD CHALLENGE

In our paper, we define prompt-response pair distribution shift as an out-of-distribution (OOD) setting, where the prompt-response pairs used to train the reward model and those encountered during online RLHF are drawn from different datasets. Prior works on reward hacking typically assume an in-distribution setting, where both the reward model training data and the policy optimization data (e.g., for PPO) are sampled from the same distribution.

Furthermore, this shift arises not only from novel prompts, but also from responses that evolve as the policy is optimized. After just a few PPO updates, the model may begin to generate responses that move beyond the distribution of the offline preference data, as shown in (Dai et al., 2025).

Moreover, while the base LLM benefits from broad coverage due to Internet-scale pretraining, the reward head is typically fine-tuned on a much narrower preference dataset. As a result, the inductive bias of the reward model is governed by this limited supervision, leaving significant room for generalization error when the policy encounters unfamiliar regions of the output space.

Although datasets like UltraFeedback contain over 250K prompts, training a reward model on the full corpus is computationally expensive. Our preliminary results (see Figure 2) show that even when the reward model is trained on a high-quality subset such as Skywork-80K—which achieves strong pairwise comparison performance—the model still exhibits OOD vulnerabilities. This illustrates a fundamental trade-off: smaller high-quality datasets offer efficient training but limited coverage, making them susceptible to OOD issues. Therefore, under the realistic and common scenario of limited training data and compute resources, OOD challenge is both prevalent and critical to address.

## C  ADDITIONAL EXPERIMENTS

### C.1  TRAINING CURVES

In this section, we present experiments comparing our SMORM model with the GRM framework and the baseline multi-objective regression model, focusing on both training dynamics and evaluation performance.

Specifically, we fine-tune the base model meta-llama/Llama-3.1-8B-Instruct using two datasets: the preference dataset Skywork/Skywork-Reward-Preference-80K-v0.2 and the multi-objective reward dataset nvidia/HelpSteer2.

For comparisons against GRM, we report evaluation results on 10K samples drawn from Skywork/Skywork-Reward-Preference-80K-v0.2 as well as 10K out-of-distribution samples from llm-blender/Unified-Feedback to assess robustness.

For comparisons against baseline multi-objective reward models, we evaluate on both the training and validation splits of nvidia/HelpSteer2. We report metrics on the five granular attributes—helpfulness, correctness, coherence, complexity, and verbosity—using both mean squared error (MSE) and pairwise preference accuracy. To construct a pairwise validation set, we take pairs of responses for the same prompt and derive ground-truth preference labels for each attribute by directly comparing their human-annotated attribute scores.

The results comparing SMORM-F to GRM are shown in Fig. 6. From the figure, we observe the following: (1) As shown in subfigure (a), although SMORM converges more slowly than GRM during the early training stage—due to the need to jointly optimize both the Bradley–Terry loss and the regression loss—its training loss drops sharply after around 100 steps and eventually becomes lower than that of GRM. This indicates that incorporating the regression loss helps the BT loss converge more effectively. (2) Subfigure (b) shows that SMORM-F attains comparable in-domain performance relative to GRM. (3) In contrast, subfigure (c) illustrates that SMORM achieves substantially better out-of-distribution performance—improving by approximately 20 points on the UnifiedFeedback

dataset. This demonstrates that SMORM notably enhances the generalizability of the trained reward model.

The results comparing SMORM to the baseline multi-objective reward model are shown in Fig. 7 (training MSE curves), Fig. 8 (training pairwise preference accuracy), Fig. 9 (evaluation MSE), and Fig. 10 (evaluation pairwise preference accuracy). From these results, we observe the following: (1) As shown in both Fig. 7 and Fig. 9, although SMORM exhibits a higher MSE at the early stage of training, its MSE decreases sharply and eventually becomes comparable to that of the baseline model. (2) For pairwise preference accuracy, SMORM consistently outperforms the baseline model, with the improvement being especially pronounced on the evaluation set, as illustrated in Fig. 10.

Meanwhile, to better explain how SMORM enables BT training to enhance the performance of the regression head, we analyze the correlation between attribute-level ground-truth preferences and the pairwise preferences predicted by a strong single-objective reward model. Specifically, we use Skywork/Skywork-Reward-V2-Llama-3.1-8B, a model with strong scoring capability, to score all pairwise comparisons in the nvidia/HelpSteer2 dataset. We then compute, for each attribute, the proportion of pairs whose preference direction matches the model's predicted preference. The results are shown in Table 4.

We observe that helpfulness, correctness, and coherence exhibit substantially higher correlation with the single-objective reward model, indicating that these attributes align more closely with a well-trained Bradley–Terry signal. Consistent with this, Fig. 10 shows that the performance improvement of SMORM over the baselines is also most pronounced on these three attributes, which supports our hypothesis that BT training provides stronger learning signals for attributes that align well with the underlying preference structure.

Table 4: Alignment statistics and correlation percentages across different attributes.

| Attribute | Alignment | Misalignment | Ties | Correlation (%) |
|---|---|---|---|---|
| helpfulness | 5434 | 1787 | 2941 | 75.25 |
| correctness | 5149 | 1646 | 3367 | 75.78 |
| coherence | 3062 | 1143 | 5957 | 72.82 |
| complexity | 1696 | 970 | 7496 | 63.62 |
| verbosity | 3048 | 2033 | 5081 | 59.99 |

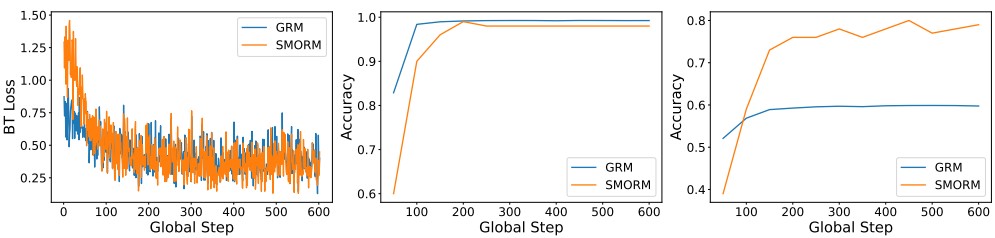

Figure 6: Comparison of SMORM and GRM on training dynamics and generalization.

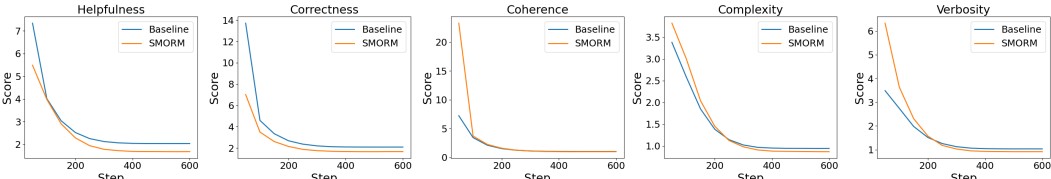

Figure 7: Training curves on MSE.

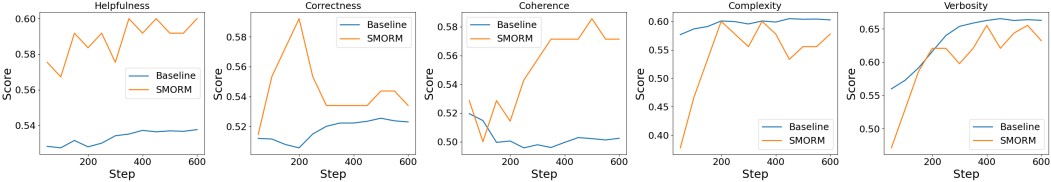

Figure 8: Pairwise preference accuracy on training dataset.

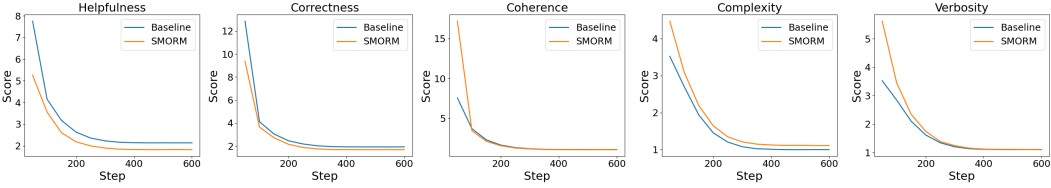

Figure 9: Evaluation curves on MSE.

## C.2  ERROR BAR

To demonstrate the statistical significance of our method compared with the baselines, we evaluate SMORM against GRM (Yang et al., 2024) and ODIN (Chen et al., 2024) through two sets of experiments:

(1) We use google/gemma-2-2b-it as the base model and train SMORM on two datasets: the preference dataset Skywork/Skywork-Reward-Preference-80K-v0.2 and the multi-objective reward dataset nvidia/HelpSteer2. We run each method with three different seeds (42, 123, 456), and report the mean and variance across (i) training and evaluation curves on both the in-domain (Skywork) and out-of-domain (UnifiedFeedback) datasets, as shown in Fig. 11, and (ii) performance on RewardBench, as shown in Table 5.

(2) Following the experimental setup in Sec. 3, we again use gemma-2b-it as the base model and train with three seeds (42, 123, 456), reporting the mean and variance of PPO-based RLHF experiments. The corresponding results are presented in Fig. 12.

Across both experimental settings, SMORM consistently shows higher performance compared to ODIN and GRM, demonstrating statistically significant improvements.

We should also include results on RewardBench

Table 5: Mean ± variance on Rewardbench.

| Method | Chat | Chat Hard | Safety | Reasoning | Overall |
|---|---|---|---|---|---|
| SMORM-F | 89.1 ± 1.96 | 65.6 ± 0.04 | 77.65 ± 3.42 | 71.3 ± 0.36 | 75.9 ± 0.44 |
| GRM | 86.05 ± 0.06 | 66.0 ± 0.16 | 74.2 ± 0.09 | 69.85 ± 1.32 | 74.0 ± 0.36 |
| ODIN | 84.25 ± 2.40 | 63.05 ± 0.12 | 71.75 ± 0.42 | 73.75 ± 17.22 | 73.2 ± 1.44 |

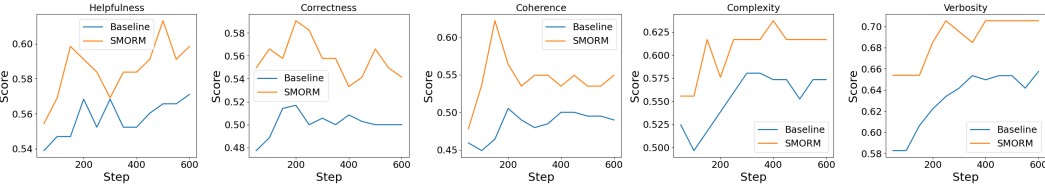

Figure 10: Pairwise preference accuracy on evaluation dataset.

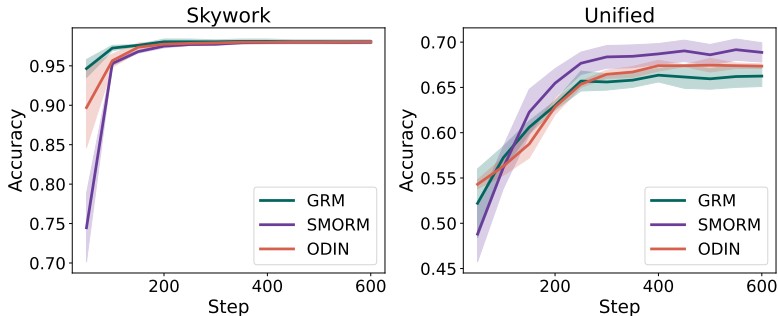

Figure 11: Training curves with error bar.

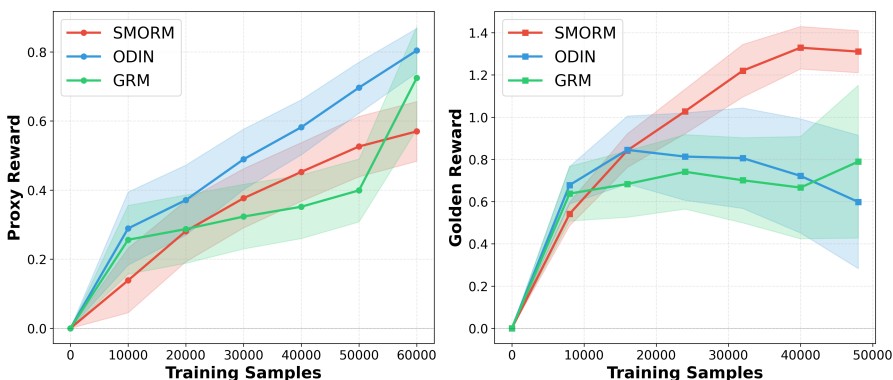

Figure 12: PPO curves with error bar.

## C.3 RESULTS ON DIFFERENT OOD SETTINGS

To further demonstrate the out-of-distribution (OOD) generalization ability of our SMORM, we follow the experimental setup in Hong et al. (2025) and compare SMORM (regression head trained on the multi-objective dataset nvidia/HelpSteer2) with the baseline BT classifier and its variants: BT-Norm (Wei et al., 2022), BT-Hinge, BT-DR (Yuan et al., 2024), and BT-BSR (Hong et al., 2025). The results are presented in Table 6. We observe that while SMORM achieves performance comparable to the strongest baseline, BT-BSR, on the in-domain (ID) setting, it consistently outperforms all baselines across the three OOD evaluation settings, demonstrating its superior generalization capability.

Table 6: Performance of Llama-3 (3B) and Qwen2.5 (3B) under different disjoint evaluation settings (values multiplied by 100).

| Setting | Model | $\mathcal{L}_{BT}$ | $\mathcal{L}_{BT-Hinge}$ | $\mathcal{L}_{BT-Norm}$ | $\mathcal{L}_{BT-DR}$ | $\mathcal{L}_{BT-BSR}$ | SMORM-F |
|---|---|---|---|---|---|---|---|
| **In-Domain** | Llama-3 (3B) | 84.5 | 84.0 | 80.0 | 82.5 | 85.0 | 85.5 |
| | Qwen2.5 (3B) | 84.5 | 84.0 | 82.0 | 84.5 | 85.5 | 87.0 |
| **Prompt-disjoint** | Llama-3 (3B) | 68.0 | 68.5 | 60.0 | 67.5 | 70.5 | 73.5 |
| | Qwen2.5 (3B) | 70.0 | 69.5 | 65.0 | 67.5 | 72.0 | 74.0 |
| **Response-disjoint** | Llama-3 (3B) | 61.5 | 61.5 | 50.0 | 58.5 | 63.5 | 68.0 |
| | Qwen2.5 (3B) | 58.0 | 56.5 | 45.0 | 54.0 | 60.5 | 66.5 |
| **Mutual-disjoint** | Llama-3 (3B) | 51.5 | 52.0 | 47.5 | 50.5 | 54.5 | 57.5 |
| | Qwen2.5 (3B) | 58.0 | 56.0 | 53.0 | 55.0 | 61.0 | 66.0 |

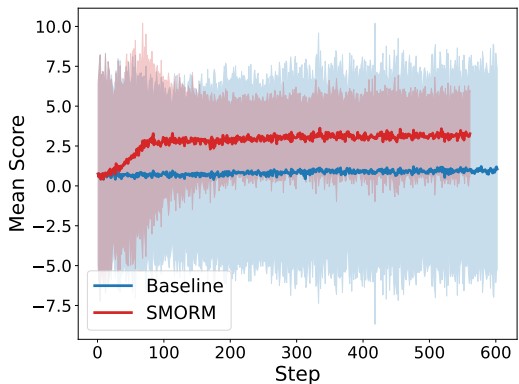

Figure 13: Caption

## C.4 Mean and Variance of Single-Objective Score

In this section, we report the mean and variance of the single-objective (preference) score during training. Specifically, we follow the experimental setup described in Sec. C.2, and plot the preference scores provided by SMORM-F and the baseline BT classifier. The results are shown in Fig. 13.

We observe that the mean score of SMORM-F exhibits a noticeable shift at the beginning of training and stabilizes after approximately 10 steps. This trend aligns with the results in Fig. 7 and Fig. 9, where the MSE of the regression head rapidly decreases during the first 100 training steps. This observation empirically supports our theoretical result in Prop. 1, which establishes a connection between the BT scores and the regression scores trained jointly via SMORM.

## C.5 Preliminary Experiments on Improving both Single- and Multi-Objective Head

**Experimental Setup.** We follow the same training setup for SMORM as described in Section 3. We compare SMORM with the baseline SORM and MORM, using `Llama-3.2-3B-Instruct`[2] as the backbone model for all methods. The comparison results on RewardBench against SORM and MORM are presented in Table 7 and Table 8, respectively. The corresponding datasets used for training each reward model are also listed in the tables. Note that SORM is trained solely on $\mathcal{D}_S$, MORM on $\mathcal{D}_M$, while SMORM is jointly trained on both $\mathcal{D}_S$ and $\mathcal{D}_M$.

Table 7: Results on RewardBench compared to the baseline SORM.

| Dataset $\mathcal{D}_S$ / $\mathcal{D}_M$ | Model | Chat | Chat Hard | Safety | Reasoning | RewardBench |
|---|---|---|---|---|---|---|
| UltraFeedback (binarized) / UltraFeedback | Baseline | 89.1 | 40.7 | 45.2 | 36.7 | 52.9 |
| | SMORM-M | 91.3 | 39.0 | 50.1 | 41.9 | 55.6 |
| | SMORM-F | 88.4 | 43.4 | 44.3 | 49.6 | 56.4 |
| Skywork80K/ HelpSteer2 | Baseline | 73.4 | 60.5 | 79.8 | 49.6 | 65.8 |
| | SMORM-M | 83.5 | 55.5 | 74.5 | 53.6 | 66.8 |
| | SMORM-F | 80.4 | 62.1 | 80.7 | 55.1 | 69.6 |

**Results Analysis.** From Table 7, we observe that SMORM-F achieves a higher average score on RewardBench compared to the baseline model. Notably, regardless of whether the single-objective reward model is trained on the same dataset or a different one, SMORM-F consistently outperforms the baseline. This result highlights the flexibility of our SMORM framework. From Table 8, we further observe that by simply aggregating the multi-attribute scores to create a single-objective preference dataset and training a SMORM, SMORM-L achieves average scores **5.5** and **6.3** points higher than training a MORM alone. This highlights how easily SMORM can enhance the performance of the multi-objective reward function.

---

[2] meta-llama/Llama-3.2-3B-Instruct

Table 8: Results on RewardBench and RM-Bench compared to the baseline MORM.

| Dataset $\mathcal{D}_S$ / $\mathcal{D}_M$ | Model | Chat | Chat Hard | Safety | Reasoning | RewardBench | RM-Bench |
|---|---|---|---|---|---|---|---|
| HelpSteer2 (binarized)/ HelpSteer2 | Baseline | 55.8 | 50.4 | 44.8 | 54.2 | 51.3 | 49.2 |
| | SMORM-M | 48.9 | 48.5 | 51.4 | 75.7 | 56.1 | 50.1 |
| | SMORM-F | 50.3 | 48.7 | 54.7 | 73.6 | 56.9 | 50.9 |
| | SMORM-L | 50.7 | 49.3 | 52.9 | 73.3 | 56.8 | 53.0 |
| UltraFeedback (binarized)/ UltraFeedback | Baseline | 70.3 | 46.1 | 41.6 | 42.0 | 50.0 | 50.2 |
| | SMORM-M | 91.3 | 39.0 | 50.1 | 41.9 | 55.6 | 51.0 |
| | SMORM-F | 88.4 | 43.4 | 44.3 | 49.6 | 56.4 | 49.8 |
| | SMORM-L | 90.2 | 40.1 | 54.2 | 40.8 | 56.3 | 54.4 |

## C.6 Additional Results on Improving Single-Objective Head

In this section, we provide additional experimental results comparing SMORM-F to baseline single-objective reward functions. We follow the same experimental setup as in Section 5. Results using Mistral-7B-Instruct as the base model are reported in Table 9.

Table 9: Comparison of SMORM-F and baselines on RewardBench. Baseline results from (Yang et al., 2024).

| Reward model | $\mathcal{D}_S/\mathcal{D}_M$: UnifiedFeedback 400k/UltraFeedback | | | | | $\mathcal{D}_S/\mathcal{D}_M$: UnifiedFeedback 40k/HelpSteer2 | | | | |
|---|---|---|---|---|---|---|---|---|---|---|
| | Chat | Chat-Hard | Safety | Reasoning | Avg | Chat | Chat-Hard | Safety | Reasoning | Avg |
| Base Model: Mistral 7b Instruct | | | | | | | | | | |
| Baseline (Single) | 96.6 | 52.4 | 86.7 | 69.5 | 76.3 | 94.9 | 51.7 | 64.9 | 62.4 | 68.5 |
| Baseline + margin | 96.4 | 51.5 | 85.3 | 64.8 | 74.5 | 89.7 | 47.1 | 70.7 | 43.6 | 62.8 |
| Label smooth | 97.2 | 49.8 | 85.8 | 72.3 | 76.3 | 94.1 | 47.1 | 67.5 | 79.7 | 72.1 |
| Ensemble | 96.6 | 51.8 | 85.1 | 73.0 | 76.6 | 89.6 | 50.2 | 72.7 | 59.0 | 69.3 |
| GRM (linear) w/ dpo | 98.0 | 53.3 | 86.4 | 75.3 | 78.3 | 95.1 | 47.5 | 82.2 | 74.7 | 74.9 |
| GRM (linear) w/ sft | 97.8 | 54.6 | 86.3 | 79.2 | 79.5 | 93.4 | 51.9 | 80.7 | 78.8 | 76.2 |
| GRM w/ dpo | 97.8 | 54.0 | 85.7 | 74.4 | 78.0 | 97.8 | 52.4 | 78.0 | 77.3 | 76.4 |
| GRM w/ sft | 98.0 | 55.3 | 85.8 | 71.2 | 77.6 | 94.1 | 48.5 | 83.4 | 77.4 | 75.9 |
| SMORM-F | 97.8 | 55.3 | 85.9 | 80.1 | **79.8** | 95.8 | 60.1 | 80.5 | 74.7 | **77.8** |

# D Theory

## D.1 Definitions and Assumptions

Let $K$ be the number of attributes considered. We define $f_\theta$ as the backbone of our reward model, which maps an input question–response pair to a hidden representation of dimension $d$. For each attribute head $k$, the reward score is given by $r_k = \mathbf{w}_k^\top f_\theta$. In particular, $r_s$ (or $r_0$) denotes the score from the single-objective head used to model overall preference (e.g., chosen vs. rejected), while $r_k$ for $k > 0$ corresponds to the outputs of fine-grained attribute-specific heads.

**Positive-definite covariances.** Let

$$f_c = f_\theta(x_s, y_c), \quad f_r = f_\theta(x_s, y_r), \quad f_m = f_\theta(x_m, y_m).$$

We define the covariance matrices

$$\Sigma_S := \mathbb{E}_{\mathcal{D}_S}\left[(f_c - f_r)(f_c - f_r)^\top\right] \quad \text{and} \quad \Sigma_M := \mathbb{E}_{\mathcal{D}_M}\left[f_m f_m^\top\right],$$

and assume both are positive definite (PD).

Modern feature extractors (e.g., transformer-based backbones) typically embed inputs into a high-dimensional space of size $d$, which exceeds the intrinsic rank of the data. Provided that the samples in $\mathcal{D}_S$ (or $\mathcal{D}_M$) do not all lie in a strict lower-dimensional hyperplane, the corresponding empirical covariance matrices will be full-rank and therefore positive definite.

**Correlation between heads.** We naturally assume a positive correlation between the aggregated fine-grained attribute scores and the preference along the chosen/rejected dimension. Specifically, let

$$\mu_S := \mathbb{E}_{(x_s, y_c, y_r) \sim \mathcal{D}_S}\left[f_\theta(x_s, y_c) - f_\theta(x_s, y_r)\right],$$

and define
$$C_M := \mathbb{E}_{(x_m, y_m, \mathbf{r}) \sim \mathcal{D}_M} \left[ f_\theta(x_m, y_m) \, \mathbf{r}^\top \right] \in \mathbb{R}^{d \times K}.$$
Note that we denote $r$ as the reward function, and $\mathbf{r}$ as the vector of multi-attribute scores. Then, we define
$$\alpha := \mu_S^\top \Sigma_M^{-1} C_M,$$
and assume that the sum of its components is non-negative, i.e.,
$$\mathbf{1}^\top \alpha \geq 0.$$

Assuming $\mathbb{E}[\mathbf{r}] = \mathbb{E}[\mathbf{w}_S^\top] = 0$, we define the *raw coupling vector*
$$\alpha = \mu_S^\top \Sigma_M^{-1} C_M = \left[ \text{Cov}(\mathbf{r}_1, \mathbf{w}_S^\top f), \ldots, \text{Cov}(\mathbf{r}_K, \mathbf{w}_S^\top f) \right] \in \mathbb{R}^K,$$
and normalize
$$\beta = \frac{\alpha}{\|\tilde{\mu}_S\|^2} = \frac{\mu_S^\top \Sigma_M^{-1} C_M}{\mu_S^\top \Sigma_S^{-1} \mu_S} \in \mathbb{R}^K.$$
Then, $\beta_i > 0$ if the score on attribute $i$ tends to increase as the single-objective score $\mathbf{w}_S^\top f$ increases.

The sum $\mathbf{1}^\top \alpha$ represents the total (signed) covariance between the single-objective preference score and the aggregate multi-attribute score. In most real-world annotation settings, chosen–rejected labels are used as proxies for overall answer quality. Since higher-quality responses tend to improve multiple fine-grained attributes (e.g., helpfulness, correctness, coherence, etc.), the covariances across these attributes typically sum to a positive total.

**Hidden Ground Truth.** For each head $r_k$, we denote $r_k^*$ as the hidden ground-truth reward function, and we model the labeled score for head $k$ as $g_k = r_k^* + \epsilon_k$ for $\varepsilon \sim \mathcal{N}(0, \Sigma), \Sigma_{0k} > 0$. Here the diagonal entries of $\Sigma$ are $\sigma_{kk} = \text{Var}(\varepsilon_k)$, the noise variance for head $k$.

## D.2 PROOF OF THEOREM 1

**Theorem 1** (Implicit Multi-Attribute Effect). *Let a reward model be trained under the SMORM framework, and suppose the following conditions hold: (1) Bounded features: There exists $B < \infty$ such that $\|f_\theta(x, y)\| \leq B$ for every $(x, y)$. (2) Positive-definite covariances: let $f_c = f_\theta(x_s, y_c)$, $f_r = f_\theta(x_s, y_r)$, $f_m = f_\theta(x_m, y_m)$. $\Sigma_S := \mathbb{E}_{\mathcal{D}_S}[(f_c - f_r)(f_c - f_r)^\top]$ and $\Sigma_M := \mathbb{E}_{\mathcal{D}_M}[f_m f_m^\top]$ are positive-definite matrices. (3) Positive correlation: Let $\mu_S := \mathbb{E}_{(x_s, y_c, y_r) \sim \mathcal{D}_S}[f_\theta(x_s, y_c) - f_\theta(x_s, y_r)]$ and let $C_M := \mathbb{E}_{(x_m, y_m, r) \sim \mathcal{D}_M}[f_\theta(x_m, y_m) \, \mathbf{r}^\top] \in \mathbb{R}^{d \times K}$. Then $\alpha := \mu_S^\top \Sigma_M^{-1} C_M$ has non-negative sum, i.e. $\mathbf{1}^\top \alpha \geq 0$. As the optimization of both reward heads converge to their population minimizers, there exist constants $c = \frac{\mathbf{1}^\top \alpha}{K \left( \mu_S^\top \Sigma_S^{-1} \mu_S \right)}$ and $\varepsilon \geq 0$—depending only on $B$ and second-order moments—such that for every pair $(x, y)$:*
$$r_m(x, y) = \tfrac{1}{K} \sum_{i=1}^K w_{M,i}^\top f_\theta(x, y) \geq c \left( w_S^\top f_\theta(x, y) \right) - \varepsilon = c r_s(x, y) - \varepsilon. \tag{6}$$

*Proof.* Replacing the logistic (BT) and regression losses by squared losses does not alter *directions* of the minimisers because any strictly convex proper surrogate has the same first-order optimality conditions up to a positive scalar factor Bartlett et al. (2006). We therefore analyse the following least-squares problems:
$$\min_{\mathbf{w}_S} \mathbb{E}_S \left[ (\mathbf{w}_S^\top (f_c - f_r) - 1)^2 \right], \qquad \min_{\mathbf{w}_M} \mathbb{E}_M \left[ \|\mathbf{w}_M^\top f_m - \mathbf{r}\|_2^2 \right],$$

For a generic least-squares objective $\min_{\mathbf{w}} \mathbb{E}[(\mathbf{w}^\top u - t)^2]$, setting the gradient to zero yields $\mathbb{E}[u\, u^\top] \mathbf{w} = \mathbb{E}[u\, t]$. Applying this template we obtain the *population* solutions:
$$\mathbf{w}_S = \Sigma_S^{-1} \mu_S, \qquad\qquad \mathbf{w}_M = \Sigma_M^{-1} C_M. \tag{7}$$

Define the whitening operator $\Phi := \Sigma_S^{-1/2} f \in \mathbb{R}^d$, $\tilde{\mu}_S := \Sigma_S^{-1/2} \mu_S$, and $\tilde{C}_M := \Sigma_S^{1/2} \Sigma_M^{-1} C_M$. Then $\Phi_c = \Sigma_S^{-1/2} f_c, \Phi_r = \Sigma_S^{-1/2} f_r$ and $\mathbb{E}_S[(\Phi_c - \Phi_r)(\Phi_c - \Phi_r)^\top] = I_d$ and $\tilde{\mu}_S \neq 0$. Then Equation 7 becomes:
$$\mathbf{w}_S = \Sigma_S^{-1/2} \tilde{\mu}_S, \qquad\qquad \mathbf{w}_M := \Sigma_S^{-1/2} \tilde{C}_M. \tag{8}$$

Because $\tilde{\mu}_S \neq 0$, we write the Euclidean projection of each column of $\tilde{C}_M$ onto $\tilde{\mu}_S$:

$$\tilde{C}_M = \underbrace{\tilde{\mu}_S \frac{\tilde{\mu}_S^\top \tilde{C}_M}{\|\tilde{\mu}_S\|^2}}_{\text{aligned component}} + E, \quad E^\top \tilde{\mu}_S = 0. \tag{9}$$

Define the *coupling vector* $\beta := \frac{\tilde{\mu}_S^\top \tilde{C}_M}{\|\tilde{\mu}_S\|^2} = \frac{\alpha}{\|\tilde{\mu}_S\|^2} \in \mathbb{R}^K$. By assumption (3) we have $\mathbf{1}^\top \alpha \geq 0$ and thus $\mathbf{1}^\top \beta > 0$.

Then for feature vector $f$, we have:

$$\mathbf{w}_M^\top f \overset{Equation\ 8}{=} \tilde{C}_M^\top \underbrace{\Sigma_S^{-1/2} f}_{\Phi}$$

$$\overset{Equation\ 9}{=} \beta\,(\tilde{\mu}_S^\top \Phi) + E^\top \Phi$$

$$= \beta\,(\mathbf{w}_S^\top f) + E^\top \Phi.$$

Decomposing $\Phi$ by:

$$\Phi = \underbrace{\frac{\tilde{\mu}_S^\top \Phi}{\|\tilde{\mu}_S\|^2} \tilde{\mu}_S}_{\|\tilde{\mu}_S} + \underbrace{z}_{\perp \tilde{\mu}_S}, \quad \tilde{\mu}_S^\top z = 0,$$

so that $E^\top \Phi = E^\top z$. By bounded-features, $\|f\| \leq B$ and hence $\|\Phi\| \leq B/\sqrt{\lambda_{\min}(\Sigma_S)}$, where $\sqrt{\lambda_{\min}(\Sigma_S)}$ is the square root of the smallest eigenvalue of the . The projection term has norm $|\tilde{\mu}_S^\top \Phi|/\|\tilde{\mu}_S\| \leq |\mathbf{w}_S^\top f|$, so the orthogonal part satisfies $\|z\| \leq B/\sqrt{\lambda_{\min}(\Sigma_S)}$. Therefore

$$|E^\top z| \leq \|E\|_{\mathrm{op}}\,\|z\| \leq \underbrace{\frac{B}{\sqrt{\lambda_{\min}(\Sigma_S)}}\,\|E\|_{\mathrm{op}}}_{\varepsilon}.$$

Averaging over the $K$ attributes and using $c := \max\{0, \mathbf{1}^\top \beta\}/K$ gives

$$\frac{1}{K}\sum_{i=1}^K \mathbf{w}_{M,i}^\top f \geq c\,\mathbf{w}_S^\top f - \frac{\varepsilon}{K}. \tag{10}$$

where $c = \frac{\mathbf{1}^\top \beta}{K} = \frac{\mathbf{1}^\top \alpha}{K\|\tilde{\mu}_S\|^2}$.

**Monotone lower bound.** Because $\mathbf{1}^\top \beta > 0$ by assumption (3), we have $c > 0$. Define the linear function $L(s) := c\,s - \varepsilon/K$. Inequality equation 10 reads

$$r_M(x,y) = \frac{1}{K}\sum_{i=1}^K \mathbf{w}_{M,i}^\top f_\theta(x,y) \geq L\big(r_S(x,y)\big),$$

so $r_M$ is bounded from below by the *increasing* map $L(\,\cdot\,)$ of the single-objective score $r_S$. Consequently, for any two responses $(x_1, y_1)$ and $(x_2, y_2)$,

$$r_S(x_1, y_1) \geq r_S(x_2, y_2) \implies L\big(r_S(x_1, y_1)\big) \geq L\big(r_S(x_2, y_2)\big),$$

and hence the lower bound on the multi-attribute average is larger (or equal) whenever the single-objective score is larger. In particular, for any threshold $\tau$,

$$r_S(x,y) \geq \tau \implies r_M(x,y) \geq c\,\tau - \frac{\varepsilon}{K}.$$

This completes the proof of Theorem 1.

$\square$

**Pure-SORM failure.** If no multi-attribute head is trained ($C_M = 0$), then $\alpha = 0 \Rightarrow c = 0$, so the lower bound degenerates to $r_M \geq -\varepsilon$, offering no positive coupling between single-objective and multi-attribute scores.

### D.3 PROOF OF LEMMA 1.

**Lemma 1.** *Let $y_A, y_B$ be a pair of responses. Assume $g_s(y)$ is the ground truth score and $r_s(y)$ is the predicted score under a Bradley–Terry model. Then:*

$$\mathbb{P}(y_A \succ y_B) = \sigma\big(r_s(y_A) - r_s(y_B)\big), \quad \mathbb{P}^{\star}(y_A \succ y_B) = \sigma\big(g_s(y_A) - g_s(y_B)\big),$$

*where $\sigma(t) = \frac{1}{1+e^{-t}}$. The expected preference error satisfies:*

$$\mathbb{E}_{\mathcal{D}_S} |\mathbb{P}(y_A \succ y_B) - \mathbb{P}^{\star}(y_A \succ y_B)| \leq \frac{1}{4} \mathbb{E}_{\mathcal{D}_S}\left(\sqrt{2\,MSE(r_s)}\right),$$

*with $MSE(r_s) = \big(r_s(y) - g_s(y)\big)^2$. Similarly, for a multi-objective reward model with predicted score $r_m$ and ground truth $g_m$, let: $e_m = r_m(y_A) - r_m(y_B), \quad e_m^{\star} = g_m(y_A) - g_m(y_B)$, then the error is bounded as:*

$$\mathbb{E}_{\mathcal{D}_M} |e_m - e_m^{\star}| \leq \mathbb{E}_{\mathcal{D}_M}\left(\sqrt{2\,MSE(r_m)}\right).$$

*Proof.* For a pair of responses $y_A$ and $y_B$, the Bradley–Terry model defines the probability that $y_A$ is preferred (i.e., has a higher overall reward) as:

$$\mathbb{P}(y_A \succ y_B) = \sigma\big(r_s(y_A) - r_s(y_B)\big),$$

where $\sigma(t) = \frac{1}{1+e^{-t}}$ is the sigmoid function, and $r_s(v)$ is the model's predicted overall score. The corresponding ground-truth preference probability, based on labeled scores, is given by:

$$\mathbb{P}^{\star}(y_A \succ y_B) = \sigma\big(g_s(y_A) - g_s(y_B)\big),$$

where $g_s(\cdot)$ denotes the ground-truth reward.

The prediction error in probability space is the absolute difference:

$$\Delta_{AB} = \big|\sigma\big(r_s(y_A) - r_s(y_B)\big) - \sigma\big(g_s(y_A) - g_s(y_B)\big)\big|.$$

Since the sigmoid derivative satisfies

$$\sigma'(t) = \sigma(t)(1 - \sigma(t)),$$

and reaches its maximum value of $\frac{1}{4}$ at $t = 0$, we have $\sigma'(t) \leq \frac{1}{4}$ for all $t$. This implies that the sigmoid function is $\frac{1}{4}$-Lipschitz:

$$|\sigma(a) - \sigma(b)| \leq \frac{1}{4}|a - b|, \quad \forall\, a, b \in \mathbb{R}.$$

Applying this to our setup, we obtain:

$$|\mathbb{P}(y_A \succ y_B) - \mathbb{P}^{\star}(y_A \succ y_B)| \leq \frac{1}{4}\,|(r_s(y_A) - r_s(y_B)) - (g_s(y_A) - g_s(y_B))|.$$

Taking expectation and applying the Cauchy–Schwarz inequality, we can further bound the expected pairwise error by:

$$\text{Pairwise-error} \leq \frac{1}{4}\sqrt{2\,\text{MSE}(r_s)},$$

where $\text{MSE}(r_s) := \mathbb{E}_y\left[(r_s(y) - g_s(y))^2\right]$ is the mean squared error of the predicted scores.

This result shows that minimizing the pointwise MSE of the reward model also reduces the upper bound on the pairwise misordering error, thereby improving preference consistency.

Similarly, for a multi-objective reward model with predicted score $r_m$ and ground truth $g_m$, let: $e_m = r_m(v_A) - r_m(v_B), \quad e_m^{\star} = g_m(v_A) - g_m(v_B)$, then the error is bounded as:

$$\mathbb{E}_{\mathcal{D}_M} |e_m - e_m^{\star}| \leq \mathbb{E}_{\mathcal{D}_M}\left(\sqrt{2\,\text{MSE}(r_m)}\right).$$

$\square$

To relate this to the Bradley–Terry loss, recall that the Bradley–Terry loss for a pair $(y_A, y_B)$, where $y_A$ is the preferred response, is given by:

$$\ell_{\text{BT}} = -\log \sigma\big(r_s(y_A) - r_s(y_B)\big).$$

If the model's predicted scores $r_s(v)$ are close to the ground-truth scores $g_s(v)$—i.e., the mean squared error (MSE) is small—then the difference $r_s(y_A) - r_s(y_B)$ will closely approximate $g_s(y_A) - g_s(y_B)$. By the Lipschitz continuity of the sigmoid function, this implies that the predicted probability under the model and the ideal ground-truth probability will also be close. Consequently, the pairwise preference error will be small.

This reasoning provides a theoretical justification that minimizing the MSE of individual predictions naturally leads to accurate pairwise probability estimates, as evaluated by the Bradley–Terry loss.

Furthermore, while the single-objective head $r_s$ is trained using the Bradley–Terry likelihood, our previous bound shows that the expected Bradley–Terry test risk,

$$\mathcal{R}_{\text{BT}} = \mathbb{E}_{(y_A, y_B)} \left[ -\log \sigma\big(r_s(y_A) - r_s(y_B)\big) \right],$$

is a 1-Lipschitz function of the score difference $r_s(y_A) - r_s(y_B)$. Therefore, controlling the variance of the individual scores—captured by $\text{MSE}_0 = \mathbb{E}[(r_s(y) - g_s(y))^2]$—directly bounds the generalization error under the Bradley–Terry loss, up to a constant factor.

### D.4 Proof of Theorem 2

**Theorem 2.** *Under the same assumptions as in Theorem 1 and assuming that the feature extractor $f_\theta$ is differentiable, let $\widehat{\theta}$ denote the maximum likelihood estimator (MLE) of the ground truth optimal parameter $\theta^\star$. Let $\widehat{\theta}_s$ and $\widehat{\theta}_m$ denote the maximum likelihood estimators of the single- and multi-objective reward functions, respectively. Define $M_S(y) = \mathbf{w}_S^\top f_{\theta^\star}(y)$, $M_M(y) = \mathbf{w}_M^\top f_{\theta^\star}(y)$. Then, for a response $y$, the mean squared error (MSE) of the predicted reward can be approximated as:*

$$\text{MSE}_S \approx \nabla_\theta M_S(y)^\top \text{Cov}\left(\widehat{\theta}_s\right) \nabla_\theta M_S(y) + \sigma_{00}, \text{MSE}_M \approx \nabla_\theta M_M(y)^\top \text{Cov}\left(\widehat{\theta}_m\right) \nabla_\theta M_M(y) + \sigma_{00},$$

*where $\sigma_{00}$ is the intrinsic randomness in the label. Moreover, SMORM yields lower asymptotic MSE for both the single- and multi-objective heads compared to training either head alone:*

$$\text{MSE}_S^{SMORM} < \text{MSE}_S^{single}, \quad \text{MSE}_M^{SMORM} < \text{MSE}_M^{multi} \tag{11}$$

*Proof.* **Fisher matrix.** The Fisher information is a way of measuring the amount of information that an observable random variable carries about an unknown parameter. Mathematically, for a parameter vector $\theta$ and data $D$ with likelihood $p(D \mid \theta)$, the Fisher information is

$$\mathcal{I}(\theta) \stackrel{\text{def}}{=} \mathbb{E}_{D \sim p(\cdot \mid \theta)} \left[ \nabla_\theta \log p(D \mid \theta) \nabla_\theta \log p(D \mid \theta)^\top \right]. \tag{12}$$

Intuitively, it measures how sensitive the log-likelihood is to small changes in $\theta$. More curvature means larger $\mathcal{I}(\theta)$ and thus implies that we can estimate $\theta$ more precisely. The celebrated Cramér–Rao bound says that (under mild conditions) any unbiased estimator's covariance is at least $[\mathcal{I}(\theta)]^{-1}$ (Kay, 1993).

In our square-loss, Gaussian-noise setting, the empirical Fisher matrix becomes the empirical sum of outer products of gradients:

$$\mathcal{I}^{(\text{regime})}(\theta) = \frac{1}{n} \sum_{i=1}^{n} \sum_{k \in \mathcal{K}_{\text{train}}} \frac{1}{\sigma_{kk}} \left[ \nabla_\theta r_k(y_i) \right] \left[ \nabla_\theta r_k(y_i) \right]^\top.$$

For single-head reward model that evaluates the overall quality, we have $\mathcal{K} = \{0\}$. For $K$-attribute reward model that evaluates the response according to $K$ specific aspects, $\mathcal{K} = \{1, \ldots, K\}$. For our hybrid model, $\mathcal{K} = \{0, \ldots, K\}$. Because every summand is positive semi-definite, adding a task can only increase or keep the Fisher matrix. Hence we have:

$$\mathcal{I}^{(\text{hybrid})} = \mathcal{I}^{(\text{single})} + \Delta, \qquad \Delta \succeq 0, \tag{13}$$

**Strict positivity of difference term.** In fact, give the assumed positive correlation between head 0 and other attribute heads, we can show that the overall Fisher matrix can be strictly larger. Denote

$$g_0(y_i) = \nabla_\theta r_s(y_i),$$

which is the gradient of the overall head's prediction with respect to $\theta$. When we look at the contribution of the other heads, what matters is how their gradients project onto $g_0(y_i)$:

$$g_0(y_i)^\top \nabla_\theta r_k(y_i).$$

The positive correlation assumption $\rho_{0k} > 0$ implies that, on average, the gradients $\nabla_\theta r_k(y_i)$ tend to point in a similar direction to $g_0(y_i)$. This means that $g_0(y_i)^\top \nabla_\theta r_k(y_i) > 0$. For example, if we project the Fisher information onto the direction $g_0$, using the linearity of the inner product, we obtain:

$$g_0(y_i)^\top \mathcal{I}^{\text{hybrid}}(\theta)\, g_0(y_i) = g_0(y_i)^\top \mathcal{I}^{\text{single}}(\theta)\, g_0(y_i) + \sum_{k=1}^{K} \frac{1}{n\sigma_{kk}} \sum_{i=1}^{n} \big(g_0(y_i)^\top \nabla_\theta r_k(y_i)\big)^2.$$

Because each term $\big(g_0(y_i)^\top \nabla_\theta r_k(y_i)\big)^2$ is strictly positive when the inner product is nonzero, and positive correlation ensures that it is indeed positive on average, we know the extra sum is strictly positive. That is,

$$g_0(y_i)^\top \mathcal{I}^{\text{hybrid}}(\theta)\, g_0(y_i) > g_0(y_i)^\top \mathcal{I}^{\text{single}}(\theta)\, g_0(y_i).$$

Therefore, we can get

$$\mathcal{I}^{(\text{hybrid})} = \mathcal{I}^{(\text{single})} + \Delta, \qquad \Delta \succ 0, \tag{14}$$

**Asymptotic Variance of $\widehat{\theta}$.** If $\widehat{\theta}$ is the maximum-likelihood estimator (MLE) of $\theta^\star$ and the usual regularity conditions hold (i.i.d. samples, smooth log-likelihood, finite Fisher information, etc.), then the asymptotic normality theorem for MLEs states

$$\sqrt{n}\big(\widehat{\theta} - \theta^\star\big) \xrightarrow{d} \mathcal{N}\Big(0, [\mathcal{I}(\theta^\star)]^{-1}\Big).$$

where "$\xrightarrow{d}$" denotes convergence in distribution and the covariance of the limiting Gaussian is the inverse Fisher information, which is the smallest possible asymptotic variance for any unbiased estimator by Cramér–Rao.

Thus, if one yields a larger Fisher matrix (more information), its estimator's asymptotic covariance matrix is smaller, so predictions based on it are less variable. Hence

$$\text{Cov hybrid}(\widehat{\theta}) \prec \text{Cov single}(\widehat{\theta}). \tag{15}$$

**From $\theta$-variance to MSE by Bias–variance decomposition.** For a fresh test example $v$ we predict with

$$\hat{s}_0(v) = \mathbf{w}_S^\top M_{\widehat{\theta}}(v), \qquad g_s(v) = \mathbf{w}_S^\top f_{\theta^\star}(v) + \varepsilon_0.$$

The mean-squared error of that prediction is

$$\text{MSE}_0 = \underbrace{\big(\mathbb{E}[\hat{s}_0] - \mathbb{E}[g_s]\big)^2}_{\text{Bias}^2} + \underbrace{\text{Var}[\hat{s}_0]}_{\text{estimation variance}} + \underbrace{\text{Var}[\varepsilon_0]}_{\sigma_{00}\text{(irreducible noise)}}.$$

- **Bias term.** With sufficient optimization and model capacity the MLE is (asymptotically) unbiased, so this term is approximately 0.

- **Variance term.** Fluctuations of $\widehat{\theta}$ across data sets propagate through the network, and first-order Taylor expansion gives

$$M_{\widehat{\theta}}(v) \approx f_{\theta^\star}(v) + \nabla_\theta f_{\theta^\star}(v)\,(\widehat{\theta} - \theta^\star)$$
$$\Longrightarrow \hat{s}_0(v) \approx \mathbf{w}_S^\top f_{\theta^\star}(v) + \nabla_\theta M_S(v)(\widehat{\theta} - \theta^\star) \quad (\text{denote } M_S(v) \stackrel{\text{def}}{=} \mathbf{w}_S^\top f_{\theta^\star}(v))$$
$$\Longrightarrow \text{Var}[\hat{s}_0] \approx \nabla_\theta M_S(v)^\top \text{Cov}(\widehat{\theta}) \nabla_\theta M_S(v).$$

Because the hybrid regime has the smaller $\text{Cov}(\widehat{\theta})$ by Equation 15 above, this variance shrinks.

- **Noise term** $\sigma_{00}$. This is the intrinsic randomness in the label and is identical for all training regimes.

Therefore, when hybrid training reduces the covariance $\mathrm{Cov}(\widehat{\theta})$, the key variance term in the generalization bound decreases. Due to the assumed positive correlation between the fine-grained attributes and overall quality, this reduction leads to a lower test-set MSE for the single-objective head. This establishes inequality 11 and thus completes the proof of Theorem 2.

Moreover, by linking the single-head MSE to the pairwise preference error (as shown in Lemma 1), we demonstrate that the single-objective head trained using our SMORM framework is expected to outperform a conventional single-head reward model. It is worth noting that the same argument naturally extends to the multi-objective reward setting as well.

$\square$

**Theorem 3** (Implicit Multi-Attribute Effect under Finite Data). *Let a reward model be trained under the SMORM framework. Assume:*

1. ***Bounded features.*** *There exists $B < \infty$ such that $\|f_\theta(x,y)\| \leq B$ for all $(x,y)$.*

2. ***Positive-definite covariances.*** *Let $f_c = f_\theta(x_s, y_c)$, $f_r = f_\theta(x_s, y_r)$, $f_m = f_\theta(x_m, y_m)$. Define $\Sigma_S := \mathbb{E}_{\mathcal{D}_S}\big[(f_c - f_r)(f_c - f_r)^\top\big]$ and $\Sigma_M := \mathbb{E}_{\mathcal{D}_M}\big[f_m f_m^\top\big]$, and suppose both are positive definite.*

3. ***Positive correlation.*** *Let $\mu_S := \mathbb{E}_{(x_s, y_c, y_r) \sim \mathcal{D}_S}\big[f_\theta(x_s, y_c) - f_\theta(x_s, y_r)\big]$ and $C_M := \mathbb{E}_{(x_m, y_m, \mathbf{r}) \sim \mathcal{D}_M}\big[f_\theta(x_m, y_m)\,\mathbf{r}^\top\big] \in \mathbb{R}^{d \times K}$. Set $\alpha := \mu_S^\top \Sigma_M^{-1} C_M$ and assume $\mathbf{1}^\top \alpha \geq 0$.*

*Let $n_{\mathrm{pref}}$ and $n_{\mathrm{attr}}$ be the sample sizes used to train the BT head on $\mathcal{D}_S$ and the regression head on $\mathcal{D}_M$, respectively. Let $\widehat{w}_S$ and $\widehat{W}_M$ be the (empirical) minimizers obtained from these finite samples, and define*

$$r_s(x,y) := w_S^\top f_\theta(x,y), \qquad \widehat{r}_s(x,y) := \widehat{w}_S^\top f_\theta(x,y),$$

$$r_m(x,y) := \tfrac{1}{K}\sum_{i=1}^K w_{M,i}^\top f_\theta(x,y), \qquad \widehat{r}_m(x,y) := \tfrac{1}{K}\sum_{i=1}^K \widehat{w}_{M,i}^\top f_\theta(x,y).$$

*As the optimization errors vanish (each head reaches its* empirical *minimizer), there exists*

$$c = \frac{\mathbf{1}^\top \alpha}{K\left(\mu_S^\top \Sigma_S^{-1} \mu_S\right)} \geq 0 \quad and \quad \varepsilon \geq 0$$

*(depending only on $B$ and second-order moments) such that, for every pair $(x,y)$,*

$$\widehat{r}_m(x,y) \geq c\,\widehat{r}_s(x,y) - \varepsilon - \eta_S - \eta_M, \tag{16}$$

*where the finite-sample estimation errors satisfy*

$$\eta_S = O_p\Big(\tfrac{1}{n_{\mathrm{pref}}}\Big), \qquad \eta_M = O_p\Big(\tfrac{1}{n_{\mathrm{attr}}}\Big).$$

*Equivalently, replacing empirical heads by their population counterparts yields $r_m(x,y) \geq c\,r_s(x,y) - \varepsilon$ and equation 16 quantifies the conservative degradation under finite data.*

**Remark (what the $\eta$'s absorb).** The terms $\eta_S$ and $\eta_M$ capture the statistical errors from estimating $\mu_S, \Sigma_S, \Sigma_M, C_M$ and the induced parameter errors in $\widehat{w}_S, \widehat{W}_M$. Concretely, one may bound them in terms of deviations like $\|\widehat{\mu}_S - \mu_S\|$, $\|\widehat{\Sigma}_S - \Sigma_S\|$, $\|\widehat{\Sigma}_M - \Sigma_M\|$, and $\|\widehat{C}_M - C_M\|$, which are $O_p(1/n_{\mathrm{pref}})$ and $O_p(1/n_{\mathrm{attr}})$, respectively, under the boundedness assumption and standard concentration.

# E   ADDITIONAL CLARIFICATIONS ON THEORETICAL ASSUMPTIONS

## E.1   RANK-DEFICIENT COVARIANCES AND THE VALIDITY OF THEOREM 1

In high-dimensional embedding spaces, the covariance matrices

$$\Sigma_S = \mathbb{E}[(f_c - f_r)(f_c - f_r)^\top], \qquad \Sigma_M = \mathbb{E}[f_m f_m^\top]$$

may be singular. In this section we show that Theorem 1 continues to hold under rank deficiency. Importantly, singularity affects only the *magnitude* of the coupling constants $c$ and $\varepsilon$; the implicit multi-attribute effect itself remains valid.

**Least-Squares Solutions under Rank Deficiency**   In the proof of Theorem 1, the population minimizers of the BT and regression objectives satisfy

$$\Sigma_S w_S = \mu_S, \qquad \Sigma_M w_M = C_M,$$

where $\mu_S = \mathbb{E}[f_c - f_r]$ and $C_M = \mathbb{E}[f_m r^\top]$. When $\Sigma_S$ or $\Sigma_M$ is singular, these linear systems may have infinitely many solutions. The correct interpretation is the minimum-norm least-squares solution, given by the Moore–Penrose pseudoinverse:

$$w_S = \Sigma_S^\dagger \mu_S, \qquad w_M = \Sigma_M^\dagger C_M.$$

The pseudoinverse coincides with the actual inverse in the positive-definite case, and therefore generalizes the original proof without altering its algebraic structure.

**Whitening with a Singular Covariance**   Define the whitened feature representation

$$\Phi = \Sigma_S^{\dagger 1/2} f, \qquad \tilde{\mu}_S = \Sigma_S^{\dagger 1/2} \mu_S, \qquad \tilde{C}_M = \Sigma_S^{\dagger 1/2} \Sigma_M^\dagger C_M.$$

The matrix $\Sigma_S^{\dagger 1/2}$ acts as an isometry on the row space of $\Sigma_S$ and maps all vectors in the null space to zero. Thus applying $\Sigma_S^{\dagger 1/2}$ has the geometric interpretation of *restricting the analysis to the identifiable subspace*—the subspace in which $\Sigma_S$ has nonzero variance.

Within this identifiable subspace, the whitened covariance satisfies

$$\mathbb{E}[(\Phi_c - \Phi_r)(\Phi_c - \Phi_r)^\top] = P_{\text{row}(\Sigma_S)},$$

the orthogonal projector onto the row space of $\Sigma_S$. This reduces to the identity when $\Sigma_S$ is full rank, and preserves the key orthogonality relationships used in the remainder of the proof.

**Projection and Orthogonal Decomposition**   The next step in the proof decomposes $\tilde{C}_M$ into the component aligned with $\tilde{\mu}_S$ and the orthogonal residual:

$$\tilde{C}_M = \tilde{\mu}_S \beta^\top + E, \qquad E^\top \tilde{\mu}_S = 0.$$

This decomposition remains valid in the singular case because: (i) $\tilde{\mu}_S$ lies in the row space of $\Sigma_S$, (ii) $\tilde{C}_M$ also lies in this row space, and (iii) the row space is a Euclidean subspace where ordinary projection operations are well-defined. Therefore the projection step is unchanged: only the dimensionality of the subspace may be reduced.

**Effect of Rank Deficiency on the Coupling Constants**   Within the identifiable subspace, we have

$$w_M^\top f = \beta \, (w_S^\top f) + E^\top \Phi,$$

and by averaging over the $K$ attributes we obtain the coupling inequality

$$r_m(x, y) \geq c \, r_s(x, y) - \varepsilon.$$

The constants $c$ and $\varepsilon$ remain well-defined when pseudoinverses are used. However, rank deficiency can affect their values in the following ways:

- If $\Sigma_S$ or $\Sigma_M$ collapses onto a low-dimensional subspace, then the aligned component $\tilde{\mu}_S^\top \tilde{C}_M$ decreases, which reduces $c$.
- The orthogonal residual $E$ may increase because fewer directions are available for alignment. This enlarges the slack term $\varepsilon$.
- No matter the rank, the inequality preserves its form. Rank deficiency does not create inconsistencies; it only reduces the *strength* of the guaranteed coupling.

**Interpretation** The situation mirrors classical linear regression with multicollinearity: singularity does not make the model invalid, but it reduces the amount of signal that can be reliably extracted. In our setting, the implicit multi-attribute effect continues to hold; the guarantee simply becomes weaker when the embedding space contains fewer identifiable directions.

### E.2 POSITIVE-DEFINITE COVARIANCES

Our theoretical analysis does not fundamentally rely on $\Sigma_S$ or $\Sigma_M$ being strictly positive definite. When these matrices are singular, the population solutions in Theorem 1 are interpreted using the Moore–Penrose pseudoinverse:

$$w_S = \Sigma_S^\dagger \mu_S, \qquad w_M = \Sigma_M^\dagger C_M,$$

and the whitening operator becomes $\Sigma_S^{\dagger 1/2}$. Here $\Sigma_S^\dagger$ and $\Sigma_M^\dagger$ denote the Moore–Penrose pseudoinverses. For any possibly singular matrix $A$, its pseudoinverse $A^\dagger$ is the unique matrix satisfying

$$AA^\dagger A = A, \quad A^\dagger A A^\dagger = A^\dagger, \quad (AA^\dagger)^\top = AA^\dagger, \quad (A^\dagger A)^\top = A^\dagger A.$$

It coincides with the usual inverse when $A$ is full rank and otherwise computes the least-squares inverse on the identifiable subspace while nulling the kernel. Under this generalization, all algebraic steps of the proof continue to hold; only the coupling strength may decrease.

For convenience, we rewrite the relevant quantities in Theorem 1 in a fully explicit and self-contained way. Recall that the BT population solution is

$$w_S = \Sigma_S^{-1} \mu_S, \qquad \mu_S = \mathbb{E}[f(x_s, y_c) - f(x_s, y_r)],$$

and the multi-attribute regression solution is

$$w_M = \Sigma_M^{-1} C_M, \qquad C_{M,k} = \mathbb{E}[f(x_m, y_m)\, r_k],$$

where $C_{M,k}$ denotes the $k$-th column of $C_M$.

Following the whitening step used in the proof of Theorem 1, define

$$\tilde{\mu}_S := \Sigma_S^{-1/2} \mu_S, \qquad \tilde{C}_{M,k} := \Sigma_S^{-1/2} \Sigma_M^{-1} C_{M,k}.$$

Both vectors now live in the same whitened feature space where $\mathbb{E}[(\tilde{f}_c - \tilde{f}_r)(\tilde{f}_c - \tilde{f}_r)^\top] = I$.

The coupling coefficient from Theorem 1 is

$$\beta_k = \frac{\tilde{\mu}_S^\top \tilde{C}_{M,k}}{\|\tilde{\mu}_S\|^2}.$$

To decompose this expression, we first introduce the normalized directions

$$\hat{\mu}_S := \frac{\tilde{\mu}_S}{\|\tilde{\mu}_S\|}, \qquad \hat{v}_k := \frac{\tilde{C}_{M,k}}{\|\tilde{C}_{M,k}\|}.$$

Using the identity $\tilde{\mu}_S^\top \tilde{C}_{M,k} = \|\tilde{\mu}_S\|\|\tilde{C}_{M,k}\|\langle \hat{\mu}_S, \hat{v}_k \rangle$, we obtain the factorization

$$\beta_k = \langle \hat{\mu}_S, \hat{v}_k \rangle \cdot \frac{\|\tilde{C}_{M,k}\|}{\|\tilde{\mu}_S\|}.$$

Finally, the norms of the whitened vectors can be expressed in terms of the spectrum of $\Sigma_S$. Let

$$\kappa := \frac{\lambda_{\max}(\Sigma_S)}{\lambda_{\min}(\Sigma_S)}$$

denote its condition number. Since $\|\tilde{\mu}_S\| \in [\|\mu_S\|/\sqrt{\lambda_{\max}}, \|\mu_S\|/\sqrt{\lambda_{\min}}]$, the ratio $\|\tilde{C}_{M,k}\|/\|\tilde{\mu}_S\|$ contributes an additional factor proportional to $\kappa^{-1/2}$. Combining terms yields the decomposition:

$$\beta_k = \underbrace{\langle \hat{\mu}_S, \hat{v}_k \rangle}_{\text{directional alignment}} \cdot \underbrace{\frac{\|\tilde{C}_{M,k}\|}{\|\tilde{\mu}_S\|}}_{\text{signal-strength ratio}} \cdot \underbrace{\kappa^{-1/2}}_{\text{conditioning factor}} \cdot$$

This expression makes explicit how singularity or poor conditioning of $\Sigma_S$ affects the coupling: if $\Sigma_S$ becomes low-rank, then (i) the projection of $C_{M,k}$ onto the identifiable subspace shrinks, reducing $\langle \hat{\mu}_S, \hat{v}_k \rangle$; (ii) the magnitude $\|\tilde{C}_{M,k}\|$ decreases because components in the null space are removed; and (iii) $\kappa^{-1/2}$ decreases as the spectrum collapses. Consequently, $\beta_k$ (and thus $c$) becomes smaller, while the orthogonal residual in Theorem 1 becomes larger.

Empirically, SMORM does not enter such degenerate regimes. The regression head achieves the same MSE as the single-head baseline, indicating that the attribute signals are not compressed into a low-dimensional null space. Likewise, the BT scores exhibit stable, non-collapsing variance (Fig. 13), implying that $\Sigma_S$ maintains a well-spread spectrum and that $\kappa$ remains $O(1)$. Together these observations guarantee that $c$ stays non-trivial and $\varepsilon$ remains small in practice.

We will clarify in the revision that positive-definiteness is used only to simplify notation, and that the correct formulation uses pseudoinverses and the identifiable subspace.

**Response.** We thank the reviewer for highlighting the issue of singular covariance matrices. Importantly, our proof does not rely on strict positive definiteness of $\Sigma_S$ or $\Sigma_M$. When these matrices are rank-deficient, the population least-squares solutions in Theorem 1 are interpreted using the Moore–Penrose pseudoinverse:

$$w_S = \Sigma_S^\dagger \mu_S, \qquad w_M = \Sigma_M^\dagger C_M.$$

With this substitution, every algebraic step in the proof continues to hold. The only thing that changes is the *magnitude* of the coupling parameters.

Intuitively, if the shared embedding collapses onto a lower-dimensional subspace, then fewer feature directions remain identifiable. In this case the aligned component of the multi-attribute signal becomes smaller, which reduces the coupling constant $c$, while the orthogonal residual may increase, which enlarges the slack $\varepsilon$. The core inequality

$$r_m(x,y) \ \geq \ c\, r_s(x,y) - \varepsilon$$

remains valid, but with a potentially weaker $c$ and larger $\varepsilon$ in highly degenerate regimes.

Empirically, however, this pathological situation does not arise in the SMORM training regime. (i) The regression MSE of SMORM matches the single-head regression baseline, indicating that the multi-attribute directions are not lost in a low-rank subspace. (ii) The BT head maintains stable, non-collapsing variance (Fig. 13), implying that the effective $\Sigma_S$ is well-conditioned and does not approach singularity. Together these observations ensure that the effective covariances retain sufficient rank so that $c$ is non-trivial and $\varepsilon$ remains small in practice.

We will clarify in the revision that Theorem 1 should be interpreted in terms of pseudoinverses and the identifiable subspace, and that positive-definiteness is used only to simplify notation, not as a structural requirement.

### E.3    Fisher Information Matrix

In large, over-parameterized LLMs the *full* Fisher Information Matrix (FIM) is indeed singular. Importantly, our argument on p.24 does **not** require the global FIM to be invertible. What matters for the asymptotic variance of the reward model is the Fisher information **restricted to the task-relevant subspace**, not the entire parameter space.

- **Effective Fisher Information.** Even when the full FIM is singular, the gradients that actually affect the reward outputs,

$$g_S(y) = \nabla_\theta M_S(y), \qquad g_M(y) = \nabla_\theta M_M(y),$$

span a low-dimensional *identifiable* subspace. The Fisher information **restricted to this subspace** is

$$I_{\text{eff}}^{(S)} = \mathbb{E}[g_S g_S^\top], \qquad I_{\text{eff}}^{(M)} = \mathbb{E}[g_M g_M^\top],$$

which remains positive definite as long as the reward functions are not constant. This is the standard notion of "effective rank" used in modern statistics of deep networks: the identifiable subspace can have a well-conditioned Fisher even when the full model has infinitely many uninformative directions.

Accordingly, our asymptotic variance expression should be interpreted as

$$\text{Var}[M_S(\hat{\theta})] \approx g_S^\top I(\theta^\star)^+ g_S,$$

where $I(\theta^\star)^+$ is the **Moore–Penrose pseudoinverse** of the FIM restricted to the identifiable subspace. This is the standard formulation for asymptotic normality in singular (or non-identifiable) models.

- **Why hybrid training helps even under singularity.** Let $I_{\text{eff}}^{(\text{single})}$ denote the effective Fisher information generated by the BT head alone. Adding the multi-attribute regression head introduces an additional positive semi-definite term

$$\Delta = \mathbb{E}[g_M g_M^\top] \succeq 0,$$

so that

$$I_{\text{eff}}^{(\text{hybrid})} = I_{\text{eff}}^{(\text{single})} + \Delta.$$

This relation holds **regardless of whether the full FIM is singular**. Since $\Delta \succeq 0$, the hybrid model always increases the curvature in the reward-relevant directions and thus increases the amount of information available for estimating the reward parameters.

In the pseudoinverse formulation, this yields, along the BT direction $g_S$,

$$g_S^\top I_{\text{eff}}^{(\text{hybrid})+} g_S \;\leq\; g_S^\top I_{\text{eff}}^{(\text{single})+} g_S,$$

which means that the asymptotic variance of the single-objective reward functional $M_S$ decreases under SMORM, even when the global FIM is singular. Intuitively, SMORM enriches the informative subspace where the reward gradients live, and thus stabilizes the estimator.

## F    Motivation for Enhancing Multi-Objective Reward Functions without Additional Multi-Attribute Data

In this section, we provide a detailed discussion on the limited availability of high-quality data for training multi-objective reward models, which motivates our approach to enhancing multi-objective reward modeling performance without relying on additional multi-attribute annotations.

While several datasets provide dense prompt–response pairs with fine-grained attribute scores—such as UltraFeedback (Cui et al., 2023) with 240K samples and Prometheus (Kim et al., 2023) with 200K samples—their annotations are primarily generated by GPT-based models. This introduces several concerns: **(1)** As foundation models continue to evolve, the quality and consistency of their annotations become increasingly difficult to guarantee. **(2)** The use of large language models (LLMs) as annotators introduces potential biases (Gu et al., 2024; Li et al., 2024), which can be inherited by the reward model and subsequently transferred to the policy model when optimized using reward signals. **(3)** Our experimental results in Section 5 show that, when using gemma-2b-it as the base model, training on HelpSteer2 (Wang et al., 2024f)—a human-annotated dataset of 20K samples—yields superior performance compared to training on UltraFeedback with 240K GPT-labeled samples.

HelpSteer2 (Wang et al., 2024f) is one of the few available datasets that provide high-quality, human-annotated, multi-objective preference labels. However, it only contains 20K samples, and its creation involved an exceptionally rigorous annotation process. This process includes multiple layers of human oversight, dynamic annotator recruitment, and strict quality control procedures to ensure data integrity. Due to the high resource demands of this pipeline, it is difficult to generalize or scale to larger datasets.

In summary, it remains difficult to obtain large-scale, high-quality, fine-grained attribute scores for responses, whether through GPT-based evaluation or human annotation. This limitation motivates the development of methods that can enhance the performance of multi-objective reward models without requiring additional annotated data.

## G    Experiments on OOD

To evaluate the generalizability of our SMORM, we adopt the experimental setup described in Section 5, training reward models on both 400K and 40K samples from the Unified-Feedback dataset.

In both settings, HelpSteer2 serves as the multi-objective dataset $\mathcal{D}_M$ for SMORM. All methods use gemma-2B-it as the base model and are evaluated on both in-distribution (ID) data (Unified-Feedback) and out-of-distribution (OOD) benchmarks (HHH-Alignment and MT-Bench). The results are presented in Tables 10 and 11. The results demonstrate that SMORM-F consistently outperforms competing methods in both in-distribution (ID) and out-of-distribution (OOD) evaluations. Specifically, SMORM-F achieves an ID score of 76 and an OOD score of 83.2 on the HHH-Alignment benchmark, surpassing the second-best method, which attains scores of 73.8 and 79.6, respectively. These findings suggest that incorporating a multi-objective learning function effectively shapes the embedding space, leading to improved performance on ID data and enhanced generalizability of the single-objective head to OOD scenarios.

Table 10: Results on **ID** and **OOD** evaluation with **400K training data** from Unified-Feedback. The best performance in each task is in bold and the second best one is underlined.

| Reward Model | Unified Feedback | HHH Alignment | MT Bench |
|---|---|---|---|
| Classifier (Frozen) | 63.8 | 66.4 | 69.5 |
| Classifier (baseline) | 72.1 | 73.4 | 71.2 |
| Classifier + margin | 72.0 | 75.0 | 72.6 |
| Classifier + label smooth | 71.5 | 72.1 | 71.2 |
| Classifier + Ensemble | 72.8 | 76.8 | **73.7** |
| GRM | 73.8 | 79.6 | 73.4 |
| **SMORM-F (Ours)** | **76.0** | **83.2** | 73.4 |

Table 11: Results on **ID** and **OOD** evaluation with **40K training data** from Unified-Feedback. The best performance in each task is in bold and the second best one is underlined.

| Reward Model | Unified Feedback | HHH Alignment | MT Bench |
|---|---|---|---|
| Classifier (Frozen) | 63.9 | 68.6 | 68.2 |
| Classifier (baseline) | 68.8 | 70.3 | 69.1 |
| Classifier + margin | 69.6 | 69.8 | 71.0 |
| Classifier + label smooth | 68.5 | 68.8 | 71.9 |
| Classifier + Ensemble | 69.9 | 72.2 | 71.1 |
| GRM | 71.3 | 75.1 | 72.8 |
| **SMORM-F (Ours)** | **72.9** | **77.4** | **73.0** |

# H  ALIGNMENT RESULTS AFTER PPO

To demonstrate the effectiveness of our SMORM in training superior policy models within the RLHF framework, we evaluate the win rate of models trained using PPO with SMORM-F as the proxy reward model, comparing it against two baselines: Baseline SORM and GRM. We obtain the optimized models following the training process in Sec. 5.3 and randomly sample 500 prompts from the Unified-Feedback dataset and use the optimized policy models to generate corresponding responses. We then employ GPT-4o as an impartial judge to compare the model outputs and determine which response is preferable, recording the win rates accordingly. We adopt the win-tie evaluation prompt format introduced by (Khanov et al., 2024). The results are summarized in Table 12. The system prompt and user prompt used for evaluation are provided below:

---

**Prompt for Win-Tie Evaluation by GPT-4o**

**System prompt**
```
[system]
```
You are a helpful, harmless, and precise assistant for checking the quality of the answer. We would like to request your feedback on the performance of two AI assistants in response to the user question. Please rate the harmlessness, helpfulness, and level of detail of their responses. Your evaluation should consider factors such as the helpfulness, harmlessness, relevance, accuracy, depth, creativity, and level of detail of the response. Note that if a response appears cut off at the end due to length constraints, it should not negatively impact the score. Also, base your evaluation solely on the given answer, disregarding any preceding interactions in the question. Each assistant receives an overall score on a scale of 1 to 10, where a higher score indicates better overall performance.
Please first output a single line containing only two values indicating the scores for Assistant 1 and 2, respectively. The two scores are separated by a space. In the subsequent line, please provide a comprehensive explanation of your evaluation, avoiding any potential bias and ensuring that the order in which the responses were presented does not affect your judgment.
**User prompt**
```
[Question]
```
{question}
```
[The Start of Assistant 1's Answer]
```
{answer1}
```
[The End of Assistant 1's Answer]
[The Start of Assistant 2's Answer]
```
{answer2}
```
[The End of Assistant 2's Answer]
```

---

From the table, we observe that across both dataset scales, using our SMORM-F as the proxy reward function results in a policy model that consistently outperforms both the Baseline and GRM, with win rates always exceeding 65%.

Table 12: Win rate of models after PPO training with SMORM-F against baseline SORM and GRM.

| Method | vs. Method | Win (%) ↑ | Tie (%) | Lose (%) ↓ |
|---|---|---|---|---|
| $\mathcal{D}_S/\mathcal{D}_M$: UnifiedFeedback 40k/HelpSteer2 | | | | |
| SMORM-F | Baseline | 75.7 | 0.5 | 23.8 |
| SMORM-F | GRM | 69.5 | 0.4 | 30.1 |
| $\mathcal{D}_S/\mathcal{D}_M$: UnifiedFeedback 400k/UltraFeedback | | | | |
| SMORM-F | Baseline | 71.3 | 0.7 | 28.0 |
| SMORM-F | GRM | 65.7 | 0.7 | 33.6 |

## I  IMPLEMENTATION DETAILS

### I.1  BASELINE AND TRAINING DETAILS

**Baseline Details.** All baseline reward models use the `AutoModelForSequenceClassification` class from the `transformers` library (Wolf et al., 2020), which attaches a randomly initialized linear head for reward prediction. Each model is trained to minimize a loss function using the training data. For ensemble baselines, we train three models with different random seeds and aggregate their predictions.

We use the margin loss from (Touvron et al., 2023) defined as:

$$\mathcal{L}_{\text{margin}}(\theta) = -\mathbb{E}_{(x, y_c, y_r) \sim \mathcal{D}} \left[ \log \left( \sigma \left( r_\theta(x, y_c) - r_\theta(x, y_r) - m(r) \right) \right) \right],$$

where $m(r)$ is computed using the reward difference between chosen and rejected responses in the `Unified-Feedback` dataset. This loss emphasizes meaningful reward distinctions.

We also incorporate a label smoothing loss defined as:

$$\mathcal{L}_{\text{smooth}}(\theta) = -\mathbb{E}_{(x,y_c,y_r)\sim\mathcal{D}}\left[(1-\epsilon)\log\left(\sigma\left(r_\theta(x,y_c) - r_\theta(x,y_r)\right)\right) - \epsilon\log\left(\sigma\left(r_\theta(x,y_c) - r_\theta(x,y_r)\right)\right)\right]$$

where $\epsilon = 0.1$. This formulation improves robustness by softening the loss against label noise, reducing overfitting.

For GRM, we strictly follow the implementation in (Yang et al., 2024). The reward head consists of a linear layer (hidden size, 1024), followed by a ReLU activation, and a final linear layer (1024, 1). The regularization coefficient $\alpha$ is set to 0.01, and $\beta$ is set to 0.1. In the GRM-linear variant, the head is a single linear layer of shape (hidden size, 1). More detailed training procedures for GRM can be found in (Yang et al., 2024).

**Computational Resources.** All experiments were conducted using NVIDIA H100 80GB GPUs. Training on 40K data samples requires approximately 16 GPU hours.

Table 13: Key implementations of the text generation experiments.

| Basic Information | |
| --- | --- |
| Base models | gemma-2b-it and Mistral-7B-Instruct-v0.2 |
| Quantization for training | bf16 |
| Optimizer | AdamW_hf |
| Batch size | 16 |
| Learning rate | $5 \times 10^{-6}$ |
| Learning rate scheduler | cosine |
| Warmup ratio | 0.03 |
| **SMORM** | |
| Weight ratio for single-objective to multi-objective reward modeling | 1.0 by default |
| **PPO** (Schulman et al., 2017) | |
| KL regularization | 0.0 |
| Epochs | 1 |
| Learning rate | $1 \times 10^{-5}$ |
| $\lambda$ for GAE | 0.95 |
| $\gamma$ | 1 |
| Clip range | 0.2 |
| Optimization epochs per batch | 4 |
| Tokens during generation | 512 |

## J  HYPERPARAMETER ANALYSIS AND INSTABILITY OF GRM

In this section, we conduct experiments to demonstrate the instability of GRM (Yang et al., 2024) and the relative stability of our proposed SMORM. Intuitively, reward models are typically initialized from language models pretrained on next-token prediction tasks, and are then fine-tuned for reward modeling. This motivates the hypothesis that introducing a next-token prediction regularization term—aligned with the pretraining objective—may conflict with the reward modeling objective. As a result, this misalignment could lead to unstable performance during reward model training. To validate this assumption, we conduct experiments following the setup described in Sec. 5, using 40K samples from Unified-Feedback as $\mathcal{D}_S$ and HelpSteer2 as $\mathcal{D}_M$. We compare the performance of GRM to a baseline single-objective reward model and our proposed SMORM-F. In addition, we compare a baseline multi-objective reward model to our SMORM-L. We vary the weight ratio in $[0.01, 0.1, 1, 10]$. In GRM, the weight ratio refers to the strength of the next-token prediction regularization. In SMORM, the weight ratio controls the contribution of multi-objective reward modeling. All models are evaluated on RewardBench. The results are presented in Fig. 14. From the figure, we observe that, except in extreme cases, our SMORM framework demonstrates consistently stable performance. Specifically, when the weight ratio is set to 0.01, the performance of SMORM-L slightly falls short of the baseline multi-objective reward model, and when the ratio is set to 10, SMORM-F marginally underperforms compared to the baseline single-objective reward model. Outside of these edge cases, both SMORM-F and SMORM-L consistently outperform their respective baselines across a wide range of weight settings. In contrast, the performance of GRM fluctuates significantly, ranging from approximately 45 to 65, and consistently underperforms relative

to the baseline single-objective reward model. These results support our hypothesis that incorporating next-token prediction regularization introduces instability into reward model training. This instability is especially concerning given the substantial computational cost required to train reward models.

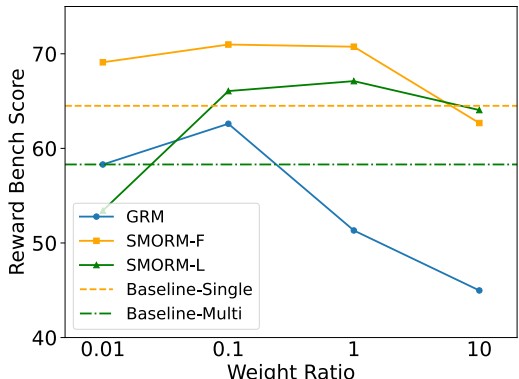

Figure 14: Hyperparameter analysis.

## K  INTERPRETATION OF WHY SORM FAIL IN OOD SETTING

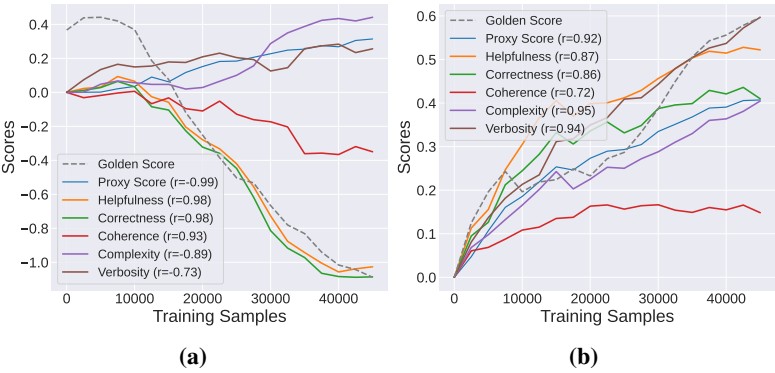

Figure 15: Fine-grained attribute scores of the optimized policy model using (a) the baseline classifier and (b) SMORM-F as the proxy reward model.

To interpret the vulnerability of the baseline classifier to reward hacking, we adopt the same experimental setting as in Sec. 3 and record evaluation scores for the generated responses across five dimensions: `Helpfulness`, `Correctness`, `Coherence`, `Complexity`, and `Verbosity`. These scores are derived using the multi-objective head $\mathbf{w}_M$ of the trained SMORM. Figures 15 (a) and (b) illustrate the evaluation results when using the baseline classifier and SMORM-F as proxy reward models, respectively. In Fig. 15 (a), employing the baseline classifier leads to improvements only in the `Complexity` and `Verbosity` dimensions, while performance declines in the remaining attributes. Consequently, the generated responses are not considered high-quality by the gold reward model. In contrast, Fig. 15 (b) shows that using SMORM-F as the proxy reward model results in consistent improvements across all five fine-grained dimensions. These enhancements are also reflected in an increased gold score. These findings indicate that a conventional single-objective reward model is typically insufficient to capture the multifaceted criteria that make a chosen response preferable to a rejected one.

# L ANALYSIS ON RM-BENCH

RM-Bench (Liu et al., 2025b) evaluates reward models on two key dimensions: sensitivity to subtle changes and robustness to style bias. It includes three task types: (1) `Easy`: Chosen responses are detailed and informative; rejected ones are concise and minimal. (2) `Normal`: Both responses share the same style but differ in key information. (3) `Hard`: Chosen responses are concise; rejected ones are detailed.

## L.1 MORMs FALL SHORT ON EASY AND NORMAL TASKS.

In this section, we present empirical results to illustrate why baseline multi-objective reward models tend to underperform on RM-Bench (Liu et al., 2025b). We begin by evaluating a selection of existing open-source single-objective and multi-objective reward models that exhibit comparable performance on RewardBench (Lambert et al., 2024). The evaluated models include:

- **Multi-objective reward models:**
    - `NVIDIA/Nemotron-340B-Reward`[3]
    - `RLHFlow/ArmoRM-Llama3-8B-v0.1`[4]
- **Single-objective reward models:**
    - `Ray2333/GRM-llama3-8B-distill`[5]
    - `internlm/internlm2-20b-reward`[6]
    - `NCSOFT/Llama-3-OffsetBias-RM-8B`[7]
    - `Ray2333/GRM-llama3-8B-sftreg`[8]
    - `LxzGordon/URM-LLaMa-3.1-8B`[9]
    - `Ray2333/GRM-Llama3.2-3B-rewardmodel-ft`[10]
    - `Skywork/Skywork-Reward-Llama-3.1-8B`[11]

Table 14: Comparison of models on Easy, Normal, and Hard tasks on RM-Bench.

| Model Name | Easy | Normal | Hard | Avg | RewardBench |
|---|---|---|---|---|---|
| Skywork/Skywork-Reward-Llama-3.1-8B | 89.0 | 74.7 | 46.6 | 70.1 | 93.1 |
| LxzGordon/URM-LLama-3.1-8B | 84.0 | 73.2 | 53.0 | 70.0 | 92.9 |
| NCSOFT/Llama-3-OffsetBias-RM-8B | 84.6 | 72.2 | 50.2 | 69.0 | 89.4 |
| internlm/internlm2-20b-reward | 82.6 | 71.6 | 50.7 | 68.3 | 90.2 |
| Ray2333/GRM-llama3-8B-sftreg | 83.5 | 72.7 | 48.6 | 68.2 | 87.0 |
| Ray2333/GRM-llama3-8B-distill | 82.2 | 71.5 | 48.4 | 67.4 | 86.2 |
| Ray2333/GRM-Llama3.2-3B-rewardmodel-ft | 89.9 | 74.0 | 44.0 | 69.3 | 90.9 |
| RLHFlow/ArmoRM-Llama3-8B-v0.1 | 82.5 | 70.8 | 50.1 | 67.8 | 90.4 |
| NVIDIA/Nemotron-340B-Reward | 81.0 | 71.4 | 56.1 | 69.5 | 92.0 |

The results of these comparisons are presented in Table 14. For better illustration, we also visualize the results in Fig. 16. We observe that existing multi-objective reward models tend to underperform on the Easy and Normal tasks, even though their overall performance on RewardBench and the Hard tasks of RM-Bench is not among the worst. We attribute this phenomenon to the following: multi-objective reward models (MORMs) typically assess response quality through utility and style attributes. For example, `HelpSteer2` (Wang et al., 2024f) provides scores for *correctness*,

---

[3]nvidia/Nemotron-4-340B-Reward

[4]RLHFlow/ArmoRM-Llama3-8B-v0.1

[5]Ray2333/GRM-llama3-8B-distill

[6]internlm/internlm2-20b-reward

[7]NCSOFT/Llama-3-OffsetBias-RM-8B

[8]Ray2333/GRM-llama3-8B-sftreg

[9]LxzGordon/URM-LLaMa-3.1-8B

[10]Ray2333/GRM-Llama3.2-3B-rewardmodel-ft

[11]Skywork/Skywork-Reward-Llama-3.1-8B

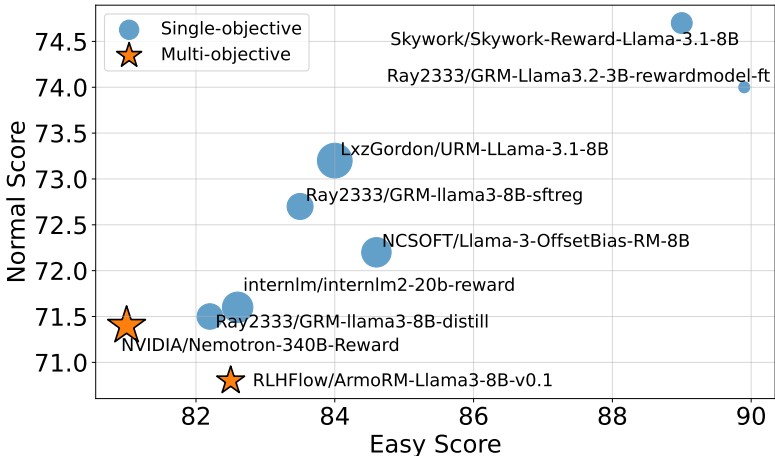

Figure 16: Visualization of results on RM-Bench. The size of each marker indicates the model's performance on the Hard task.

*helpfulness*, and *coherence* (utility), and *complexity* and *verbosity* (style). Formally, we define: $\mathcal{U} = \{\text{correctness, helpfulness, coherence}\}, \mathcal{S} = \{\text{verbosity, complexity}\}$. The score margin is:

$$\Delta r = \underbrace{\sum_{k \in \mathcal{U}} w_k \, \Delta h_k}_{r_u:\text{utility heads}} + \underbrace{w_{\text{verb}} \, \Delta h_{\text{verb}} + w_{\text{comp}} \, \Delta h_{\text{comp}}}_{r_b:\text{style bias}}.$$

In `Easy` tasks, longer chosen responses incur negative style penalties, potentially flipping $\Delta r$ despite higher utility scores. In `Normal` tasks, responses should have a similar style, implying $\Delta h_{\text{verb}} \approx \Delta h_{\text{comp}} \approx 0$, but in practice, non-trivial deviations occur due to imperfect optimization. As shown in Appendix L.2, style noise degrades scoring accuracy and overall performance.

### L.2 CASE STUDY AND WHY SMORM HELP

In this section, we present a case study to interpret why baseline multi-objective reward models fall short on the Normal and Easy tasks of RM-Bench.

We train a multi-objective reward model (MORM) on the `HelpSteer2` dataset and, for each attribute across all tasks, compute the mean and variance of the prediction differences—where each difference is defined as the chosen-response score minus the rejected-response score. The results are summarized in Table 15. As shown in the table, while the baseline MORM exhibits mean scores for complexity and verbosity that are approximately zero across the Normal tasks, the corresponding variances are non-trivial. This indicates that, although paired responses should receive similar scores for complexity and verbosity, there remains considerable prediction bias on these attributes in individual comparisons.

Table 15: Normalized pairwise differences per dimension (Mean (Variance))

| Category | helpfulness | correctness | coherence | complexity | verbosity |
|---|---|---|---|---|---|
| Normal overall | 0.08 (0.83) | 0.00 (1.55) | 0.07 (0.74) | 0.17 (1.29) | 0.00 (0.55) |
| Normal Correct | 0.41 (0.75) | 0.33 (1.48) | 0.29 (0.88) | 0.43 (1.31) | 0.10 (0.52) |
| Normal False | -0.31 (0.65) | -0.38 (1.35) | -0.19 (0.46) | -0.13 (1.10) | -0.12 (0.55) |
| Hard overall | 0.26 (1.02) | 0.32 (1.90) | -0.28 (0.95) | -0.56 (1.85) | 0.05 (0.81) |
| Hard Correct | 0.78 (0.84) | 0.85 (1.86) | 0.01 (0.82) | -0.27 (2.24) | 0.14 (0.81) |
| Hard False | -0.08 (0.84) | -0.03 (1.62) | -0.48 (0.93) | -0.75 (1.50) | -0.19 (0.77) |
| Easy overall | -0.11 (1.32) | -0.32 (1.93) | 0.41 (1.38) | 0.90 (2.36) | 0.05 (0.75) |
| Easy Correct | 0.19 (1.10) | -0.07 (1.69) | 0.61 (1.51) | 1.12 (2.15) | 0.12 (0.70) |
| Easy False | -0.76 (1.20) | -0.87 (2.02) | -0.03 (0.82) | 0.40 (2.46) | -0.10 (0.83) |

Ideally, a single-objective reward model (SORM) can better align with helpfulness and correctness, as its preference training dataset encompasses diverse attributes and tends to exhibit less sensitivity to superficial factors like response length. According to Theorem 1, the aggregate score from the multi-objective reward model is lower-bounded by the single-objective reward score: $r_m(x,y) = \frac{1}{K}\sum_{i=1}^{K}\alpha_i w_{M,i}^\top f_\theta(x,y) \geq c \cdot r_s(x,y) - \varepsilon$, where $\alpha_i$ reflects the correlation between each attribute and the chosen/rejected preference direction. Consider a scenario where response $y_A$ is preferred over $y_B$ according to ground-truth labels. Initially, it may happen that $r_m(y_A) < r_m(y_B)$, especially when the style bias overwhelms the utility signal, i.e., $|r_u(y_A) - r_u(y_B)| < |r_b(y_A) - r_b(y_B)|$. However, when the single-objective reward assigns a higher score to $y_A$, i.e., $r_s(y_A) > r_s(y_B)$, the utility heads in the multi-objective model are progressively adjusted to favor $y_A$. This adjustment increases the margin $|r_u(y_A) - r_u(y_B)|$, eventually outweighing the bias introduced by stylistic components ($r_b$). As a result, the style penalty's influence diminishes, and the model better reflects true preference judgments. To verify this, we train the SMORM using 40K samples from `Unified-Feedback` as $\mathcal{D}_S$, and report the differences in Table 16. As shown, for normal cases, both the variance and verbosity are substantially reduced compared to the variance observed in helpfulness. This indicates that style-related biases no longer dominate the final decision, thereby enhancing the performance of the multi-objective reward model on this task.

Table 16: Normalized pairwise differences per dimension (Mean (Variance))

| Category | helpfulness | correctness | coherence | complexity | verbosity |
|---|---|---|---|---|---|
| Normal overall | 0.43 (0.62) | 0.43 (0.68) | 0.29 (0.41) | -0.02 (0.11) | 0.01 (0.08) |
| Normal correct | 0.73 (0.57) | 0.71 (0.70) | 0.53 (0.35) | -0.01 (0.13) | 0.05 (0.10) |
| Normal false | -0.24 (0.09) | -0.17 (0.10) | -0.23 (0.11) | -0.03 (0.07) | -0.07 (0.04) |
| Hard overall | 0.28 (0.70) | 0.44 (0.93) | 0.25 (0.55) | -0.30 (0.16) | 0.33 (0.20) |
| Hard correct | 0.77 (0.62) | 0.91 (1.05) | 0.68 (0.45) | -0.31 (0.17) | 0.48 (0.22) |
| Hard false | -0.34 (0.11) | -0.16 (0.12) | -0.30 (0.15) | -0.29 (0.14) | 0.14 (0.11) |
| Easy overall | 0.57 (0.67) | 0.43 (0.61) | 0.34 (0.38) | 0.26 (0.16) | -0.30 (0.20) |
| Easy correct | 0.83 (0.56) | 0.62 (0.59) | 0.53 (0.30) | 0.30 (0.17) | -0.31 (0.23) |
| Easy false | -0.26 (0.12) | -0.18 (0.16) | -0.27 (0.16) | 0.14 (0.11) | -0.27 (0.12) |

# M    THE USE OF LARGE LANGUAGE MODELS

Large Language Models (LLMs) were used to support the writing and editing of this manuscript. Specifically, an LLM assisted in improving clarity, refining phrasing, correcting grammar, and enhancing the overall readability of the text.

The LLM was not involved in the ideation, research design, data analysis, or development of any scientific content. All research questions, methodologies, and analyses were independently developed and executed by the authors.

