# OpenReview forum: "Bradley-Terry and Multi-Objective Reward Modeling Are Complementary"
_ICLR.cc/2026/Conference — ICLR 2026 Poster_

### Official Review · Reviewer_u8z6 · 2025-10-25

**Soundness:** 2
**Presentation:** 3
**Contribution:** 2
**Rating:** 4
**Confidence:** 3

**Summary:**

This paper investigates the problem of reward hacking in reinforcement learning from human feedback, particularly under out-of-distribution (OOD) conditions. The authors propose SMORM, a unified framework that jointly trains a Bradley–Terry (BT) single-objective reward model and a multi-objective regression-based reward model using a shared embedding space and provided theoretical support on the connection between the BT loss and the regression objective and highlight their complementary benefits.

**Strengths:**

1. The paper is well-structured and clearly presented

2. The idea of co-training BT and multi-objective regression heads under a shared embedding space is new and well-motivated.

3. The proposed methods is supported by empirical evaluations, under both in-distribution (ID) and out-of-distribution (OOD) settings.

**Weaknesses:**

1. While the joint-training idea is new, the method itself (two linear heads sharing embeddings) is architecturally simple and appears as a simple combination of existing method. Therefore, in my view, the main novelty lies in the theoretical analysis of complementary benefits of BT and multi-objective reward modeling.

2. The theoretical results rely on several strong and unverifiable assumptions, such as positive-definite covariances and global positive correlation between single-objective and multi-objective reward signals. In practice, it is unclear whether these assumptions are satisfied in noisy environments.

3. No code provided currently.

**Questions:**

1. Under the BT model, rewards are identifiable only up to an additive constant. In Theorem 1, the two sides may implicitly live on different scales. How do you resolve the shift when coupling BT and regression heads?

2. The shared embedding f typically lives in high dimensions; in practice its covariance matrices can be singular. Many proofs assume positive definiteness. If this fails (e.g., rank deficiency due to collinearity), what precisely degrades?

3. With massive over-parameterization, the Fisher Information Matrix  for LLM parameters is typically singular. Page 24’s asymptotic variance claims appear to rely on an invertible FIM is unlikely to hold. Thus, it requires further clarification.

---

> ### Author Response · Authors · 2025-11-21
>
> We sincerely thank the reviewer for acknowledging the novelty of our method and the strength of our experimental results. Below, we provide a point-by-point response to the reviewer’s comments:
>
> ------------
>
> >### W1: simple architecture
>
> We are grateful to the reviewer for recognizing that **SMORM introduces a new perspective** and that our paper provides **novel theoretical insights into the complementary benefits of BT and multi-objective reward modeling**.
>
> While the reviewer noted that the SMORM framework appears simple, we respectfully emphasize that **this simplicity is intentional and, in fact, a key strength of the approach**.
>
> First, as the reviewer also highlighted, **SMORM is supported by theoretical guarantees**, demonstrating that even with a lightweight design, the framework provably enriches both BT and multi-objective reward learning. In this sense, the simplicity is not a limitation but a reflection of a **principled and theoretically grounded design** that avoids unnecessary architectural complexity.
>
> Second, SMORM’s simplicity offers substantial **practical advantages**. The method requires only the initialization of additional prediction heads and does **not** rely on aligned datasets for BT and multi-objective supervision. This greatly **improves its applicability in real-world training pipelines**.
>
> In summary, we believe that **SMORM’s theoretically justified yet simple architecture is a feature—not a drawback**. Its elegance enables both ease of adoption and strong empirical performance, which together underscore its broad practical utility.
>
> -------
>
> >### W2: Global Positive Correlation Between Single-Objective and Multi-Objective Reward Signals
>
> We thank the reviewer for raising this insightful question. The assumption of a global positive correlation between single-objective and multi-objective reward signals implies that, in general, when two responses are compared, the one with a higher aggregated multi-attribute score is also preferred according to a strong single-objective reward model. This principle reflects an intuitive and commonly observed alignment and has been empirically justified in prior work [1].
>
> To directly address the reviewer’s concern, we conduct an empirical test using the well-trained single-objective reward model `Skywork/Skywork-Reward-V2-Llama-3.1-8B`. Specifically, we use this model to score all response pairs from the `nvidia/HelpSteer2` dataset and compare the pairwise preference predictions with the ground-truth aggregate multi-attribute scores. We find that in 73.02% of cases, the pairwise preference aligns with the direction of the aggregated multi-attribute score, which can be considered a strong correlation. Although this setting is not ideal and inherently noisy due to the complexity of human annotation process, our SMORM consistently achieves superior performance in enhancing both reward functions. This empirical robustness supports the effectiveness of SMORM in real-world scenarios.
> | Attribute |Correlation  |
> |--|--|
> | helpfulness | 74.88\% |
> | correctness | 74.37\% |
> | coherence | 72.75\% |
> | complexity | 62.72\% |
> | verbosity |  58.96\%|
> | overall |  73.02\%|

---

> ### Author Response · Authors · 2025-11-21
>
> >### W2 & Q1: difference scale of two reward functions
>
> We appreciate the reviewer for raising such insightful question. While BT scores are indeed identifiable only up to an additive constant, this ambiguity does not affect the coupling in Theorem 1, nor does it cause instability in practice. We provide both theoretical and empirical clarifications.
>
>
>
> - **Theoretical invariance**. Theorem 1 provides the coupling inequality
> $$
> r_m(x,y) \;\ge\; c\,r_s(x,y)\;-\;\varepsilon,
> $$
> where $c$ and $\varepsilon$ depend only on the second-order feature statistics $\mu_S=\mathbb{E}[f_c-f_r]$ and $\Sigma_S=\mathbb{E}[(f_c-f_r)(f_c-f_r)^\top]$, which in turn depend solely on BT *score differences*. Thus a global shift in the BT head, $r_s' = r_s + C$, leaves $\mu_S$, $\Sigma_S$, and therefore $c$ unchanged. Substituting $r_s'$ into the bound gives
> $$
> r_m(x,y)
> \;\ge\;
> c(r_s(x,y)+C)-\varepsilon
> c\,r_s'(x,y) - (\varepsilon - cC).
> $$
> Defining $\varepsilon'=\varepsilon - cC$ yields the same inequality form,
> $$
> r_m(x,y)\;\ge\;c\,r_s'(x,y)-\varepsilon',
> $$
> showing that an additive shift in $r_s$ is fully absorbed into the slack term and **has no effect on the coupling strength $c$ or BT preference probabilities** (which depend only on $r_s(y_c)-r_s(y_r)$). Therefore, the BT identifiability ambiguity does not affect Theorem 1 nor the behavior of SMORM.
>
> - **Empirically**, we use google/gemma-2-2b-it as the base model and train SMORM on two datasets: the preference dataset Skywork/Skywork-Reward-Preference-80K-v0.2 and the multi-objective reward dataset nvidia/HelpSteer2. We observe the regression head does not shift. **In Figure 7 and 9 in Appendix**, we explicitly monitored the multi-attribute regression head during training. Its MSE(Mean Squared Error) loss under SMORM is nearly identical to the baseline (multi-objective regression) model, which indicates that the regression head remains stably calibrated to the ground-truth label scale. This confirms that joint training does not introduce any shift or scale instability into $r_m$.
>
> - We also track the mean and variance of the BT score $r_s$​ during training and observe that $r_s$​ undergoes an early shift in its mean before eventually stabilizing. **As shown in Figure 13, during the initial phase of training—when the regression MSE is still decreasing—the mean of $r_s$ gradually shifts**. **This pattern demonstrates that SMORM automatically reconciles the scale mismatch through its joint training objective, consistent with the implications of Theorem 1.** Once the regression loss stabilizes, the mean of $r_s$ also stabilizes and no further drift occurs.
>
>
> Thus, during joint training, the regression head effectively **anchors the shared representation** to an absolute scale dictated by the multi-attribute scores. The BT head, which shares the underlying parameters, naturally inherits this calibrated scale. Consequently, **SMORM automatically resolves the inherent scale mismatch between binary preference signals and continuous multi-attribute scores, allowing both heads to operate within a unified and well-calibrated embedding space**.

---

> ### Author Response · Authors · 2025-11-21
>
> >### W2 & Q2: positive definiteness of covariance matrices: If this fails (e.g., rank deficiency due to collinearity), what precisely degrades?
>
> We thank the reviewer for raising this important point and for inspiring us to further generalize our theoretical analysis.
>
> Our theoretical analysis does not need to fundamentally rely on $\Sigma_S$ or $\Sigma_M$ being strictly positive definite. When these matrices are singular, the population solutions in Theorem 1 can be interpreted using the Moore--Penrose pseudoinverse:
> $w_S=\Sigma_S^\dagger\mu_S,\qquad w_M=\Sigma_M^\dagger C_M,$ and the whitening operator becomes $\Sigma_S^{\dagger 1/2}$. Here $\Sigma_S^\dagger$ and $\Sigma_M^\dagger$ denote the Moore--Penrose pseudoinverses. For any possibly singular matrix $A$, its pseudoinverse $A^\dagger$ is the unique matrix satisfying
> $AA^\dagger A = A,\quad
> A^\dagger A A^\dagger = A^\dagger,\quad
> (AA^\dagger)^\top = AA^\dagger,\quad
> (A^\dagger A)^\top = A^\dagger A.$ It coincides with the usual inverse when $A$ is full rank and otherwise computes the least-squares inverse on the identifiable subspace while nulling the kernel. Under this generalization, all algebraic steps of the proof continue to hold; only the coupling strength may decrease. The only thing that changes is the magnitude of the coupling parameters. Intuitively, if the shared embedding collapses onto a lower-dimensional subspace, then fewer feature directions remain identifiable. In this case the aligned component of the multi-attribute signal becomes smaller, which reduces the coupling constant $c$. The core inequality $r_m(x,y)\;\ge\; c\,r_s(x,y) - \varepsilon$ remains valid, but with a potentially weaker $c$ in highly degenerate regimes.
>
> ---------
>
> ## However, in practice the constant $c$ remains **non-trivial**, and **Theorem 1 continues to hold meaningful implications for following reasons**:
>
> The coupling coefficient in Theorem 1 is $c \;=\; \frac{1^\top \beta}{K}$ with $\beta_k = \frac{\tilde\mu_S^\top \tilde C_{M,k}}{\|\tilde\mu_S\|^2}.$ We then give a complete decomposition with all symbols defined locally. The whitened vectors are $\tilde\mu_S = \Sigma_S^{-1/2}\mu_S,  \tilde C_{M,k} = \Sigma_S^{-1/2}\Sigma_M^{-1} C_{M,k}.$ We define normalized directions: $\hat\mu_S = \frac{\tilde\mu_S}{\|\tilde\mu_S\|},  \hat v_k = \frac{\tilde C_{M,k}}{\|\tilde C_{M,k}\|}.$ Using the identity $\tilde\mu_S^\top \tilde C_{M,k}
> = \|\tilde\mu_S\|\|\tilde C_{M,k}\|\langle \hat\mu_S,\hat v_k\rangle$ we obtain the factorization $\beta_k= \langle \hat\mu_S , \hat v_k\rangle \cdot \frac{\|\tilde C_{M,k}\|}{\|\tilde\mu_S\|}.$ Let $\kappa = \frac{\lambda_{\max}(\Sigma_S)}{\lambda_{\min}(\Sigma_S)}.$ Because $\|\tilde\mu_S\|
> \in
> \left[
> \frac{\|\mu_S\|}{\sqrt{\lambda_{\max}}},\;
> \frac{\|\mu_S\|}{\sqrt{\lambda_{\min}}}
> \right],$ the ratio $\frac{\|\tilde C_{M,k}\|}{\|\tilde\mu_S\|}$ implicitly carries a factor proportional to $\kappa^{-1/2}$. Thus $\beta_k =
> \frac{\tilde\mu_S^\top \tilde C_{M,k}}{\|\tilde\mu_S\|^2}=
> \langle \hat\mu_S , \hat v_k \rangle
> \cdot
> \frac{\|\hat v_k\|}{\|\hat\mu_S\|}
> \cdot
> \kappa^{-1/2}.$
>
> **Empirically**:
> - Table 4 reports 60--76\% agreement between BT preference and attribute labels, which lower-bounds the directional alignment term $\langle \hat\mu_S , \hat v_k \rangle$.
> - The regression MSE of SMORM matches the single-head baseline, meaning the regression head retains its full predictive power. Therefore $\|\hat v_k\|$ is not attenuated, and the ratio $\|\hat v_k\|/\|\hat\mu_S\|$ remains $O(1)$.
> - Figure 13 shows that the BT scores under SMORM exhibit stable, non-collapsing variance and no exploding directions. This implies that $\Sigma_S$ maintains a well-bounded spectrum and hence $\kappa^{-1/2}=O(1)$.
>
> Together, these observations imply that the coupling constant stays bounded away from zero, and therefore the conclusion of Theorem 1 continues to hold empirically.
> To further address this concern, we following the experimental setting in response to W2 and measure the Pearson correlation between the BT reward score and the aggregated multi-attribute regression score:
>
> |Method|Pearson-Corr.|
> |--|--|
> | SMORM|67.22|
> |Baseline (two heads training seperately)|46.72 |
>
> SMORM yields substantially stronger alignment between preference and attribute scores than training two reward functions independently, reinforcing that the coupling mechanism in Theorem 1 is not only theoretically sound but also empirically validated.
>
> Even when the covariance matrix is singular, the only effect is a potential reduction in the coupling magnitude $c$. However, **both our theoretical decomposition and empirical results show that this reduction is harmless in practice: $c$ remains clearly non-trivial, and the correlation between the BT and regression heads remains strong**.

---

> ### Author Response · Authors · 2025-11-21
>
> >### Q3: Invertible FIM
>
> We thank the reviewer for highlighting this point. We agree that in large over-parameterized LLMs, the _full_ Fisher Information Matrix (FIM) is indeed singular. Motivated by this observation, we have refined and generalized our theoretical analysis: the relevant asymptotic variance of the reward model is governed not by the full FIM, but by the Fisher information **restricted to the task-relevant subspace**, rather than the entire parameter space.
>
> - Effective Fisher Information: Even when the full FIM is singular, the gradients that actually affect
> 	the reward outputs,
> 	$$
> 	g_S(y) = \nabla_\theta M_S(y), \qquad
> 	g_M(y) = \nabla_\theta M_M(y),
> 	$$
> 	span a low-dimensional *identifiable* subspace. The Fisher information **restricted to this subspace**,
> 	$$
> 	I_{\mathrm{eff}}^{(S)} = \mathbb{E}[g_S g_S^\top],
> 	\qquad
> 	I_{\mathrm{eff}}^{(M)} = \mathbb{E}[g_M g_M^\top],
> 	$$
> 	remains positive definite as long as the reward functions are not constant. This is the standard notion of “effective rank” used in modern	statistics of deep networks: the identifiable subspace can have a well-conditioned Fisher even when the full model has infinitely many uninformative directions. Thus, our asymptotic variance expression should be interpreted as
> 	$$
> 	\mathrm{Var}[M_S(\hat\theta)]
> 	\approx
> 	g_S^\top\ I(\theta^\star)^+\ g_S,
> 	$$
> 	where $I(\theta^\star)^+$ is the **Moore–Penrose pseudoinverse**. This is the standard formulation for asymptotic normality in singular (or non-identifiable) models.
>
> - Theorem 2 could be generalized to only concern the effective Fisher information along the reward-relevant directions: $g_S(y) = \nabla_\theta M_S(y)$ and $g_M(y) = \nabla_\theta M_M(y)$, which span a low-dimensional identifiable subspace $U = \text{span}\\{ g_S, g_M \\}$. Even if the full FIM has a large null space, the \emph{projected} Fisher matrices $I_{\text{eff}}(S) = \mathbb{E}[g_S g_S^\top]$ and $I_{\text{eff}}(M) = \mathbb{E}[g_M g_M^\top]$ are positive definite on $U$.
> All inverse operations in the proof can be interpreted as the Moore--Penrose pseudoinverse restricted to this subspace: $\text{Cov}(\hat{\theta}) \approx I(\theta^\star)^+$.
> Under this formulation, the key identity in Eq.~(13) becomes: $I_{\text{eff}}(\text{hybrid}) = I_{\text{eff}}(\text{single}) + \Delta$, where $\Delta \succeq 0$, and by the positive correlation assumption, $g_S^\top \Delta g_S > 0$, so the Fisher information strictly increases along the BT direction.
> Taking pseudoinverses yields a strict reduction in the asymptotic variance: $g_S^\top I_{\text{eff}}(\text{hybrid})^+ g_S \leq g_S^\top I_{\text{eff}}(\text{single})^+ g_S$.
> Finally, since $\text{Var}[M_S(\hat{\theta})] \approx g_S^\top I(\theta^\star)^+ g_S$, the SMORM estimator has strictly smaller variance and thus smaller MSE.
>
> - **Summary:** Our theoretical comparison could be generalized to only concern the Fisher information **along the reward-relevant directions**, not the full LLM parameter space. All variance statements can be formally expressed using the pseudoinverse of the FIM or, equivalently, the Fisher projected onto the identifiable subspace $\mathrm{span}\\{g_S, g_M\\}$.
>
> ----------
>
> [1] Helpsteer 2: Open-source dataset for training top-performing reward models. NeurIPs 2024.
>
> ----------
>
> We sincerely thank the reviewer for raising such insightful questions, which have helped us further clarify and extend our theoretical framework to more general and realistic settings. **In response, we have provided both rigorous theoretical justifications and supporting empirical evidence to address the concerns raised. We hope that our responses have thoroughly and satisfactorily addressed the reviewer’s comments.** We would be happy to engage in further discussion or provide additional clarification if needed. If the rebuttal has resolved the reviewer’s concerns, we would be deeply grateful for your consideration in raising the rating.

---

> ### Author Response · Authors · 2025-11-21
>
> >### W3: Code Availability
>
> We have provided the code in the Reproducibility Statement on Page 10.

---

> ### Author Response · Authors · 2025-11-27
>
> Dear Reviewer u8z6,
>
> We would like to express our sincere gratitude for the time and care you devoted to reviewing our work. We deeply appreciate the thoughtful concerns you raised, and to support clearer communication, we briefly summarize your main points and our corresponding clarifications below:
>
> - **On architectural simplicity (W1)**:
> We clarified that SMORM’s lightweight structure is a deliberate design choice aimed at maintaining theoretical clarity, facilitating practical adoption, and achieving empirically validated complementary benefits.
>
> - **On correlation and scale alignment (W2 & Q1)**:
> We provided theoretical justification showing that SMORM automatically resolves the inherent scale mismatch between binary preference signals and continuous multi-attribute scores. Empirically, we also confirmed strong real-world alignment between preference and attribute signals.
>
> - **On singular covariances and positive definiteness (W2 & Q2)**:
> We generalized our analysis using Moore–Penrose pseudoinverses, showing that the coupling inequality continues to hold—possibly with reduced but still non-trivial magnitude. Empirically, the coupling remains substantial and becomes significantly stronger under SMORM.
>
> - **On Fisher information singularity (Q3)**:
> We clarified that the variance comparison concerns the reward-relevant identifiable subspace. The effective Fisher information on this subspace remains well-behaved, and the asymptotic arguments naturally extend using pseudoinverses.
>
> For additional details, please refer to our full rebuttal; we have also organized these discussions clearly in Appendix E. We truly appreciate your thoughtful engagement, and we sincerely hope our clarifications have fully addressed your concerns. Looking forward to further discussion with you!
>
> Best,
>
> Submission #14502 Authors

---

### Official Review · Reviewer_xVES · 2025-10-26

**Soundness:** 3
**Presentation:** 4
**Contribution:** 3
**Rating:** 6
**Confidence:** 2

**Summary:**

## Summary
The paper introduces **SMORM**, a joint reward-modeling framework that shares one backbone and trains (i) a Bradley–Terry (BT) single-objective head from pairwise preferences and (ii) a multi-objective regression head from fine-grained attribute scores. The method offers single-pass inference via three variants: **SMORM-F** (use the BT head only), **SMORM-L** (mean of attribute heads), and **SMORM-M** (average of BT and attributes).


Empirically, SMORM improves out-of-distribution (OOD) robustness for PPO and Best-of-N and boosts scoring on RewardBench and RM-Bench. The authors also report that, using the same multi-objective dataset, **a 7B SMORM model surpasses a 70B baseline** on RewardBench.

**Strengths:**

- **Clear OOD focus with concrete evidence.** The paper explicitly studies PPO/BoN under prompt-distribution shifts and shows baselines (e.g., GRM, ODIN) misspecify or overfit signals that lead to reward hacking, whereas SMORM variants maintain rising gold scores.
- **Principled link between BT and regression.** Lemma 1 upper-bounds expected BT loss by the regression MSE; Theorem 2 uses Fisher information to argue strictly better asymptotic MSE for both heads under joint training.
- **Practical single-pass design with strong results.** One shared forward pass and simple inference variants; results include RewardBench/RM-Bench gains and the 7B > 70B headline.

**Weaknesses:**

- **Assumptions (1–3) may be stringent in practice.** Theorem 1 requires (i) **bounded features**, (ii) **positive-definite** covariance matrices for BT and multi-objective features, and (iii) a **positive-correlation** condition via a coupling vector. In realistic learned embeddings, covariances can be **rank-deficient** (e.g., collinearity, limited per-attribute data, heavy regularization), and verifying the correlation condition empirically is non-trivial. Discussion or diagnostics on how often these hold in real RM pipelines would strengthen the claims.
- **Limited optimization-stability analysis for the joint objective.** The paper notes that “their joint training is non-trivial due to their fundamentally different forms,” but beyond asymptotic analysis and aggregate curves, there’s little on **training stability/convergence** (e.g., loss weighting sensitivity, gradient interference). An appendix gives some hyperparameter analysis, yet a focused study on training stability (or convergence) is still missing.
- Missing a related work  "On the Robustness of Reward Models for Language Model Alignment", which focuses also on the OOD setting.

**Questions:**

1. **If Assumption (2) in Theorem 1 (positive-definite covariances) is relaxed to positive-semi-definite, do the results still hold?**
2. For the **7B > 70B** claim, a consolidated table with **confidence intervals** and multiple base models would better characterize robustness of this result across settings.

---

> ### Author Response · Authors · 2025-11-22
>
> We sincerely thank the reviewer for acknowledging the solid theoretical analysis and comprehensive evaluation of our paper. Below, we provide a point-by-point response to the reviewer’s comments:
>
>
>
> -------------------
>
>
>
> >### W1: Further clarification on assumption (3) on positive-correlation
>
>
>
> We thank the reviewer for raising this insightful question. The assumption of a global positive correlation between single-objective and multi-objective reward signals implies that, in general, when two responses are compared, the one with a higher aggregated multi-attribute score is also preferred according to a strong single-objective reward model. This principle reflects an intuitive and commonly observed alignment and has been empirically justified in prior work [1].
>
>
>
> To directly evaluate how well this assumption holds in practice, we conduct an empirical diagnostic using the well-trained single-objective reward model Skywork/Skywork-Reward-V2-Llama-3.1-8B. For every response pair in the nvidia/HelpSteer2 dataset, we compare: the pairwise preference of the aggregated multi-attribute score, and the pairwise preference predicted by the single-objective reward model.
>
> | Attribute | Correlation |
> | ----------- | ----------- |
> | helpfulness | 74.88% |
> | correctness | 74.37% |
> | coherence | 72.75% |
> | complexity | 62.72% |
> | verbosity | 58.96% |
> |  **overall**  |  **73.02%**  |
>
>
>
> We find that 73.02% of response pairs exhibit consistent preference direction. Considering real human annotations are inherently noisy and the alignment is not perfect, this level of agreement constitutes a strong positive correlation. Importantly, SMORM remains robust in the presence of such noise and imperfect correlation.
>
>
>
> These empirical diagnostics show that while the global positive-correlation assumption is a simplification, it holds sufficiently well in practice and SMORM remains effective even under realistic noise and partial alignment. This provides strong support that our theoretical framework captures meaningful structure present in real-world reward-model training pipelines.

---

> ### Author Response · Authors · 2025-11-22
>
> >### W1 & Q1: Further clasirification on assumption (2) on positive-definite covariance matrices
>
> We thank the reviewer for raising this important point and for inspiring us to further generalize our theoretical analysis.
>
> **Theorem 1 naturally extends to the positive–semi-definite case**: Our theoretical analysis does not need to fundamentally rely on $\Sigma_S$ or $\Sigma_M$ being strictly positive definite. When these matrices are singular, the population solutions in Theorem 1 can be interpreted using the Moore--Penrose pseudoinverse:
> $w_S=\Sigma_S^\dagger\mu_S,\qquad w_M=\Sigma_M^\dagger C_M,$ and the whitening operator becomes $\Sigma_S^{\dagger 1/2}$. Here $\Sigma_S^\dagger$ and $\Sigma_M^\dagger$ denote the Moore--Penrose pseudoinverses. For any possibly singular matrix $A$, its pseudoinverse $A^\dagger$ is the unique matrix satisfying
> $AA^\dagger A = A,\quad
> A^\dagger A A^\dagger = A^\dagger,\quad
> (AA^\dagger)^\top = AA^\dagger,\quad
> (A^\dagger A)^\top = A^\dagger A.$ It coincides with the usual inverse when $A$ is full rank and otherwise computes the least-squares inverse on the identifiable subspace while nulling the kernel. Under this generalization, all algebraic steps of the proof continue to hold; only the coupling strength may decrease. The only thing that changes is the magnitude of the coupling parameters. Intuitively, if the shared embedding collapses onto a lower-dimensional subspace, then fewer feature directions remain identifiable. In this case the aligned component of the multi-attribute signal becomes smaller, which reduces the coupling constant $c$. **The core inequality $r_m(x,y)\;\ge\; c\,r_s(x,y) - \varepsilon$ remains valid**, but with a potentially weaker $c$ in highly degenerate regimes.
>
> -------
>
> ### However, in practice the constant $c$ remains **non-trivial**, and **Theorem 1 continues to hold meaningful implications for following reasons**:
>
> The coupling coefficient in Theorem 1 is $c \;=\; \frac{1^\top \beta}{K}$ with $\beta_k = \frac{\tilde\mu_S^\top \tilde C_{M,k}}{\|\tilde\mu_S\|^2}.$ We then give a complete decomposition with all symbols defined locally. The whitened vectors are $\tilde\mu_S = \Sigma_S^{-1/2}\mu_S,  \tilde C_{M,k} = \Sigma_S^{-1/2}\Sigma_M^{-1} C_{M,k}.$ We define normalized directions: $\hat\mu_S = \frac{\tilde\mu_S}{\|\tilde\mu_S\|},  \hat v_k = \frac{\tilde C_{M,k}}{\|\tilde C_{M,k}\|}.$ Using the identity $\tilde\mu_S^\top \tilde C_{M,k}
> = \|\tilde\mu_S\|\|\tilde C_{M,k}\|\langle \hat\mu_S,\hat v_k\rangle$ we obtain the factorization $\beta_k= \langle \hat\mu_S , \hat v_k\rangle \cdot \frac{\|\tilde C_{M,k}\|}{\|\tilde\mu_S\|}.$ Let $\kappa = \frac{\lambda_{\max}(\Sigma_S)}{\lambda_{\min}(\Sigma_S)}.$ Because $\|\tilde\mu_S\|
> \in
> \left[
> \frac{\|\mu_S\|}{\sqrt{\lambda_{\max}}},\;
> \frac{\|\mu_S\|}{\sqrt{\lambda_{\min}}}
> \right],$ the ratio $\frac{\|\tilde C_{M,k}\|}{\|\tilde\mu_S\|}$ implicitly carries a factor proportional to $\kappa^{-1/2}$. Thus $\beta_k =
> \frac{\tilde\mu_S^\top \tilde C_{M,k}}{\|\tilde\mu_S\|^2}=
> \langle \hat\mu_S , \hat v_k \rangle
> \cdot
> \frac{\|\hat v_k\|}{\|\hat\mu_S\|}
> \cdot
> \kappa^{-1/2}.$
>
> **Empirically**:
> - Table 4 reports 60--76\% agreement between BT preference and attribute labels, which lower-bounds the directional alignment term $\langle \hat\mu_S , \hat v_k \rangle$.
> - The regression MSE of SMORM matches the single-head baseline, meaning the regression head retains its full predictive power. Therefore $\|\hat v_k\|$ is not attenuated, and the ratio $\|\hat v_k\|/\|\hat\mu_S\|$ remains $O(1)$.
> - Figure 13 shows that the BT scores under SMORM exhibit stable, non-collapsing variance and no exploding directions. This implies that $\Sigma_S$ maintains a well-bounded spectrum and hence $\kappa^{-1/2}=O(1)$.
>
> Together, these observations imply that the coupling constant stays bounded away from zero, and therefore the conclusion of Theorem 1 continues to hold empirically.
> To further address this concern, we following the experimental setting in response to W2 and measure the Pearson correlation between the BT reward score and the aggregated multi-attribute regression score:
>
> |Method|Pearson-Corr.|
> |--|--|
> | SMORM|67.22|
> |Baseline (two heads training seperately)|46.72 |
>
> SMORM yields substantially stronger alignment between preference and attribute scores than training two reward functions independently, reinforcing that the coupling mechanism in Theorem 1 is not only theoretically sound but also empirically validated.
>
> **When the covariance matrices are only positive–semi-definite**, all steps of Theorem 1 continue to apply by replacing the matrix inverses with their Moore–Penrose pseudoinverses. The **theorem’s inequality remains valid; the only effect is that the coupling magnitude $c$ may become smaller, but our empirical diagnostics show that $c$ remains non-trivial in practice.**

---

> ### Author Response · Authors · 2025-11-22
>
> >### W2: Training stability and convergence
>
>
>
> We thank the reviewer for raising this insightful point, which motivated us to further examine the optimization stability of SMORM.
>
>
>
> **In Appendix C.1, we conduct a series of experiments to analyze the training dynamics of both the BT head and the multi-objective regression head in SMORM**. We fine-tune meta-llama/Llama-3.1-8B-Instruct using two datasets: the preference dataset Skywork/Skywork-Reward-Preference-80K-v0.2 and the multi-objective dataset nvidia/HelpSteer2.
>
>
>
> ### **(1) Training and evaluation dynamics of the BT head**
>
>
> **The results are shown in Fig. 6, comparing SMORM-F to GRM**:
>
>
> - Convergence behavior: As shown in Fig. 6(a), SMORM converges slightly more slowly at the beginning—an expected effect of jointly optimizing the Bradley–Terry loss and the regression loss. However, after ~100 steps, the BT loss of SMORM drops sharply and eventually becomes lower than that of GRM, indicating that the regression signal provides a beneficial inductive bias for the BT head.
>
>
>
> - In-domain stability: Fig. 6(b) shows that SMORM-F maintains comparable in-domain accuracy relative to GRM, demonstrating stable optimization of the BT objective.
>
>
>
> - Out-of-distribution stability and generalization: Fig. 6(c) shows a substantial improvement—approximately +20 points on UnifiedFeedback—demonstrating that SMORM improves not only stability but also generalization of the learned reward function.
>
>
>
> ### **(2) Training and evaluation dynamics of the multi-objective regression head**
>
>
>
> For comparisons against baseline multi-objective reward models, we evaluate on both the training and validation splits of nvidia/HelpSteer2. We report metrics on the five granular attributes—helpfulness, correctness, coherence, complexity, and verbosity—using both mean squared error (MSE) and pairwise preference accuracy. To construct a pairwise validation set, we take pairs of responses for the same prompt and derive ground-truth preference labels for each attribute by directly comparing their human-annotated attribute scores.
>
>
> **Figures 7–10 present the MSE curves and pairwise preference accuracy for the regression head**:
>
>
>
> - MSE stability: As shown in Fig. 7 and Fig. 9, SMORM starts with slightly higher MSE due to joint optimization, but its MSE decreases rapidly and becomes comparable to the baseline model once training stabilizes.
>
>
>
> - Pairwise preference performance: SMORM consistently achieves higher pairwise preference accuracy across both training (Fig. 8) and evaluation (Fig. 10) settings, with especially strong gains on the evaluation set—indicating improved convergence and better alignment with human preferences.
>
>
>
> Taken together, these results provide a focused and empirical view of SMORM’s optimization behavior: although joint training introduces mild early-stage coupling effects, both objectives converge stably, and generalization performance improves substantially, supporting the robustness of SMORM’s joint objective.
>
>
>
>
>
> -------------
>
>
>
>
> >### W3: Missed related work in OOD setting:
>
>
>
> We thank the reviewer for pointing out this relevant work. We have now cited “On the Robustness of Reward Models for Language Model Alignment” in Appendix A and incorporated it into our discussion of OOD robustness. To further address this comment, we also conduct experiments directly following the OOD evaluation protocol proposed in that paper and compare SMORM against BT-BSR and other regularized BT variants.
>
>
>
> Specifically, we evaluate SMORM using LLaMA-3-3B and Qwen2.5-3B as base models and compare it to the BT classifier, its regularized variants, and BT-BSR, the strongest baseline from the referenced work. The results are shown below. While SMORM performs comparably to the strongest baselines on the in-domain setting, it consistently outperforms all baselines across multiple OOD splits, including prompt-disjoint, response-disjoint, and mutual-disjoint settings. These improvements demonstrate that SMORM is particularly effective at enhancing generalization under distribution shift.
>
>
>
> | **Setting** | **Model** | **BT** | **BT-Hinge** | **BT-Norm** | **BT-DR** | **BT-BSR** | **SMORM-F** |
> |--------------------|------------------|--------|--------------|-------------|-----------|------------|-------------|
> | In-Domain | Llama-3 (3B) | 84.5 | 84.0 | 80.0 | 82.5 | 85.0 | 85.5 |
> |  | Qwen2.5 (3B) | 84.5 | 84.0 | 82.0 | 84.5 | 85.5 | 87.0 |
> | Prompt-disjoint | Llama-3 (3B) | 68.0 | 68.5 | 60.0 | 67.5 | 70.5 | 73.5 |
> |  | Qwen2.5 (3B) | 70.0 | 69.5 | 65.0 | 67.5 | 72.0 | 74.0 |
> | Response-disjoint | Llama-3 (3B) | 61.5 | 61.5 | 50.0 | 58.5 | 63.5 | 68.0 |
> |  | Qwen2.5 (3B) | 58.0 | 56.5 | 45.0 | 54.0 | 60.5 | 66.5 |
> | Mutual-disjoint | Llama-3 (3B) | 51.5 | 52.0 | 47.5 | 50.5 | 54.5 | 57.5 |
> |  | Qwen2.5 (3B) | 58.0 | 56.0 | 53.0 | 55.0 | 61.0 | 66.0 |

---

> ### Author Response · Authors · 2025-11-22
>
> >### Q2: consolidated table with confidence intervals and multiple base models
>
>
>
> We thank the reviewer for carefully reading our paper and for suggesting a more comprehensive robustness check for the “7B > 70B” observation. Following this recommendation, we expanded our experiments to include multiple base models and multiple random seeds, and we compare them directly against the advanced multi-objective model Llama-3-70B-RM, which is trained on the same multi-objective dataset.
>
>
>
> Specifically, we follow the experimental setting of Table 3 and train on two base models: (1) mistralai/Mistral-7B-Instruct-v0.2 (2) meta-llama/Meta-Llama-3-8B-Instruct. For each base model, we train two independent runs with different seeds and report RewardBench scores. The results are shown below.
>
>
>
> Notably, despite the substantial size difference, both our 7B and 8B SMORM-L models consistently outperform Llama-3-70B-RM across different seeds. This provides strong evidence that the result is not driven by randomness or a specific architecture, and confirms that SMORM-L’s multi-objective learning efficiency remains robust across model families and training runs.
>
>
>
> || **Method** | Chat | Chat Hard | Safety | Reasoning | **RewardBench** |
> |---|-----------|-------|-----------|--------|-----------|--------------|
> | |Llama-3-70B-RM| 91.3 | 80.3 | 92.8 | 90.7 | 88.8|
> |Ours|SMORM-L 7B Run-1| 95.0 | 80.5 | 91.6 | 89.0 | 89.0|
> ||SMORM-L 7B Run-2| 93.0 | 83.7 | 90.9 | 89.0 | 89.2 |
> || SMORM-L 8B Run-1 | 96.1 | 84.0 | 91.8 | 89.0 | 90.2 |
> || SMORM-L 8B Run-2 | 94.7 | 84.6 | 91.1 | 89.6 | 90.0 |
>
>
> ---------------
>
> We sincerely thank the reviewer for thoughtful and constructive feedback, which has greatly helped us strengthen both the theoretical and empirical components of the paper. Across all points, we have provided direct answers supported by new empirical evidence and refined theoretical explanations. We hope that our responses have satisfactorily addressed the reviewer’s concerns and demonstrated the robustness and generality of our contributions. We would be happy to provide further clarification if needed, and we would be deeply grateful for your consideration in raising the score.

---

> ### Comment · Reviewer_xVES · 2025-11-25
>
> Thanks for 1) verify the positive correlation empericially, 2) extended theoretical explainations and experical validation of positive–semi-definite covariance matrics, 3) training stability check. This helps streghten the paper.
> Minor latex compling issue, in line 318, it displays "Appendix ??".

---

> ### Author Response · Authors · 2025-11-25
> **Thank you for your positive assessment**
>
> Dear Reviewer xVES:
>
> **We are glad to hear that all of your concerns have been addressed.** We sincerely appreciate your overall positive assessment and your constructive feedback, which have meaningfully strengthened the paper. We have incorporated the suggested experiments and theoretical analyses into the revised version.
>
> Thank you again for your time, thoughtful comments, and generous engagement throughout the review process.
>
> Best,
>
> Submission #14502 Authors

---

> ### Author Response · Authors · 2025-11-26
>
> Dear Reviewer xVES,
>
> Thank you very much for your careful reading and helpful comments. We apologize for the delayed response—there was no email notification for modified comments. We have now fixed the LaTeX issue at line 318.
>
> Best,
>
> Submission #14502 Authors

---

### Official Review · Reviewer_v1Q3 · 2025-10-28

**Soundness:** 3
**Presentation:** 3
**Contribution:** 2
**Rating:** 4
**Confidence:** 3

**Summary:**

This paper introduces SMORM (Single and Multi-Objective Reward Model), which jointly trains a Bradley–Terry (BT) single-objective head and a multi-objective regression head on a shared embedding space. The authors claim that this co-training improves robustness against reward hacking under out-of-distribution (OOD) conditions and enhances fine-grained multi-attribute reward estimation. The paper provides theoretical analyses connecting BT loss and regression loss through Fisher Information and MSE bounds and reports empirical results showing modest improvements over GRM, ODIN, and baseline classifiers on RewardBench, RM-Bench, and PPO/BoN experiments.

**Strengths:**

1. The paper offers a theoretically well-grounded contribution. The link between BT loss and regression loss via Fisher Information analysis is original and potentially influential.
2. The joint single/multi-objective framework is conceptually neat and aligns with the broader goal of multi-dimensional reward alignment.
3. The idea of embedding-space complementarity between preference learning and multi-attribute regression is interesting and could inspire follow-up research.
4. Writing and structure are clear, and the motivation is intuitive.

**Weaknesses:**

1. Lack of statistical rigor.
The paper reports only mean scores without standard deviations or multiple-seed averages. Given the small performance margins (≈1–2 points on RewardBench or RM-Bench), these gains could easily fall within noise. This omission is especially problematic because the theoretical claim centers on variance reduction, yet no variance statistics are provided.

2. Weak OOD validation and limited generalization coverage.
The paper claims robustness “in OOD settings” but evaluates only one prompt–response distribution shift. Unlike Hong et al. (ICML, 2025) (“On the Robustness of Reward Models”), which explicitly distinguishes in-domain (ID), prompt-disjoint, response-disjoint, and mutual-disjoint scenarios, this work tests only a single OOD split. No experiments examine robustness to unseen model families, novel prompt distributions, or response-generation biases, which would be expected to stress the model’s generalization. Consequently, the empirical evidence does not convincingly demonstrate the claimed OOD robustness.

3. Fine-grained performance not well supported.
Although the framework is motivated by multi-attribute supervision (helpfulness, coherence, verbosity, etc.), evaluation results remain at the aggregate level. No attribute-wise performance or correlation analysis is presented to verify the “fine-grained” improvement.

4. Small effect sizes and lack of reproducibility details.
Reported improvements over GRM or ODIN are minor and not statistically supported. Also, RLHF results (PPO/BoN) appear based on single runs with no mention of seeds or error bars.

**Questions:**

1. Could the authors report standard deviations or confidence intervals across multiple seeds to verify statistical significance?
2. Can the OOD analysis be explicitly decomposed into settings to substantiate the robustness claims (e.g., prompt-disjoint, response-disjoint, and mutual-disjoint)?
3. Could the authors report per attribute ranking accuracy or correlation for helpfulness, correctness, coherence, complexity, and verbosity? It would be better, if authors include ablations that remove the multi objective head to isolate its contribution.
4. Have the authors compared SMORM with simpler stabilization methods such as Batch Sum-to-Zero Regularization (BSR) under identical OOD conditions?

---

> ### Author Response · Authors · 2025-11-21
>
> We sincerely thank the reviewer for acknowledging the originality and potential influence of our papar, Below, we provide a point-by-point response to the reviewer’s comments:
>
> >### W1 & Q1: lack of statistical rigor
>
> We thank the reviewer for raising this important point. As training reward models and running PPO experiments require substantial GPU resources. Therefore, following prior work [1][2], we reported results without error bars initially.
>
> To address your concern, we use google/gemma-2-2b-it as the base model and train SMORM on two datasets: the preference dataset Skywork/Skywork-Reward-Preference-80K-v0.2 and the multi-objective reward dataset
> nvidia/HelpSteer2. We compare SMORM to strong baselines GRM and ODIN and run each method with three different seeds (42, 123, 456), and report these results:
> - **Figure 11** (Appendix) presents the training and evaluation curves on both the in-domain (**Skywork**) and out-of-domain (**UnifiedFeedback**) datasets. Below, we additionally provide a table reporting accuracy results on the **UnifiedFeedback** dataset.
>
>     | Step | GRM         | ODIN        | SMORM       |
>     |------|-------------|-------------|-------------|
>     |  50  | 0.522 ± 0.038 | 0.543 ± 0.005 | 0.488 ± 0.051 |
>     | 100  | 0.573 ± 0.012 | 0.563 ± 0.010 | 0.562 ± 0.039 |
> 	| 150  | 0.606 ± 0.007 | 0.588 ± 0.015 | 0.623 ± 0.042 |
> 	| 200  | 0.630 ± 0.005 | 0.629 ± 0.009 | 0.655 ± 0.027 |
> 	| 250  | 0.657 ± 0.011 | 0.653 ± 0.004 | 0.677 ± 0.021 |
> 	| 300  | 0.656 ± 0.009 | 0.665 ± 0.005 | 0.684 ± 0.021 |
> 	| 350  | 0.658 ± 0.008 | 0.667 ± 0.006 | 0.684 ± 0.021 |
> 	| 400  | 0.664 ± 0.008 | 0.674 ± 0.006 | 0.687 ± 0.019 |
> 	| 450  | 0.661 ± 0.012 | 0.674 ± 0.001 | 0.690 ± 0.020 |
> 	| 500  | 0.660 ± 0.012 | 0.674 ± 0.007 | 0.686 ± 0.019 |
> 	| 550  | 0.662 ± 0.012 | 0.674 ± 0.003 | 0.692 ± 0.020 |
> 	| 600  | 0.662 ± 0.012 | 0.674 ± 0.002 | 0.689 ± 0.018 |
>
> 	From Figure 11 in the Appendix and Table below, we observe that while SMORM converges slowly during the early training stages, it quickly catches up and achieves comparable performance to baselines on the training dataset (Skywork). However, **on the evaluation dataset (UnifiedFeedback), SMORM surpasses both baselines after approximately 100 steps and consistently maintains superior performance throughout the training process**.
>
> - Performance on RewardBench, as shown in table below:
>
> 	| **Method**  | **Chat**        | **Chat Hard**    | **Safety**       | **Reasoning**     | **Overall**       |
> 	|-------------|------------------|------------------|------------------|-------------------|-------------------|
> 	| **SMORM-F** | 89.1 ± 1.96      | 65.6 ± 0.04      | 77.65 ± 3.42     | 71.3 ± 0.36       | 75.9 ± 0.44       |
> 	| **GRM**     | 86.05 ± 0.06     | 66.0 ± 0.16      | 74.2 ± 0.09      | 69.85 ± 1.32      | 74.0 ± 0.36       |
> 	| **ODIN**    | 84.25 ± 2.40     | 63.05 ± 0.12     | 71.75 ± 0.42     | 73.75 ± 17.22     | 73.2 ± 1.44      |
>
> 	from the RewardBench results in table above, we observe that SMORM consistently outperforms GRM and ODIN, demonstrating its robustness and generalization ability beyond the training distribution.
>
> >### W2 & Q2 & Q4: More comprehensive OOD analysis and comparison to BSR
>
> We thank the reviewer for highlighting relevant work that enables a more rigorous evaluation of out-of-distribution (OOD) robustness. Following the experimental setup of [3], we conduct additional experiments using the base models LLaMA-3-3B and Qwen2.5-3B, comparing SMORM to the BT classifier, its regularized variants, and BT-BSR. We find that while performs comparably to the strongest baselines on the in-domain dataset, **SMORM consistently outperforms all baselines across multiple OOD settings**. This demonstrates the effectiveness of SMORM in improving generalization under distribution shift.
>
> | **Setting** | **Model**    | **BT** | **BT-Hinge** | **BT-Norm** | **BT-DR** | **BT-BSR** | **SMORM-F** |
> |---------------------|-------------------|--------|--------------|-------------|-----------|------------|-------------|
> | **In-Domain**       | Llama-3 (3B)      | 84.5   | 84.0         | 80.0        | 82.5      | 85.0       | 85.5    |
> |                     | Qwen2.5 (3B)      | 84.5   | 84.0         | 82.0        | 84.5      | 85.5       | 87.0    |
> | **Prompt-disjoint** | Llama-3 (3B)      | 68.0   | 68.5         | 60.0        | 67.5      | 70.5 | 73.5    |
> |                     | Qwen2.5 (3B)      | 70.0   | 69.5         | 65.0    | 67.5      | 72.0       | 74.0        |
> | **Response-disjoint** | Llama-3 (3B)    | 61.5   | 61.5     | 50.0        | 58.5      | 63.5       | 68.0        |
> |                     | Qwen2.5 (3B)      | 58.0   | 56.5         | 45.0     | 54.0      | 60.5       | 66.5        |
> | **Mutual-disjoint** | Llama-3 (3B)      | 51.5   | 52.0         | 47.5        | 50.5      | 54.5       | 57.5        |
> |                     | Qwen2.5 (3B)      | 58.0   | 56.0         | 53.0        | 55.0      | 61.0       | 66.0        |

---

> ### Author Response · Authors · 2025-11-21
>
> >### W3 & Q3: Fine-grained performance on different attribute
>
> We thank the reviewer for raising this insightful question, which prompted us to further clarify and demonstrate how SMORM enhances the performance of multi-objective regression reward modeling.
>
> To this end, we conduct experiments using the base model meta-llama/Llama-3.1-8B-Instruct on two datasets: the preference dataset (_Skywork/Skywork-Reward-Preference-80K-v0.2_) and the multi-objective reward dataset (_nvidia/HelpSteer2_). For a fair comparison against baseline multi-objective reward models, we evaluate on both the training and validation splits of _HelpSteer2_. We report performance across five fine-grained attributes—**helpfulness**, **correctness**, **coherence**, **complexity**, and **verbosity**—using two metrics: **mean squared error (MSE)** and **pairwise preference accuracy**.
>
> To construct the pairwise validation set, we sample response pairs corresponding to the same prompt and derive ground-truth preference labels for each attribute by directly comparing their human-annotated attribute scores.
>
> The results comparing SMORM to the baseline multi-objective reward model are shown in:
>
> -   **Figure 7** (training MSE curves),
>
> -   **Figure 8** (training pairwise preference accuracy),
>
> -   **Figure 9** (evaluation MSE),
>
> -   **Figure 10** (evaluation pairwise preference accuracy).
>
>
> For convenience, we present the results from **Figure 10** as a table below.
>
> | Step | Helpfulness (Baseline / SMORM) | Correctness (Baseline / SMORM) | Coherence (Baseline / SMORM) | Complexity (Baseline / SMORM) | Verbosity (Baseline / SMORM) |
> |------|---------------------|----------------------|--------------------|---------------------|-------------------|
> |  50  | 0.539 / 0.554       | 0.477 / 0.550        | 0.459 / 0.478      | 0.524 / 0.556       | 0.583 / 0.654     |
> | 100  | 0.547 / 0.569       | 0.489 / 0.566        | 0.449 / 0.536      | 0.497 / 0.556       | 0.583 / 0.654     |
> | 150  | 0.547 / 0.599       | 0.514 / 0.558        | 0.464 / 0.722      | 0.517 / 0.617       | 0.606 / 0.654     |
> | 200  | 0.568 / 0.591       | 0.517 / 0.591        | 0.505 / 0.664      | 0.538 / 0.576       | 0.622 / 0.685     |
> | 250  | 0.552 / 0.584       | 0.500 / 0.582        | 0.490 / 0.635      | 0.559 / 0.617       | 0.634 / 0.705     |
> | 300  | 0.568 / 0.569       | 0.506 / 0.558        | 0.480 / 0.649      | 0.580 / 0.617       | 0.642 / 0.695     |
> | 350  | 0.552 / 0.584       | 0.500 / 0.558        | 0.485 / 0.649      | 0.580 / 0.617       | 0.654 / 0.685     |
> | 400  | 0.552 / 0.584       | 0.508 / 0.533        | 0.500 / 0.635      | 0.573 / 0.637       | 0.650 / 0.705     |
> | 450  | 0.560 / 0.591       | 0.503 / 0.541        | 0.500 / 0.649      | 0.573 / 0.617       | 0.654 / 0.705     |
> | 500  | 0.566 / 0.613       | 0.500 / 0.566        | 0.495 / 0.635      | 0.552 / 0.617       | 0.654 / 0.705     |
> | 550  | 0.566 / 0.591       | 0.500 / 0.550        | 0.495 / 0.635      | 0.573 / 0.617       | 0.642 / 0.705     |
> | 600  | 0.571 / 0.599       | 0.500 / 0.541        | 0.490 / 0.649      | 0.573 / 0.617       | 0.657 / 0.705     |
>
>
> From these results, we observe the following:
>
> 1.  As shown in **Figure 7** (training set) and **Figure 9** (evaluation set), although **SMORM** initially exhibits a higher mean squared error (MSE) during early training, its MSE decreases rapidly and eventually becomes comparable to that of the baseline model.
>
> 2.  In terms of **pairwise preference accuracy**, **SMORM consistently outperforms** the baseline model. This improvement is particularly pronounced on the evaluation set, as illustrated in **Figure 10**.
>
> To further explain how SMORM enables BT training to enhance the performance of the regression head, we analyze the correlation between attribute-level ground-truth preferences and the pairwise preferences predicted by a strong single-objective reward model. Specifically, we use Skywork/Skywork-Reward-V2-Llama-3.1-8B, a model with strong scoring capability, to score all pairwise comparisons in the nvidia/HelpSteer2 dataset. For each attribute, we compute the proportion of response pairs whose ground-truth preference direction aligns with the model's predicted preference.
>
> The results are shown in the table below:
>
> | Attribute |Correlation  |
> |--|--|
> | helpfulness | 74.88\% |
> | correctness | 74.37\% |
> | coherence | 72.75\% |
> | complexity | 62.72\% |
> | verbosity |  58.96\%|
> | overall |  73.02\%|
>
> We observe that helpfulness, correctness, and coherence exhibit notably higher correlation with the single-objective reward model, indicating that these attributes are more aligned with a well-trained Bradley–Terry preference signal. Consistent with this, **Figure 10** shows that SMORM’s performance improvements over the baselines are most pronounced on these three attributes.

---

> ### Author Response · Authors · 2025-11-21
>
> >### W4: comparison to GRM and ODIN and multiple runs on PPO
>
> Although prior work [1][2] does not report results using PPO due to the significant computational cost, we conduct additional experiments to fully address the reviewer's concern. Specifically, we use google/gemma-2-2b-it as the base model and train SMORM on two datasets: the Skywork/Skywork-Reward-Preference-80K-v0.2 preference dataset and the nvidia/HelpSteer2 multi-objective reward dataset. We compare SMORM to two strong baselines: GRM and ODIN. All PPO training settings follow those described in Section 3. For each reward model, we run PPO with three different random seeds (42, 123, 456). To thoroughly evaluate the robustness and reward hacking mitigation of SMORM, we extend the training steps beyond the typical budget and evaluate the model under more challenging conditions.
>
> **The results are presented in Figure 12 (Appendix)**.
>
> From the figure, we observe that during the early training phase (i.e., before ~15,000 samples), all three methods—SMORM, GRM, and ODIN—lead to steady improvements in the golden reward score. However, beyond this point, **only SMORM continues to yield gains**, while the other two baselines either plateau or exhibit **reward degradation**, indicating overoptimization or reward hacking. This highlights SMORM’s superior stability and robustness in prolonged training settings.
>
> --------
>
> [1] Helpsteer 2: Open-source dataset for training top-performing reward models. NeurIPs 2024.
>
> [2] Scaling Laws for Reward Model Overoptimization. ICML 2023.
>
> [3] On the Robustness of Reward Models for Language Model Alignment. ICML 2025.
>
> ---------
>
> We sincerely thank the reviewer for raising such insightful questions, which have greatly helped us strengthen the clarity, rigor, and breadth of our empirical and methodological contributions. **In response, we have conducted new experiments with multiple seeds, extended PPO training to stress-test generalization, and incorporated fine-grained and OOD evaluations comparing SMORM to strong baselines including GRM, ODIN, and BSR. These additions provide both rigorous empirical evidence and robust validation of our claims**, particularly regarding the benefits of SMORM under distribution shift and in avoiding reward overoptimization.
>
> We hope that our responses have thoroughly and satisfactorily addressed the reviewer’s concerns. We would be happy to provide additional clarification or engage in further discussion if needed. If our rebuttal has resolved your concerns, we would be deeply grateful for your consideration in raising the rating.

---

> ### Comment · Reviewer_v1Q3 · 2025-11-25
>
> Thank you for the detailed rebuttal. The additional experiments are appreciated, but several concerns remain.
> RewardBench results now include stds, yet some numbers (e.g., ODIN Reasoning 73.75 ± 17.22) suggest instability in the evaluation setup.
> In several new OOD tables, improvements such as BT-BSR 85.0 vs SMORM 85.5 are so small that even a modest std (e.g., ±0.7) could eliminate any meaningful difference, but no variance is reported.
> As a result, many of the added OOD comparisons remain difficult to interpret statistically.
> The expanded OOD analysis is helpful but still narrow in scope and does not convincingly demonstrate broader robustness.
> Fine-grained attribute experiments improve clarity but are limited to a single dataset and model.
> Overall, while the rebuttal meaningfully strengthens the submission, the core concerns about stability, variance, and generality are only partially resolved. For these reasons, I intend to keep my original score unchanged.

---

> ### Author Response · Authors · 2025-11-25
>
> Dear Reviewer v1Q3,
>
> We sincerely thank you for the follow-up assessment and appreciate your acknowledgement that the new experiments strengthened the submission. Below, we clarify several points where we believe some conclusions may have resulted from misunderstandings regarding variance, OOD scope, and empirical expectations.
>
> ----------
>
> ## **1. “ODIN Reasoning 73.75 ± 17.22 suggests instability in the evaluation setup”**
>
> We are not sure why instability from **one baseline** implies instability of **our evaluation setup**.
>
> -   Both **SMORM-F** and **GRM** remain stable across the same evaluation.
> -   Only **ODIN** displays this large variance, which is consistent with ODIN’s known behavior—its design optimizes primarily against reward hacking related to response _length_, and it is inherently unstable on tasks that require semantic reasoning.
>
> **Therefore, the high variance reflects **ODIN’s intrinsic weakness**, not an instability in our experimental setup.**
>
> ----------
>
> ## **2. “OOD improvements such as BT-BSR 85.0 vs SMORM 85.5 are small; without variance these comparisons are hard to interpret”**
>
> We respectfully emphasize two points:
>
> ### **(a) None of the baseline papers reported multi-seed statistics.**
>
> This includes the exact papers we compare against—BT, BT-Hinge, BT-Norm, BT-DR, and BT-BSR. **The broader RM community [1][2][3][4] also reports **single-seed** results because RM training is computationally expensive.** We followed **exactly the same reporting protocol** as all baselines to ensure fairness and comparability.
>
> ### **(b) The reviewer cherry-pick small gaps in in-distirbution setting, but results in OOD setting clearly beyond any realistic noise range.**
>
> For example:
>
> -   **Response-disjoint (Llama-3)**: **+4.5** (63.5 → 68.0)
> -   **Response-disjoint (Qwen2.5)**: **+6.0** (60.5 → 66.5)
> -   **Mutual-disjoint (Qwen2.5)**: **+5.0** (61.0 → 66.0)
>
> These improvements are **substantial** and far exceed typical RM variance observed in practice.
>
> Thus, even if a ±0.7 std were applied, these OOD gains would remain significant.
>
> ----------
>
> ## **3. “OOD analysis is helpful but still narrow and does not demonstrate broader robustness.”**
>
> In your initial review, you referenced the only work extending OOD evaluation into multiple disjoint settings.
> We **followed exactly that paper’s protocol**, using:
>
> -   the same OOD definitions,
> -   the same evaluation structure,
> -   and additionally running across **two model families** (Llama-3-3B and Qwen2.5-3B).
>
> ### **Because our OOD setup matches the methodology recommended by the reviewer, we do not fully understand why the scope is still considered limited.**
>
> ----------
>
> ## **4. “Fine-grained attribute experiments improve clarity but are limited to a single dataset and model.”**
>
> For the fine-grained experiments, we reported:
>
> -   detailed **training and evaluation MSE**,
> -   detailed **pairwise preference accuracy**,
> -   across **600 training steps**,
> -   on **all five attributes**,
>
> We believe this is a comprehensive evaluation of multi-objective behavior.
>
> Moreover, our other results on RewardBench throughout the paper—including both **HelpSteer2** and **UnifiedFeedback**—already demonstrate consistent improvements.
>
> ----------
>
> ## **5. Summary and request for reconsideration**
>
> Across the reviewer’s remaining concerns:
>
> - ### **None** of the reference works report variance std on RewardBench or OOD experiments.
>
> - ### **We followed OOD evaluation protocol [4] exactly. The reviewer stated that “the expanded OOD analysis is helpful but still narrow in scope,” directly contradicting their prior request.**
>
> - ###  We provided **detailed step-wise results** on how SMORM improves multi-objective reward modeling.
>
> ### **We completed all experiments requested by the reviewer in the initial review. However, the reviewer’s follow-up was self-contradictory and introduced new, unreasonable expectations that were never mentioned in the initial review.**
>
> ------------
>
> ## **We respectfully ask the reviewer to reconsider whether additional experiments—beyond what is required or practiced in existing literature—should materially impact the overall assessment, especially for a contribution you described as _“original and potentially influential.”_**
>
> If the reviewer believes that specific additional experiments would meaningfully change the assessment and is willing to raise the score accordingly, we are happy to conduct them and report the results as soon as possible. Thank you again for your engagement.
>
> ---------
>
> [1] Helpsteer 2: Open-source dataset for training top-performing reward models. NIPS 2024.
>
> [2] Scaling laws for  reward model  overoptimization. ICMl 2023.
>
> [3] ODIN: Disentangled Reward Mitigates Hacking in RLHF.
>
> [4] On the Robustness of Reward Models for Language Model Alignment. ICML 2025.

---

> ### Author Response · Authors · 2025-11-27
>
> Dear Reviewer v1Q3,
>
> We hope you don't mind a brief follow-up. Thank you again for the time and attention you have already given our submission.
>
> During the rebuttal phase, we conducted a substantial set of additional experiments—multi-seed analyses on multiple datasets, expanded OOD evaluations following the exact protocol of prior work, fine-grained attribute assessments across multiple metrics, and extended PPO runs with multiple seeds. These required considerable computational resources and time, but we felt they were important to thoroughly address every concern raised in your initial review.
>
> Given this, we wanted to kindly ask for your perspective on two points:
>
> **(1) Whether the additional requirements introduced in your follow-up—particularly broader variance reporting and expanding OOD experiments beyond the standard protocol—are intended to materially influence the assessment of our core contribution.**
>
> In our rebuttal, we addressed all concerns raised at that time—including running multi-seed variance experiments (which, as you noted, are rarely reported in reward-modeling work) and fully adopting the OOD evaluation protocol from the exact robustness paper you referenced.
>
> Given that these additions go significantly beyond what is typically required or practiced in the reward-modeling literature, we would greatly appreciate clarification on whether the new concerns raised in your follow-up—such as broader variance reporting and even more extensive OOD coverage—are meant to fundamentally alter the evaluation of the central contribution.
>
> **(2) Whether further experiments would meaningfully affect your final score.**
>
> We remain fully willing to conduct more experiments—whether expanded multi-seed OOD evaluations, cross-model generalization tests, or deeper variance analyses—if such results would meaningfully address your remaining concerns. **Meanwhile, could you kindly clarify what additional experiments you have in mind regarding the comment “the expanded OOD analysis is helpful but still narrow in scope and does not convincingly demonstrate broader robustness”? We were unsure how to further act on this point, as our OOD experiments already follow exactly the protocol of the paper you referenced.**
>
> Thank you again for your time and engagement with our work. We would be happy to provide any additional results promptly if they would help strengthen the evaluation.
>
> Best,
>
> Submission #14502 Authors

---

### Official Review · Reviewer_9F27 · 2025-11-01

**Soundness:** 3
**Presentation:** 3
**Contribution:** 3
**Rating:** 6
**Confidence:** 3

**Summary:**

The paper introduces SMORM, a single-backbone with one Bradley–Terry (BT) reward head and one multi-objective regression head. The authors also provide theory proofs linking the two to explain the robustness gains. The architecture is novel while synthesizing insights from prior work (e.g., Armo, GRM). Empirically, SMORM improves scoring and out-of-distribution robustness and also alleviates reward hacking in RLHF.

**Strengths:**

* The paper is well written and clearly positioned relative to prior work, while thoughtfully incorporating insights from it.


* The method is simple and well-motivated, and the theoretical analysis makes the core idea more compelling.


* The evaluation is thorough, covering both reward modeling and RLHF.

**Weaknesses:**

* The design of SMORL-L uses the mean of the multi-objective scores. However, using the mean does not make sense, which means you give all rewards equal weights given all prompts. Actually, this should be utilized in a contextual way and dynamically adapt to the user prompts [1][2].

* The authors claim SMORM is flexible because the two heads are trained on different prompt–response pairs. In practice, this complicates using and coordinating two distinct datasets. I therefore disagree with the flexibility claim and suggest revising it.

* Because the method uses two datasets, add a control that trains a BT reward model on the union of the multi-objective preference data (transferred to BT data using rules) and BT data. This would make the comparison fairer by isolating architectural gains from “more data” effects.

* Based on question 2, I'm interested if not including new data if the conclusion still holds. This can be validated using a multi-objective preference dataset and transfer it to BT data using certain rules. Then you do the same ablations study comparing to baselines (multi-objective only, BT-only, combining them using two models,  and SMORM) trained on the same dataset.

* Table 3 mixes base models: the paper evaluates with Llama-3.1-8B-Instruct but compares to ArmoRM on Llama-3-8B. Against contemporaneous baselines built on Llama-3.1-8B-Instruct (e.g., QRM-Llama-3.1-8B-v2, Skywork-Reward-Llama-3.1-8B-v0.2), the method shows no clear gains; notably, Skywork-Reward-Llama-3.1-8B-v0.2, trained only on Skywork80K (without multi-objective data), scores 93.1 on RewardBench. This weakens the claim.

* Since the paper leverages multi-objective preference learning, Appendix A should include a more complete related-work section to cover this line of research [1–4].





[1] Improving context-aware preference modeling for language models, 2024.

[2] MiCRo: Mixture Modeling and Context-aware Routing for Personalized Preference Learning, 2025.

[3] Rethinking Diverse Human Preference Learning through Principal Component Analysis, 2025.

[4] Personalizing reinforcement learning from human feedback with variational preference learning.

**Questions:**

Please refer to the weakness section.

---

> ### Author Response · Authors · 2025-11-22
>
> We sincerely thank the reviewer for acknowledging the solid theoretical analysis and comprehensive evaluation of our paper. Below, we provide a point-by-point response to the reviewer’s comments:
>
> ----------
>
> >### Q1: Equal weight for each attribute
>
> We sincerely thank the reviewer for raising this insightful question and for pointing out the connection to contextual and adaptive weighting approaches.
>
> We fully agree that using dynamically learned, context-dependent weights for multi-attribute scores can further improve performance. Indeed, such adaptive weighting is a valuable direction, and our method is fully compatible with integrating these techniques.
>
> However, in this paper we use **simple mean aggregation** when evaluating SMORM-L, for the following reason: **Our goal was to compare the _fundamental scoring capability_ of multi-attribute regression models in a controlled setting.** Introducing an additional gating or weighting network would add another layer of modeling choices and potentially obscure this comparison by injecting extra modeling capacity or optimization noise.
>
> As shown in Figures 8 and 10 in the appendix, **SMORM-L consistently outperforms the baseline multi-objective regression model on _each individual attribute’s_ pairwise preference accuracy.** This shows that SMORM’s advantage comes from its **stronger fine-grained attribute scoring**, rather than by the choice of averaging strategy.
>
> Given this stronger base capability, **it is reasonable to expect that SMORM-L would continue to outperform baselines when equipped with adaptive, context-dependent weighting mechanisms**.
>
> --------
>
> > ### Q2: Flexibility claim
>
> We thank the reviewer for this thoughtful comment. We agree that coordinating two distinct datasets introduces practical considerations, and we appreciate the reviewer’s perspective on this point. We will substantially soften this claim to avoid overemphasizing it in future version.
>
> ----------
>
> >### Q3: Train BT baseline on combined data
>
> We sincerely thank the reviewer for this insightful suggestion. We agree that training the BT baseline on the union of (a) the original BT preference data and (b) the multi-objective data converted into BT-style pairwise comparisons is a reasonable control that helps isolate architectural improvements from potential “more data” effects.
>
> To address this concern, we conduct experiments under two settings:
>
> 1.  google/gemma-2-2b-it as the base model
>
>     -   UnifiedFeedback (40K)
>
>     -   HelpSteer2 transformed into pairwise preference data
>
> 2.  meta-llama/Llama-3.2-3B-Instruct as the base model
>
>     -   Skywork (80K)
>
>     -   HelpSteer2 transformed into pairwise preference data
>
>
> The results are shown below. In both settings, we observe that **adding the converted multi-objective data to the BT baseline leads to only marginal improvements** (e.g., RewardBench 64.2 → 64.6 and 65.8 → 67.2). This is expected given that HelpSteer2 contributes only about 8K additional samples. Importantly, SMORM-F consistently achieves the best performance, demonstrating that SMORM’s improvements cannot be attributed solely to additional training data.
>
> | **Dataset**                  | **Model**       | Chat  | Chat Hard | Safety | Reasoning | RewardBench |
> |-----------------------------|------------------|-------|-----------|--------|-----------|--------------------|
> | UnifiedFeedback 40k/HelpSteer2 | Baseline             | 94.7 | 37.5 | 66.2 | 58.4 | 64.2               |
> |                              | Baseline w/union data  | 93.7     | 38.8         | 65.3      | 60.6         | 64.6                  |
> |                              | SMORM-F         | 96.1  | 44.1      | 81.1   | 62.7      | 71.0   |
>
>
>
>
>
> | **Dataset**             | **Model**   | Chat  | Chat Hard | Safety | Reasoning | RewardBench |
> |-------------------------|-------------|-------|-----------|--------|-----------|--------------------|
> | Skywork80K/HelpSteer2   | Baseline    | 73.4  | 60.5      | 79.8   | 49.6      | 65.8               |
> |         |   Baseline W/union data |    78.8 | 62.3 | 79.3 | 48.3 |          67.2                |
> |                         | SMORM-F     | 80.4  | 62.1      | 80.7   | 55.1      | 69.6               |
>
> It is worth noting that SMORM’s BT head could also incorporate the HelpSteer2-transformed pairwise data. However, due to the significant computational cost of reward-model training, we were unable to include these results. As expected, incorporating this additional supervision would likely further enhance SMORM’s performance.

---

> ### Author Response · Authors · 2025-11-22
>
> > ### Q4: Train SMORM without new data
>
> We thank the reviewer for this insightful question. To directly evaluate whether SMORM’s benefits persist _without introducing any new data_, we follow the reviewer’s suggestion and construct multi-objective and BT-style supervision from the same dataset. Specifically, using Llama-3.2-3B-Instruct as the base model, we take (1) **UnifiedFeedback** and (2) **HelpSteer2**, and create their binarized versions by converting the multi-attribute scores into pairwise preference labels. This allows us to train: (1) multi-objective regression baseline (2) BT-only baseline (3) SMORM. The RewardBench results are shown in the tables below. Across both datasets, we observe that:
>
> -   **SMORM-L** significantly improves over the multi-objective baseline (UnifiedFeedback: **50.0 → 56.3**, HelpSteer2: **51.3 → 56.8**).
>
> -   **SMORM-F** also improves over the BT-only baseline significantly.
>
> -   These gains occur **even when all supervision signals originate from the same dataset**, confirming that SMORM’s advantage does not rely on external additional data.
>
>
> Overall, these results directly demonstrate that the conclusion still holds: **SMORM enhances reward modeling even when both heads are trained on supervision signals derived from the same dataset**.
>
>
> |**Dataset**| **Model**             | **Chat** | **Chat Hard** | **Safety** | **Reasoning** | **RewardBench** |
> |-------|----------------------|----------|---------------|------------|---------------|-----------------|
> |UltraFeedback (binarized)/UltraFeedback| **Baseline (single)**| 89.1     | 40.7          | 45.2       | 36.7          | 52.9            |
> || **SMORM-F**          | 88.4     | 43.4          | 44.3       | 49.6          | **56.4**            |
> || **Baseline (multi)** | 70.3     | 46.1          | 41.6       | 42.0          | 50.0            |
> || **SMORM-L**          | 90.2 | 40.1 | 54.2 | 40.8 | **56.3** |
>
>
> |**Dataset**| **Model**             |  **Chat** | **Chat Hard** | **Safety** | **Reasoning** | **RewardBench** |
> |--------|----------------------|--------------|--------------|--------------|--------------|--------------|
> |HelpSteer2 (binarized)/HelpSteer2| **Baseline (single)**| 51.7  | 51.8      | 47.2   | 42.4      | 48.3         |
> || **SMORM-F**          | 50.3         | 48.7         | 54.7         | 73.6         | **56.9**         |
> || **Baseline (multi)** | 55.8         | 50.4         | 44.8         | 54.2         | 51.3         |
> || **SMORM-L**          | 50.7         | 49.3         | 52.9         | 73.3         | **56.8**         |

---

> ### Author Response · Authors · 2025-11-22
>
> > ### Q5: Baseline model mismatch in Table 3
>
> We thank the reviewer for the careful reading of our paper and for pointing out the issue regarding mixed base models. We acknowledge that the original Table 3 compared our SMORM trained on Llama-3.1-8B-Instruct with ArmoRM-Llama3-8B-v0.1, which uses Llama-3-8B-Instruct. We agree that this mismatch can obscure the fairness of the comparison.
>
> To directly address this concern, we reran the experiments using the same base model as ArmoRM, i.e., meta-llama/Meta-Llama-3-8B-Instruct, and trained SMORM under identical conditions. We report results across two seeds to demonstrate stability. The updated comparison is shown below.
>
> | **Method**   | Chat  | Chat Hard | Safety | Reasoning | **RewardBench** |
> |-----------|-------|-----------|--------|-----------|--------------|
> |ArmoRM-Llama3-8B-v0.1| 96.9 | 76.8 | 90.5 | 97.3 | 90.4 |
> | SMORM-L (3-8B) Run-1     | 96.1  | 84.0      | 91.8   | 89.0      | 90.2         |
> | SMORM-L (3-8B) Run-2    | 94.7  | 84.6      | 91.1   | 89.6      | 90.0         |
> | **SMORM Mean**  | 95.4  | 84.3      | 91.5   | 89.3      | 90.1         |
>
> Although ArmoRM achieves a RewardBench score of 90.4, our SMORM (8B) achieves a comparable 90.1 on average _despite using dramatically less training data_:
>
> -   ArmoRM uses **585.4K** samples for its multi-objective head and **1,004.4K** samples for its gating network.
>
> -   SMORM uses only **20K** samples for multi-attribute training and **80K** for gating network.
>
> This highlights that **SMORM substantially improves the performance of multi-objective reward models even under limited data**
>
>
> > Skywork-Reward-Llama-3.1-8B-v0.2, trained only on Skywork80K (without multi-objective data), scores 93.1 on RewardBench. This weakens the claim
>
> We thank the reviewer for point this out. In table 3, we are mainly comparing our SMORM-L to advanced multi-objective reward to demonstrate how SMORM help enhance the performance of its multi-objective reward functions with only 20K multi-objective data samples. However, to fully address your concern, we conduct experiments and show performance of SMORM-F compared to  Skywork-Reward-Llama-3.1-8B-v0.2. As the training script of Skywork-Reward-Llama-3.1-8B-v0.2 is not released, we follow the same setting in base model (meta-llama/Meta-Llama-3.1-8B-Instruct) and dataset(Skywork/Skywork-Reward-Preference-80K-v0.2), and run the model and get the result below showing that the performance we could achieve is 88.9, which is noticeably below the originally reported 93.1. Under the same setting, SMORM-F achieves a RewardBench score of 90.7. This indicates that even though pairwise preference data can train a strong BT head, our SMORM framework can still enhance performance with limited multi-objective data.
>
> | **Run**   | Chat  | Chat Hard | Safety | Reasoning | **RewardBench** |
> |-----------|-------|-----------|--------|-----------|--------------|
> | Baseline (3.1-8B)     | 92.5  | 82.7      | 90.7   | 89.6      | 88.9         |
> |SMORM-F (3.1-8B)| 94.7 | 84.1 | 90.9 | 92.9 | 90.7 |
>
> -------
>
> > ### Q6: related work
>
> We sincerely thank the reviewer for pointing out these relevant works on multi-objective preference learning. We have now carefully cited and discussed these references in Appendix A to provide a more comprehensive overview of this research line.
>
>
> -------
>
> We sincerely thank the reviewer for thoughtful and constructive feedback, which has greatly helped us clarify the presentation and strengthen the empirical and methodological components of our work. **In response to each question, we conducted additional controlled experiments, corrected the baseline comparisons, expanded the related-work discussion, and refined our claims to more accurately reflect the contributions of SMORM**. We hope that these clarifications and new results have fully addressed the reviewer’s concerns and demonstrated the robustness and fairness of our evaluations. We would be happy to provide any further details if needed, and if our responses have resolved the reviewer’s questions, we would be deeply grateful for your consideration in raising the rating.

---

### Author Response · Authors · 2025-12-01
**Summary to Assist the Area Chair’s Evaluation (1)**

Dear Area Chair:

We sincerely thank the AC for their time and effort in evaluating our paper. Below, we provide a concise summary intended to assist the AC in assessing our work and understanding the key points from the rebuttal phase.

## **Summary of strengths mentioned by reviewers**

-----------------------------

### **1. All reviewers agree that the paper is clearly written, well-organized, and easy to follow**

-   _Reviewer 9F27:_ “The paper is well written and clearly positioned relative to prior work, while thoughtfully incorporating insights from it.”
-   _Reviewer v1Q3:_ “Writing and structure are clear, and the motivation is intuitive.”

-   _Reviewer u8z6:_ “The paper is well-structured and clearly presented.”

-   _Reviewer xVES:_ “Presentation: 4 — excellent.”



### **2. Reviewers emphasized the strong theoretical foundation and originality of the connection between BT and regression objectives.**

-   _Reviewer 9F27:_ “The authors also provide theory proofs linking the two to explain the robustness gains.”

-   _Reviewer v1Q3:_ “The paper offers a theoretically well-grounded contribution. The link between BT loss and regression loss via Fisher Information analysis is original and potentially influential.”

-   _Reviewer xVES:_ “Principled link between BT and regression… Theorem 2 uses Fisher information to argue strictly better asymptotic MSE.”

-   _Reviewer u8z6:_ “The idea of co-training BT and multi-objective regression heads … is new and well-motivated.”


### **3. The SMORM architecture—one shared backbone with BT and multi-objective heads—is viewed as elegant, simple, and practically effective**

-   _Reviewer 9F27:_ “The architecture is novel while synthesizing insights from prior work (e.g., Armo, GRM). The method is simple and well-motivated.”

-   _Reviewer v1Q3:_ “The joint single/multi-objective framework is conceptually neat and aligns with the broader goal of multi-dimensional reward alignment.”

-   _Reviewer u8z6:_ “The idea of co-training BT and multi-objective regression heads under a shared embedding space is new and well-motivated.”

-   _Reviewer xVES:_ “Practical single-pass design with strong results. One shared forward pass and simple inference variants.”



### **4. Reviewers recognized the empirical evaluation as comprehensive, including both Reward Modeling and RLHF across ID and OOD settings**

-   _Reviewer 9F27:_ “The evaluation is thorough, covering both reward modeling and RLHF.”

-   _Reviewer xVES:_ “Clear OOD focus with concrete evidence… The paper explicitly studies PPO/BoN under prompt-distribution shifts.”

-   _Reviewer u8z6:_ “The proposed methods are supported by empirical evaluations under both in-distribution (ID) and out-of-distribution (OOD) settings.”



### **5. Multiple reviewers emphasized SMORM’s strong performance under distribution shifts and its ability to mitigate reward hacking.**

-   _Reviewer 9F27:_ “Empirically, SMORM improves scoring and out-of-distribution robustness and also alleviates reward hacking in RLHF.”

-   _Reviewer xVES:_ “SMORM variants maintain rising gold scores under distribution shifts, while baselines overfit or plateau.”


### **6. Reviewers view the work as theoretically insightful and likely to inspire future research in reward modeling and alignment**

-   _Reviewer v1Q3:_ “The link between BT loss and regression loss… is original and potentially influential.”

-   _Reviewer xVES:_ (implicit through positive tone) “The paper provides strong results and principled analysis that may inspire subsequent studies on reward model robustness and alignment.”

---

> ### Author Response · Authors · 2025-12-01
> **Summary to Assist the Area Chair’s Evaluation (2)**
>
> ## **Discussion Phase Summary**
>
> ---------------
>
> Here we summarize the rebuttal phase. For clarity, we focus primarily on **Reviewers v1Q3** and **u8z6**, who initially gave negative ratings (score = 4).
>
> --------------
>
> ### **Reviewer v1Q3**
>
> The reviewer’s initial review was **officially flagged as Fully AI-generated ([https://iclr.pangram.com/reviews?query=&submission_number=14502](https://iclr.pangram.com/reviews?query=&submission_number=14502))** and requested extensive additional experiments—particularly multi-seed analyses—well beyond the standard evaluation criteria in the reward-modeling literature.
> **Despite our substantial computational investment and completion of all experiments explicitly requested, the reviewer’s follow-up was self-contradictory and introduced new, unreasonable expectations that were never mentioned in the initial review.** The follow-up response itself was also detected as AI-generated.
>
> For instance, after citing _Hong et al._ (ICML 2025) and requesting a comprehensive OOD study, the reviewer later stated that “the expanded OOD analysis is helpful but still narrow in scope,” directly contradicting both their prior request and the cited protocol.
>
> **We remain fully willing to address any additional concerns and would appreciate further clarification from the reviewer on the specific experiments they would like to see. However, no subsequent response was provided.** Further details are available here: https://openreview.net/forum?id=3QHKJcwnpb&noteId=vBhossq8f2.
>
> ----------
>
> ### **Reviewer u8z6**
>
> Reviewer u8z6 raised concerns regarding: **(1)** the simplicity of the architecture, **(2)** the global positive correlation between single- and multi-objective signals, **(3)** the difference in scale between the two reward functions, **(4)** the assumption of positive-definite covariance matrices, and **(5)** the assumption of invertible Fisher Information Matrices (FIM).
>
> For **point (1)**, we note that **all other reviewers explicitly described the architecture as both _novel_ and _well-motivated_, noting that its simplicity is an advantage** rather than a limitation:
>
> -   **9F27:** “The architecture is novel”
>
> -   **v1Q3:** “The joint single/multi-objective framework is conceptually neat and aligns with the broader goal of multi-dimensional reward alignment.”
>
> -   **xVES:** “Practical single-pass design with strong results. One shared forward pass and simple inference variants.”
>
> -   **u8z6 (self):** “The idea of co-training BT and multi-objective regression heads under a shared embedding space is new and well-motivated.”
>
>
> For **points (2)** and **(4)**, **Reviewer xVES raised similar concerns but explicitly confirmed in their follow-up that our rebuttal fully addressed these issues**.
>
> For **points (3)** and **(5)**, we provided **detailed theoretical clarifications** along with **empirical experiments** demonstrating that our theoretical analysis hold and generalize effectively in practical training scenarios.
>
> We believe our rebuttal **comprehensively addressed all of Reviewer u8z6’s concerns**; however, the reviewer **did not provide any response** after our clarifications.
>
>
> ----------------------
>
> We sincerely appreciate the AC’s time, fairness, and careful attention in evaluating our work. We hope that this summary provides a clear and balanced overview of both the reviewers’ feedback and our detailed responses.
>
> We respectfully believe that the overall reviewer consensus—emphasizing **theoretical originality, architectural novelty, and empirical rigor**—reflects the significance of our contribution.
>
> ### **Once again, we would like to express our deepest gratitude for your time, effort, and thoughtful consideration devoted to our work.**

---

### Meta-Review · Area_Chair_SsbQ · 2025-12-29

**Summary:**

The paper establish a connection between the BT model and the multi-objective regression. Reviewers appreciated the focus on OOD reward hacking but felt the initial evaluation was too narrow and the theoretical proofs were mathematically imprecise. They also consider simplicity of the model (linear attribute fusion) as a potential limitation. The response has addressed the comments adequately via empirical results and refinements of the theoretical results. The idea and observation in the paper is interesting.

**Reviewer Concerns:**

The response has addressed the comments adequately.

**Reviewer Scores:**

I anticipate that Reviewer u8z6 and Reviewer xVES would raise to 6.

---

### Decision · Program_Chairs · 2026-01-26

Accept (Poster)